# On the Convergence of Stochastic Smoothed Multi-Level Compositional Gradient Descent Ascent

**Xinwen Zhang**
Temple University
Philadelphia, PA, USA
ellenz@temple.edu

**Hongchang Gao** *
Temple University
Philadelphia, PA, USA
hongchang.gao@temple.edu

## Abstract

Multi-level compositional optimization is a fundamental framework in machine learning with broad applications. While recent advances have addressed compositional minimization problems, the stochastic multi-level compositional minimax problem introduces significant new challenges—most notably, the biased nature of stochastic gradients for both the primal and dual variables. In this work, we address this gap by proposing a novel stochastic multi-level compositional gradient descent-ascent algorithm, incorporating a smoothing technique under the nonconvex-PL condition. We establish a convergence rate to an $(\epsilon, \epsilon/\sqrt{\kappa})$-stationary point with improved dependence on the condition number at $O(\kappa^{3/2})$, where $\epsilon$ denotes the solution accuracy and $\kappa$ represents the condition number. Moreover, we design a novel stage-wise algorithm with variance reduction to address the biased gradient issue under the two-sided PL condition. This algorithm successfully enables a translation from and $(\epsilon, \epsilon/\sqrt{\kappa})$-stationary point to an $\epsilon$-stationary point. Finally, extensive experiments validate the effectiveness of our algorithms.

## 1 Introduction

This paper investigates the stochastic multi-level compositional minimax optimization problem:

$$\min_{x\in\mathbb{R}^{d_x}} \max_{y\in\mathbb{R}^{d_y}} f(G(x), y) , \qquad (1)$$

where $f(G(x), y) = \mathbb{E}[f(G(x), y; \zeta)]$ and $\zeta$ denotes a random variable. The function $G(x) \triangleq g^{(K)}(\cdots(g^{(1)}(x)))$ is a $K$-level compositional function with $K > 1$, where each *inner-level* function $g^{(k)}(\cdot) = \mathbb{E}[g^{(k)}(\cdot; \xi^{(k)})]$ depends on the random sample $\xi^{(k)}$ for $k \in \{1, \cdots, K\}$. The function $f(\cdot, \cdot)$ is referred to as the *outer-level* objective. In this paper, we consider the general nonconvex–PL setting, where $f(G(x), y)$ is nonconvex in the primal variable $x$ and satisfies the Polyak-Lojasiewicz (PL) condition with respect to the dual variable $y$.

Multi-level compositional optimization has emerged as a vital framework in machine learning, with broad applications across numerous domains. In meta-learning, it enhances model adaptability across tasks [12, 22]; in finance, it supports risk-averse portfolio optimization under uncertainty [3, 20]; and in reinforcement learning, it aids policy evaluation and decision refinement [8, 24]. The widespread impact of the multi-level compositional structure highlights its importance in handling complex and structured optimization problems. Moreover, the scope of multi-level compositional optimization extends naturally to the minimax setting, with applications in areas such as deep AUC maximization [33], multi-instance learning [40], and multi-objective learning [19], etc. Despite the importance of these applications, the stochastic multi-level compositional minimax optimization problem remains

---

*Corresponding author

39th Conference on Neural Information Processing Systems (NeurIPS 2025).

Table 1: The comparison of convergence rate between our algorithms and existing stochastic compositional minimax algorithms.

| Algorithms | Convergence Rate | Assumption | Level |
|---|---|---|---|
| SCGDA [17] | $O(\kappa^4/\epsilon^3)$ | Nonconvex-strongly-concave | Two-level |
| SCGDAM [33] | $O(\kappa^4/\epsilon^4)$ | Nonconvex-strongly-concave | Two-level |
| CODA-Primal [9] | $O(\kappa^4/\epsilon^4)$ | Nonconvex-strongly-concave | Two-level |
| NSTORM [26] | $O(\kappa^3/\epsilon^3)$ | Nonconvex-strongly-concave | Two-level |
| Smoothed-SMCGDA-VR (Thm. 4.1) | $O(\kappa^{3/2}/\epsilon^3)$ | Nonconvex-PL | Multi-level |
| Onestage-SMCGDA-VR (Thm. C.1) | $O(\kappa^3/\epsilon^3)$ | Nonconvex-PL | Multi-level |
| Stagewise-SMCGDA-VR (Thm. C.2) | $O(\kappa^6/\epsilon)$ | Two-sided-PL | Multi-level |

largely underexplored. This gap in the literature motivates our study, which aims to develop an effective algorithmic solution for this challenging class of problems.

Solving multi-level compositional problems is challenging, even in the *minimization* setting. In particular, when the inner-level functions are nonlinear, the stochastic gradient is no longer an unbiased estimator of the full gradient. Recent research has proposed new algorithms to address this issue. Notably, [31] introduced a $K$-level stochastic compositional gradient descent algorithm, and subsequent efforts [5, 23, 35] have developed algorithms specifically tailored to address the biased characteristics inherent in the stochastic multi-level compositional framework. Unfortunately, these *minimization-targeted* algorithms cannot directly address the stochastic multi-level compositional *minimax* optimization problem in Eq. (1), as the stochastic gradients for **both primal and dual variables are biased estimators** in stochastic multi-level compositional minimax problems—posing greater algorithmic and theoretical challenges.

In addition, although [17, 26, 9] investigate the two-level compositional minimax problem, it remains within the classical minimax framework and does not consider more advanced techniques that can improve convergence. In contrast, recent progress in classical minimax optimization has demonstrated that smoothed techniques can significantly improve convergence. For instance, [36] proposed a smoothed alternating gradient method for general nonconvex–concave problems, which achieves superior performance compared to conventional approaches. Building on this, [30] further applied this technique to the nonconvex-PL setting—a milder condition than strong concavity—and showed that stochastic smoothed techniques yield improved complexity bounds with better dependence on the condition number $\kappa$. However, despite the clear benefits of smoothed techniques in traditional minimax optimization, its application to multi-level compositional minimax problems remains unexplored. This observation motivates a key question: **Can smoothed techniques be effectively integrated into the multi-level compositional framework to improve convergence performance?**

Addressing this question is not straightforward and presents substantial algorithmic and theoretical challenges. On the one hand, while existing studies [36, 30] demonstrate that smoothing techniques are effective for *unbiased* stochastic gradient estimators, the biased nature of stochastic gradients for both the primal and dual variables in Eq. (1) introduces uncertainty regarding the effectiveness of applying such techniques. It remains unclear whether their use may lead to additional convergence issues. Therefore, it is essential to develop new algorithms that can accommodate biased gradient estimators and guarantee convergence when using smoothing techniques for Eq. (1). On the other hand, as demonstrated in [30], smoothed algorithms typically guarantee an $(\epsilon_1, \epsilon_2)$-stationary point, rather than a standard $\epsilon$-stationary point, which are defined in Definition 3.5. This necessitates a translation between the two measures. While such a translation introduces *negligible* iteration complexity in classical minimax problems when using an *unbiased* gradient estimator as shown in [30], there is no known algorithm capable of performing this translation in the context of multi-level compositional minimax optimization. In particular, **it remains unclear how to design a translation algorithm without degrading the iteration complexity of the smoothed algorithm in the presence of a multi-level compositional structure.** Therefore, these challenges motivate us to address the problem through the following contributions:

- We develop a novel smoothed multi-level compositional minimax optimization algorithm for Eq. (1) by leveraging the variance reduction technique to mitigate the biased gradient estimator issue, and establish a convergence rate of $O(\kappa^{3/2}/\epsilon^3)$ to an $(\epsilon, \epsilon/\sqrt{\kappa})$-stationary point. Compared to existing algorithms, our method achieves a better dependence on the

condition number $\kappa$: improving over the $O(\kappa^3)$ rate of standard two-level compositional minimax algorithms.

- To bridge the gap between an $(\epsilon, \epsilon/\sqrt{\kappa})$-stationary point and a standard $\epsilon$-stationary point, we further propose a stage-wise variance-reduced algorithm for Eq. (1) under the two-sided PL condition. We show that the algorithm achieves a convergence rate of $O(1/\epsilon^2)$ to an $\epsilon$-stationary point. As a result, the iteration complexity from the translation is dominated by the complexity of finding an $(\epsilon, \epsilon/\sqrt{\kappa})$-stationary point.

- Meanwhile, we obtain two additional results, which may be of independent interest: the convergence rates for the multi-level compositional minimax problem under the nonconvex-PL and two-sided-PL assumptions without using the smooth technique, as summarized in Table 1.

- We conduct extensive experiments to validate the effectiveness of our proposed algorithms, demonstrating superior performance compared to existing baselines.

## 2 Related Work

### 2.1 Stochastic Compositional Minimization Optimization

Recently, a general class of stochastic compositional gradient descent methods [29, 18, 32, 15, 16] was developed for two-level compositional minimization problems and established convergence rates for nonconvex loss functions. Aiming to address practical problems with a more general stochastic compositional structure, the stochastic *two-level* compositional problem has been extended to the stochastic *multi-level* compositional problem. Stochastic multi-level compositional learning has various applications, including multi-step model-agnostic meta-learning [12], the stochastic training of graph neural networks [6], the neural networks with batch-normalization [25], etc. Consequently, a series of stochastic multi-level compositional minimization algorithms [31, 2, 5, 35, 23, 13, 14] have been developed to solve this important problem. Notably, [31] introduced the first stochastic multi-level compositional gradient descent algorithm. Then, [2] employed a moving-average estimator, and [5] used the STORM variance-reduction estimator [7] for each inner-level function, achieving a convergence rate of $O(1/\epsilon^4)$. Later, [35] improved the sample complexity to $O(1/\epsilon^3)$ by applying the SPIDER variance-reduction technique [11, 27] to both the inner-level function and Jacobian matrix at each level. Nevertheless, the large batch size required by this method makes it impractical for large-scale models, and the learning rate must be sufficiently small to maintain Lipschitz continuity of the variance-reduced gradient. By applying the STORM variance-reduction approach to both the function value and its Jacobian matrix at each level, [23] developed a convergence rate of $O(1/\epsilon^3)$ for the stochastic multi-level compositional problem with a mini-batch size of $O(1)$. More recently, for the first time, [13] showed that the variance-reduction estimator is not necessary for the Jacobian matrix in each level to achieve a convergence rate of $O(1/\epsilon^3)$. However, these stochastic multi-level compositional algorithms focus exclusively on minimization problems and therefore cannot be directly applied to multi-level compositional minimax problems.

### 2.2 Stochastic Compositional Minimax Optimization

Stochastic compositional minimax optimization [17, 33, 9, 26, 37, 38] has attracted increasing attention due to its important applications in machine learning. To solve the two-level compositional minimax problem, [17] developed the first compositional minimax algorithm based on the mini-batch compositional gradient, achieving a convergence rate of $O(\kappa^4/\epsilon^4)$ for nonconvex-strongly-concave loss functions. [33] incorporated the momentum technique to reduce the mini-batch size to $O(1)$ while achieving the same convergence rate as [17]. Similarly, [9] used a variance-reduced estimator for the inner-level function, also reducing the mini-batch size to $O(1)$ and achieving the same convergence rate as [17]. [26] introduced the STORM technique for estimating the inner-level function and gradient, achieving a convergence rate of $O(\kappa^3/\epsilon^3)$. Recently, [9] claimed to achieve a convergence rate of $O(\kappa^2/\epsilon^3)$. However, this convergence rate is established with respect to the stationary point of the *Moreau envelope* of the primal function, rather than that of the original primal function. As a result, it corresponds to the convergence rate for a strongly-convex–strongly-concave loss function, rather than for nonconvex–strongly-concave or nonconvex-PL loss functions. More recently, [37] developed the first stochastic *multi-level* compositional minimax algorithm for nonconvex-strongly-concave loss functions in the federated learning setting. However, its convergence rate $O(1/\epsilon^4)$ is suboptimal compared to the multi-level compositional minimization algorithm. On the other hand, the smoothed technique was first introduced for nonconvex–concave minimax problems in [36], where the convergence rate of a full alternating gradient descent ascent method was established. Later, [30]

extended this technique to the nonconvex–PL setting and further investigated the relationship between two stationarity measures. However, none of these algorithms are equipped to handle the challenges posed by *multi-level* compositional minimax problems, which remain largely unexplored.

## 3 Preliminaries

### 3.1 Notations

We begin by simplifying the complex formulation in Eq. (1) to facilitate analysis:

$$G^{(k)}(x) = g^{(k)}(G^{(k-1)}(x)), \qquad \nabla G^{(k)}(x) = \nabla G^{(k-1)}(x)\nabla g^{(k)}(G^{(k-1)}(x)), \qquad (2)$$

where $k \in \{1, \cdots, K\}$, $G^{(0)}(x) = x$, and $G(x) = G^{(K)}(x)$.

The partial gradients of the objective function can then be expressed as follows:

$$\nabla_x f(G(x), y) = \nabla G^{(K)}(x)\nabla_1 f(G^{(K)}(x), y), \qquad \nabla_y f(G(x), y) = \nabla_2 f(G^{(K)}(x), y). \qquad (3)$$

Following prior works [36, 30], we introduce an auxiliary variable $z$ alongside the primal variable $x$ as part of the smoothed technique, and define the smoothed loss function as:

$$f_\omega(G(x), y; z) = f(G(x), y) + \frac{\omega}{2}\|x - z\|^2, \qquad (4)$$

where $\omega > 0$ is a constant and $f_\omega(G(x), y; z)$ is strongly convex with respect to $x$ by selecting an appropriate $\omega$. Using the smoothed loss, we can derive stochastic estimators of the compositional gradients with respect to the primal and dual variables at the $t$-th iteration:

$$\nabla_x f_\omega(\cdot, \cdot; \cdot; \hat{\xi}_t, \zeta_t) = \nabla g^{(1)}(x_t; \xi_t^{(1)})\nabla g^{(2)}(g^{(1)}(x_t; \xi_t^{(1)}); \xi_t^{(2)}) \cdots \nabla g^{(K-1)}(g^{(K-2)}(\cdot; \xi_t^{(K-2)}); \xi_t^{(K-1)})$$
$$\times \nabla g^{(K)}(g^{(K-1)}(\cdot; \xi_t^{(K-1)}); \xi_t^{(K)})\nabla_1 f(g^{(K)}(\cdot; \xi_t^{(K)}), y_t; \zeta_t) + \omega(x_t - z_t),$$
$$\nabla_y f_\omega(\cdot, \cdot; \cdot; \hat{\xi}_t, \zeta_t) = \nabla_y f(g^{(K)}(\cdot; \xi_t^{(K)}), y_t; \zeta_t), \qquad (5)$$

where $\hat{\xi}_t = \{\xi_t^{(1)}, \xi_t^{(2)}, \cdots, \xi_t^{(K)}\}$.

### 3.2 Assumptions

We next introduce the following standard assumptions, which are commonly used in stochastic compositional optimization [17, 26, 9, 38, 37, 30].

**Assumption 3.1.** *(Smoothness):*

- *For any $k \in \{1, 2, \cdots, K\}$, $g^{(k)}(\cdot)$ and $g^{(k)}(\cdot; \xi)$ are $C_g$-Lipschitz continuous, $\nabla g^{(k)}(\cdot)$ and $\nabla g^{(k)}(\cdot; \xi)$ are $L_g$-Lipschitz continuous, where $C_g > 0$ and $L_g > 0$;*
- *$f(\cdot, \cdot)$ and $f(\cdot, \cdot; \zeta)$ are $C_f$-Lipschitz continuous, $\nabla f(\cdot, \cdot)$ and $\nabla f(\cdot, \cdot; \zeta)$ are $L_f$-Lipschitz continuous, where $C_f > 0$ and $L_f > 0$.*

**Assumption 3.2.** *(Variance):*

- *For any $k \in \{1, \cdots, K\}$, the stochastic gradients $\nabla g^{(k)}(\cdot; \xi^{(k)})$ and $\nabla f(\cdot, \cdot; \zeta)$ have upper bounded variance $\sigma^2$, where $\sigma > 0$.*

**Assumption 3.3.** *(PL Condition):*

- *For any fixed $x \in \mathbb{R}^{d_x}$, the maximization problem $y^* = \max_{y \in \mathbb{R}^{d_y}} f(G(x), y)$ has a non-empty solution set and a finite optimal value. Moreover, for all $x \in \mathbb{R}^{d_x}$, there exists a constant value $\mu > 0$ such that $\|\nabla_y f(G(x), y)\|^2 \geq 2\mu(f(G(x), y^*) - f(G(x), y))$.*

Here, we define $\ell = \max\{L_f, C_g^{2K}L_f + C_f \sum_{k=0}^{K-1} L_f C_g^{K-1+k}\}$, and $\kappa = \frac{\ell}{\mu}$ denotes the condition number. Then, when $\omega > \ell$, $f_\omega(G(x), y; z)$ is strongly convex with respect to $x$. We also introduce the following definitions.

**Definition 3.4.** *(Two-sided PL Condition):*

- *$f(x, y)$ satisfies the two-sided PL condition, if there exist constants $\mu_x > 0$ and $\mu_y > 0$ such that $f(\cdot, y)$ is $\mu_x$-PL for any $y \in \mathbb{R}^{d_y}$, and $-f(x, \cdot)$ is $\mu_y$-PL for any $x \in \mathbb{R}^{d_x}$.*

**Definition 3.5.** *(Stationarity measures):*

- $(x, y)$ *is an* $(\epsilon_1, \epsilon_2)$-*stationary point of* $f(\cdot, \cdot)$, *if* $\|\nabla_x f(G(x), y)\| \leq \epsilon_1$ *and* $\|\nabla_y f(G(x), y)\| \leq \epsilon_2$.

- $x$ *is an* $\epsilon$-*stationary point of* $\Phi(\cdot)$, *if* $\|\nabla \Phi(x)\| \leq \epsilon$, *where* $\Phi(x) = f(G(x), y^*)$ *and* $y^* = \arg\max_{y \in \mathbb{R}^{d_y}} f(G(x), y)$.

### 3.3 Challenges

**From the algorithmic design perspective, one of the primary challenges in incorporating smoothed techniques is managing the intrinsic bias of stochastic gradients for both the primal and dual variables**. Specifically, as shown in Eq.(3), the partial gradient regarding the dual variable $y$ relies on the *stochastic estimator* of $K$-level function $G^{(K)}(\cdot)$, while that regarding the primal variable $x$ depends on the *stochastic estimator* of both $G^{(K)}(\cdot)$ and $\nabla G^{(K)}(\cdot)$. In the stochastic setting, however, computing the stochastic estimator for both the $k$-th level function and its corresponding gradient introduces bias, as illustrated below:

$$\mathbb{E}[g^{(k)}(g^{(k-1)}(\cdot; \xi^{(k-1)}); \xi^{(k)})] \neq G^{(k)}(\cdot) ,$$
$$\mathbb{E}[\nabla_x g^{(k-1)}(\cdot; \xi^{(k-1)}) \nabla_{g^{(k-1)}} g^{(k)}(g^{(k-1)}(\cdot; \xi^{(k-1)}); \xi^{(k)})] \neq \nabla_x G^{(k)}(\cdot) . \tag{6}$$

As a result, **the stochastic gradients with respect to both primal and dual variables are biased estimators of the full gradient.** Moreover, as shown in Eq. (6), the estimation biases accumulate across all compositional levels when estimating both the inner-level functions and their gradient. This accumulation of bias introduces greater complexity compared to the two-level case and raises concerns about whether the deeper compositional structure might undermine the effectiveness of smoothed techniques, *as all existing smoothed minimax methods handle deterministic gradients or unbiased stochastic gradients.*

**From the theoretical analysis perspective, a major challenge arises from the gap between different stationarity measures induced by smoothed techniques**. As demonstrated in [30], a translation is required from an $(\epsilon_1, \epsilon_2)$-stationary point to an $\epsilon$-stationary point. In standard minimax settings, this can be achieved by applying a stochastic gradient descent-ascent algorithm to the auxiliary problem $\min_{x \in \mathbb{R}^{d_x}} \max_{y \in \mathbb{R}^{d_y}} f(x, y) + \ell \|x - \tilde{z}\|^2$, where $\tilde{z}$ is the output of the smoothed algorithm. Owing to the fact that this formulation satisfies the *the PL condition in both $x$ and $y$*, with an iteration complexity of $\tilde{O}(1/\epsilon^2)$. Therefore, if the cost of this translation remains lower than that of the smoothed algorithm itself, it does not affect the overall complexity. However, for multi-level compositional minimax problems, **there do not exist algorithms for handling the two-sided PL condition to complete the translation, and it is unclear whether the iteration complexity of the translation is smaller than that of the smoothed algorithm or not**. In particular, the existing study [31] showed that the standard compositional gradient descent algorithm can only achieve a convergence rate with an exponential dependence on the number of levels, even for strongly convex loss functions. As a result, the complexity of the translation phase could dominate the overall complexity. Therefore, it remains unclear whether there exists an efficient algorithm to translate from an $(\epsilon_1, \epsilon_2)$-stationary point to an $\epsilon$-stationary point for multi-level compositional minimax problems.

## 4 Algorithm 1: Smoothed-SMCGDA-VR

### 4.1 Algorithmic Design

To address the smoothed loss in Eq. (4), we design a novel algorithm, named stochastic smoothed multi-level compositional gradient descent ascent with variance reduction (Smoothed-SMCGDA-VR), as presented in Algorithm 1. To mitigate the accumulation of bias at each compositional level, our method incorporates a STORM-like variance-reduced estimator. Specifically, for each inner-level function $g^{(k)}(\cdot)$, where $k \in \{1, \dots, K\}$, we apply a recursive step that updates the estimator $h^{(k)}$ while controlling variance. This variance reduction technique is also employed for the stochastic gradients: $\nabla_x f_\omega(\cdot, \cdot; \cdot; \hat{\xi}_t, \zeta_t)$ and $\nabla_y f_\omega(\cdot, \cdot; \cdot; \hat{\xi}_t, \zeta_t)$.

More concretely, the variance-reduced estimator for each level-$k$ function is computed as:

$$h_{t+1}^{(k)} = g^{(k)}(h_{t+1}^{(k-1)}; \xi_{t+1}^{(k)}) + (1 - \alpha\eta^2)(h_t^{(k)} - g^{(k)}(h_t^{(k-1)}; \xi_{t+1}^{(k)})), \tag{7}$$

where $h_{t+1}^{(0)} = x_{t+1}$ when $k = 0$, and $\alpha > 0$ is a hyperparameter such that $\alpha\eta^2 \in (0, 1)$.

**Algorithm 1** Stochastic Smoothed Multi-Level Compositional Gradient Descent Ascent with Variance Reduced ( Smoothed -SMCGDA-VR)

---

**Input:** $\eta > 0$, $\alpha > 0$, $\rho_x > 0$, $\rho_y > 0$, $\gamma_x > 0$, $\gamma_y > 0$, $\gamma_z > 0$, $\rho_x\eta^2 < 1$, $\rho_y\eta^2 < 1$, $\alpha\eta^2 < 1$, $\gamma_z\eta < 1$.

Initialization: $h_0^{(0)} = x_0$, $h_0^{(k)} = g^{(k)}(h_0^{(k-1)}; \xi_0^{(k)})$, for $k \in \{1, \cdots, K\}$,
$p_0 = \nabla g^{(1)}(x_0; \xi_0^{(1)}) \cdots \nabla g^{(K)}(h_0^{(K-1)}; \xi_0^{(K)})\nabla_1 f(h_0^{(K)}, y_0; \zeta_0) + \omega(x_0 - z_0)$,
$q_0 = \nabla_2 f(h_0^{(K)}, y_0; \zeta_0)$, $u_0 = p_0$, $v_0 = q_0$.

1: **for** $t = 0, \cdots, T - 1$ **do**
2:    Update $x$ and $y$: $x_{t+1} = x_t - \gamma_x\eta p_t$,    $y_{t+1} = y_t + \gamma_y\eta q_t$,
3:    Update $z$: $z_{t+1} = z_t + \gamma_z\eta(x_{t+1} - z_t)$,
4:    $h_{t+1}^{(0)} = x_{t+1}$,
5:    **for** $k = 1, \cdots, K$ **do**
6:       Compute $k$-th inner-level function:
         $h_{t+1}^{(k)} = g^{(k)}(h_{t+1}^{(k-1)}; \xi_{t+1}^{(k)}) + (1 - \alpha\eta^2)(h_t^{(k)} - g^{(k)}(h_t^{(k-1)}; \xi_{t+1}^{(k)}))$
7:    **end for**
8:    Compute stochastic compositional gradient $u_{t+1}$ and $v_{t+1}$:
      $u_{t+1;t+1} = \nabla g^{(1)}(h_{t+1}^{(0)}; \xi_{t+1}^{(1)}) \cdots \nabla g^{(K-1)}(h_{t+1}^{(K-2)}; \xi_{t+1}^{(K-1)})\nabla g^{(K)}(h_{t+1}^{(K-1)}; \xi_{t+1}^{(K)}) \times \nabla_1 f(h_{t+1}^{(K)}, y_{t+1}; \zeta_{t+1}) + \omega(x_{t+1} - z_{t+1})$,
      $v_{t+1;t+1} = \nabla_2 f(h_{t+1}^{(K)}, y_{t+1}; \zeta_{t+1})$,
9:    Compute variance-reduced gradient $p_{t+1}$ and $q_{t+1}$:
      $p_{t+1} = u_{t+1;t+1} + (1 - \rho_x\eta^2)(p_t - u_{t;t+1})$,    $q_{t+1} = v_{t+1;t+1} + (1 - \rho_y\eta^2)(q_t - v_{t;t+1})$,
10: **end for**

---

For the outer-level update, we compute the stochastic gradient of the smoothed loss defined in Eq. (5), based on the variance-reduced estimator $\{h_{t+1}^{(k)}\}_{k=1}^K$ of the inner-level function, as presented in Step 8. Here, $u_{t+1;t+1}$ denotes the stochastic compositional gradient regarding primal variable, where the first index indicates the $t + 1$-th iteration of the variable, and the second reflects the sample indices $\hat{\xi}_{t+1} = \{\{\xi_{t+1}^{(k)}\}_{k=1}^K, \zeta_{t+1}\}$. Similarly, we compute the stochastic gradient with respect to the dual variable based on the variance-reduced estimator $h_{t+1}^{(K)}$ of the inner-level function. The algorithm then performs STORM-like updates on $p_{t+1}$ and $q_{t+1}$, as presented in Step 9, where $\rho_x > 0$ and $\rho_y > 0$ are two hyperparameters such that $\rho_x\eta^2 \in (0, 1)$ and $\rho_y\eta^2 \in (0, 1)$.

### 4.2 Theoretical Analysis

We derive the convergence rate of Algorithm 1 in the following theorem [2].

**Theorem 4.1.** *Given Assumptions 3.1-3.3, when $\rho_x > 0$, $\rho_y > 0$, $\alpha > 0$, $\omega = O(\ell)$, and the hyperparameter conditions in Eq. (94) are satisfied, Algorithm 1 achieves the following convergence upper bound:*

$$\frac{1}{T} \sum_{t=0}^{T-1} \left( \mathbb{E}[\|\nabla_x f(G(x_t), y_t)\|^2] + \kappa\mathbb{E}[\|\nabla_y f(G(x_t), y_t)\|^2] \right)$$

$$\leq O\left(\frac{\kappa\mathcal{P}_0}{\gamma_x\eta T}\right) + O\left(\frac{\kappa\sigma^2}{\rho_x\eta^2 TS}\right) + O\left(\frac{\kappa\sigma^2}{\rho_y\eta^2 TS}\right) + O\left(\frac{\kappa\sigma^2}{\alpha\eta^2 TS}\right)$$

$$+ O\left(\kappa\frac{\alpha^2\eta^2\sigma^2}{\rho_x}\right) + O\left(\kappa\rho_x\eta^2\sigma^2\right) + O\left(\kappa\frac{\alpha^2\eta^2\sigma^2}{\rho_y}\right) + O\left(\kappa\rho_y\eta^2\sigma^2\right) + O\left(\kappa\alpha\eta^2\sigma^2\right), \tag{8}$$

*where $\mathcal{P}_0 = f_\omega(G(x_0), y_0; z_0) - 2f_{\omega,d}(y_0; z_0) + 2g(z_0)$, with the definitions of the involved terms provided in Eq. (25).*

**Corollary 4.2.** *Given Assumptions 3.1-3.3, by setting $\gamma_x = O(1)$, $\gamma_y = O(1)$, $\gamma_z = O(1/\kappa)$, $\eta = O\left(\epsilon/\kappa^{1/2}\right)$, $\rho_x = O(1)$, $\rho_y = O(1)$, $\alpha = O(1)$, $S = O\left(\kappa^{1/2}/\epsilon\right)$, $T = O\left(\kappa^{3/2}/\epsilon^3\right)$,*

---

[2]Due to space limitations, the theorem with the full hyperparameter conditions is provided in the Appendix B.2.

*Algorithm 1 can achieve the $O(\epsilon, \epsilon/\sqrt{\kappa})$-stationary solution, where $\epsilon > 0$ denotes the solution accuracy, and $S$ is the batch size in the initial iteration.*

Note that our Theorem 4.1 provides the convergence rate in terms of the stationary point of *the original loss function $f(G(x_t), y_t)$*, rather than that of the smoothed loss function $f_\omega(G(x), y; z)$. Therefore, this result corresponds to the convergence rate for a nonconvex-PL loss function, rather than for a two-sided-PL loss function. As a result, the comparison of convergence rates with existing methods in Table 1 is fair and consistent with the comparison made in the context of classical smoothed minimax optimization in [30].

**Proof Sketch.** To establish the convergence rate of Algorithm 1, we propose a novel potential function as follows:

$$\mathcal{H}_t = \underbrace{f_\omega(G(x_t), y_t; z_t) - 2h_{\omega,d}(y_t; z_t) + 2h(z_t)}_{\mathcal{P}_t \triangleq \text{ Optimization Error: Lemmas B.3, B.4}} + \nu_a \underbrace{\mathbb{E}[\|p_t - \nabla_x f_\omega(H(x_t), y_t; z_t)\|^2]}_{\text{Gradient Error regarding } x: \text{ Lemma B.6}}$$

$$+ \nu_b \underbrace{\mathbb{E}[\|q_t - \nabla_y f_\omega(H(x_t), y_t; z_t)\|^2]}_{\text{Gradient Error regarding } y: \text{ Lemma B.7}} + \sum_{k=1}^{K} \lambda_k \underbrace{\mathbb{E}[\|h_t^{(k)} - g^{(k)}(h_t^{(k-1)})\|^2]}_{\text{Inner-level Estimation Error: Lemma B.5}} , \tag{9}$$

where the coefficient $\nu_a$, $\nu_b$ and $\{\lambda_k\}_{k=1}^K$ are positive, where the notations of $\nabla_x f_\omega(H(x_t), y_t; z_t)$ and $\nabla_y f_\omega(H(x_t), y_t; z_t)$ can be found in Eq. (23).

To analyze the descent of the potential function, we decompose and bound each term through a sequence of lemmas. First, we bound:

$$\mathcal{P}_t = f_\omega(G(x_t), y_t; z_t) - 2h_{\omega,d}(y_t; z_t) + 2h(z_t) , \tag{10}$$

which characterizes the optimization error introduced by the smoothed technique. Each component of $\mathcal{P}_t$ depends on $x$, $y$ and $z$, and the compositional gradient introduces additional bias:

- We first derive upper bounds for each component in $\mathcal{P}_t$.
- We then combine these bounds in Appendix B.2.1 to analyze and quantify their dependence, providing a clear characterization of how the three terms interact.

Second, three additional terms in Eq. (9) arise from the gradient errors regarding $x$ and $y$, and the inner-level estimation error in the multi-level compositional loss.

Third, the four terms in $\mathcal{H}_t$ are interdependent. We analyze these dependencies in Appendix B.2.2 and show that $\mathcal{H}_t$ satisfies a sufficient descent property, *i.e.*, $\mathcal{H}_{t+1} - \mathcal{H}_t$ can be bounded under suitable hyperparameter conditions, ensuring convergence to an $(\epsilon, \epsilon/\sqrt{\kappa})$-stationary point. The complete proof is provided in Appendix B.

## 5 Algorithm 2: Stagewise-SMCGDA-VR

### 5.1 Algorithmic Design

---

**Algorithm 2** Stagewise-SMCGDA-VR

---

**Input:** $\rho_x > 0$, $\rho_y > 0$, $\alpha > 0$, $\eta_{x,r} > 0$, $\eta_{y,r} > 0$.
1: **for** Stage $r = 0, \cdots, R - 1$ **do**
2:     $x_{r,0} = \tilde{x}_r$, $y_{r,0} = \tilde{y}_r$, $h_{r,0}^{(k)} = \tilde{h}_r^{(k)}$ for $k \in \{0, \cdots, K-1\}$,
       $p_{r,0} = \tilde{p}_r$, $q_{r,0} = \tilde{q}_r$.
3:     **for** $t = 0, \cdots, T_r - 1$, **do**
4:         Perform one iteration $t$ of SMCGDA-VR update
5:         Randomly select $(\tilde{x}_{r+1}, \tilde{y}_{r+1}, \tilde{h}_{r+1}^{(k)}, \tilde{p}_{r+1}, \tilde{q}_{r+1})$ from $\{(x_{r,t}, y_{r,t}, h_{r,t}^{(k)}, p_{r,t}, q_{r,t})\}_{t=0}^{T_r-1}$.
6:     **end for**
7: **end for**

---

However, to facilitate a fair comparison between the convergence rate of Algorithm 1 and existing stochastic two-level compositional minimax methods, which establish the rate in terms of $\epsilon$-stationary point instead of $(\epsilon_1, \epsilon_2)$-stationary point, it is necessary to convert the $(\epsilon, \epsilon/\sqrt{\kappa})$-stationary solution into an $\epsilon$-stationary solution. As discussed in Section 3.3, making this translation is challenging for

multi-level compositional minimax optimization problems. Specifically, in the classical minimax setting, [30] showed that the standard stochastic gradient descent ascent (SGDA) algorithm is sufficient for the translation by solving a strongly-convex–strongly-concave problem, since its convergence rate is only $\tilde{O}(1/\epsilon^2)$, which is dominated by that of the smoothed algorithm. However, this approach does not work for the multi-level compositional minimax optimization problem. Specifically, even for the multi-level compositional *minimization* optimization problem, the classical stochastic compositional gradient descent algorithm can only achieve a convergence rate with an exponential dependence on the number of levels for strongly convex loss functions, as shown in [31].

The aforementioned challenge motivates the development of a new algorithm to handle the translation from an $(\epsilon, \epsilon/\sqrt{\kappa})$-stationary solution into an $\epsilon$-stationary solution. To this end, we aim to develop a new algorithm to solve the multi-level compositional minimax optimization problem that satisfies the two-sided PL condition. Specifically, assume $\tilde{z}$ is the output of Algorithm 1, then we complete the translation by solving the following problem.

$$\min_{x\in\mathbb{R}^{d_x}} \max_{y\in\mathbb{R}^{d_y}} \hat{f}(G(x),y) := f(G(x),y) + \boxed{\frac{\omega}{2}\|x-\tilde{z}\|^2} . \tag{11}$$

Note that $\tilde{z}$ is the output $x$ from Algorithm 1 and it is fixed when solving this problem. Moreover, since $\omega$ is selected such that $\hat{f}(G(x),y)$ is strongly convex with respect to $x$, $\hat{f}(G(x),y)$ naturally satisfies the two-sided PL condition. Then, our next goal is to develop an efficient algorithm to solve Eq. (11) such that its iteration complexity is better than that of Algorithm 1, i.e., **the translation does not hurt the overall convergence rate**.

To this end, we propose a novel stage-wise algorithm, named Stagewise-SMCGDA-VR, as shown in Algorithm 2 (*Note that a more general algorithm is presented in Algorithm 3 for the multi-level compositional minimax optimization problem satisfying the two-sided PL condition. This algorithm may be of independent interest, beyond its use for the translation phase.*). The overall optimization is divided into $R$ stages, and in each stage, we run the SMCGDA-VR algorithm without updating $z$ (i.e., removing the component highlighted in blue) and replacing $z$ with $\tilde{z}$ in Step 8. At the end of each stage $r$, the algorithm randomly selects a tuple from the set $\{(x_{r,t}, y_{r,t}, h_{r,t}^{(k)}, p_{r,t}, q_{r,t})\}_{t=0}^{T_r-1}$, where $k \in \{1, \ldots, K\}$, to be used as the initialization for the next stage $r + 1$. A complete description of the algorithm is given in the Appendix C.

## 5.2 Theoretical Analysis

We establish the convergence rate of Algorithm 2 in the following theorem. More general results for the extended Algorithm 3, which may be of independent interest, are presented in Theorems C.1-C.2.

**Theorem 5.1.** *Given Assumption 3.1-3.4, by setting $c_0 = \frac{25L_f^2}{\mu^2}$, $\rho_x = 6400c_0L_\beta^2$, $\rho_y = 640L_\beta^2$, $\alpha = 640c_0L_\beta^2$, $\eta_{y,0} = \frac{1}{20L_\beta}$, $T_0 = \max\{225, \frac{16\mathcal{V}_0}{L_\beta\sigma^2}\}$, and for $r \geq 1$, $\eta_{x,r} = O(\mu^2/(\sqrt{2^{r-1}}L_\beta))$, $\eta_{y,r} = O(1/(\sqrt{2^{r-1}}L_\beta))$, $T_r = O(c_0/(\mu \times 2^{r-1}))$, after running Algorithm 2 for the total number of iterations (not stages) $O(1/\epsilon^2)$, we can get $\mathbb{E}[\|\nabla\Phi(\tilde{x}_R)\|^2] \leq \epsilon^2$.*

**Remark 5.2.** *From Theorem 5.1, it can be observed that the iteration complexity $O(1/\epsilon^2)$ of the translation phase is much smaller than that of Algorithm 1. Therefore, the translation does not hurt the overall convergence rate.*

**Remark 5.3.** *Since the overall iteration complexity is determined by Algorithm 1, we can conclude that our algorithm achieves an iteration complexity of $T = O(\kappa^{3/2}/\epsilon^3)$, improving upon the $O(\kappa^4/\epsilon^4)$ complexity of the two-level compositional minimax problem in [17, 9] by offering better dependence on both $\kappa$ and $\epsilon$ and the $O(\kappa^3/\epsilon^3)$ complexity in [26] by a better dependence on $\kappa$. To the best of our knowledge, this is the first algorithm to achieve an $O(\kappa^{3/2})$ dependence for (multi-level compositional) minimax problems under the nonconvex-PL setting.*

**Proof Sketch.** To prove Theorem 5.1, we use an induction approach to handle the stage-wise structure of Algorithms 2. We introduce two metrics to facilitate convergence analysis:

$$\mathbb{E}[\mathcal{V}_r] = \underbrace{\mathbb{E}[\Phi(\tilde{x}_r) - \Phi(x^*)]}_{Lemma\ C.3} + \frac{c_0\eta_{x,r}}{\eta_{y,r}}\underbrace{\mathbb{E}[\Phi(\tilde{x}_r) - f(G(\tilde{x}_r), \tilde{y}_r)]}_{Lemma\ C.5},$$

$$\mathbb{E}[\mathcal{U}_r] = \mathbb{E}[\|\nabla_x f(H(\tilde{x}_r), \tilde{y}_r) - \tilde{p}_r\|^2] + \mathbb{E}[\|\nabla_y f(H(\tilde{x}_r), \tilde{y}_r) - \tilde{q}_r\|^2]$$

$$+ 56 \sum_{k=1}^{K} \lambda'_k \mathbb{E}[\|g^{(k)}(\tilde{h}_r^{(k-1)}) - \tilde{h}_r^{(k)}\|^2] \,, \tag{12}$$

where $\mathcal{V}_t$ denotes the optimization error, and $\mathcal{U}_t$ is similar to the last three terms of Eq. (9), $c_0$ is a positive constant such that $\frac{c_0 \eta_{x,t}}{\eta_{y,t}} = \frac{1}{10}$. Importantly, following Lemma C.6 and C.7, we establish how $\mathcal{V}_t$ and $\mathcal{U}_t$ affect each other across stages and derive the following inequalities in Appendix C.3:

$$\mathbb{E}[\mathcal{U}_{r+1}] \leq \frac{20c_0}{\eta_{y,r} T_r} \mathbb{E}[\mathcal{V}_r] + \frac{320c_0}{\rho_y \eta_{y,r}^2 T_r} \mathbb{E}[\mathcal{U}_r] + 338 c_0 \rho_y \eta_{y,t}^2 L_\beta^2 \sigma^2 \,,$$

$$\mathbb{E}[\mathcal{V}_{r+1}] \leq \frac{1}{\mu} \Big( \frac{20c_0}{\eta_{y,r} T_r} \mathbb{E}[\mathcal{V}_r] + \frac{320c_0}{\rho_y \eta_{y,r}^2 T_r} \mathbb{E}[\mathcal{U}_r] + 338 c_0 \rho_y \eta_{y,t}^2 L_\beta^2 \sigma^2 \Big) \,. \tag{13}$$

These bounds differ only by a factor of $1/\mu$. Using induction, at the $r$-th stage, we assume

$$\mathbb{E}[\mathcal{V}_r] \leq \epsilon_r \,, \quad \mathbb{E}[\mathcal{U}_r] \leq \mu \epsilon_r \,, \tag{14}$$

where $\epsilon_r > 0$ is a constant. Finally, by selecting appropriate hyperparameters, we prove that

$$\mathbb{E}[\mathcal{V}_{r+1}] \leq \epsilon_{r+1} \triangleq \frac{\epsilon_r}{2} \,, \quad \mathbb{E}[\mathcal{U}_{r+1}] \leq \mu \epsilon_{r+1} \,. \tag{15}$$

As such, we establish the desired convergence rate. The complete proof is provided in Appendix C.

## 6 Experiment

### 6.1 Deep AUC Maximization

In the deep AUC maximization problem, applying $K$-step gradient descent to minimize the cross-entropy loss function results in a $K$-level inner function $G(\cdot)$ in Eq. (1), with a detailed discussion provided in Appendix A. We compare our smoothed method with three baselines: SCGDA [17], SCGDAM [33], and NSTORM [26] across three datasets: CATvsDOG, CIFAR10 and STL10. Imbalanced binary datasets are generated following the approach described in [33], with an imbalance ratio of 0.05. ResNet20 is employed as the model. For all algorithms, we set both the learning rate and the momentum or variance reduction coefficient to 0.1. In our proposed method, we employ smoothed techniques during the first 90 epochs, followed by stage-wise updates for the remaining 10 epochs.

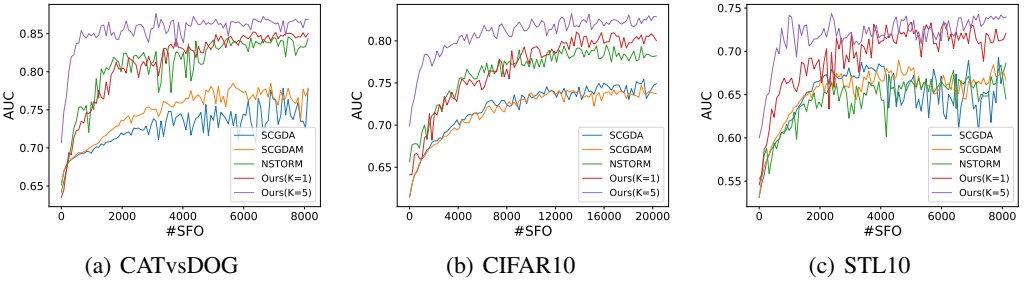

|                |                |                |
| -------------- | -------------- | -------------- |
| (a) CATvsDOG   | (b) CIFAR10    | (c) STL10      |

Figure 1: The test AUC score versus the number of stochastic first-order gradient evaluations.

We conduct experiments using our smoothed method for both $K = 1$ and $K = 5$, with results presented in Figure 1. Notably, NSTORM applies STORM-like updates to the two-level($K = 1$) compositional minimax problem without smoothed techniques. Our results show that the smoothed approach consistently outperforms all baselines. Moreover, as the number of levels increases, the smoothed method does not degrade the performance, demonstrating its robustness to increased compositional levels. This improvement is observed consistently across all datasets, highlighting the effectiveness of incorporating deeper compositional structures. Additional experiments with varying $K$ are provided in the Appendix A.

### 6.2 Multi-Instance Learning

Following [39], multi-instance learning can be reformulated as a multi-level compositional minimax problem as shown in Eq. 1, with details provided in Appendix A. For multi-instance learning tasks, our proposed approach utilizes two types of stochastic pooling operations: log-sum-exp (smx) pooling and attention-based (att) pooling. We compare the performance of our smoothed methods against six baseline methods: MIDAM(smx) and MIDAM(att) [40], both utilizing stochastic pooling

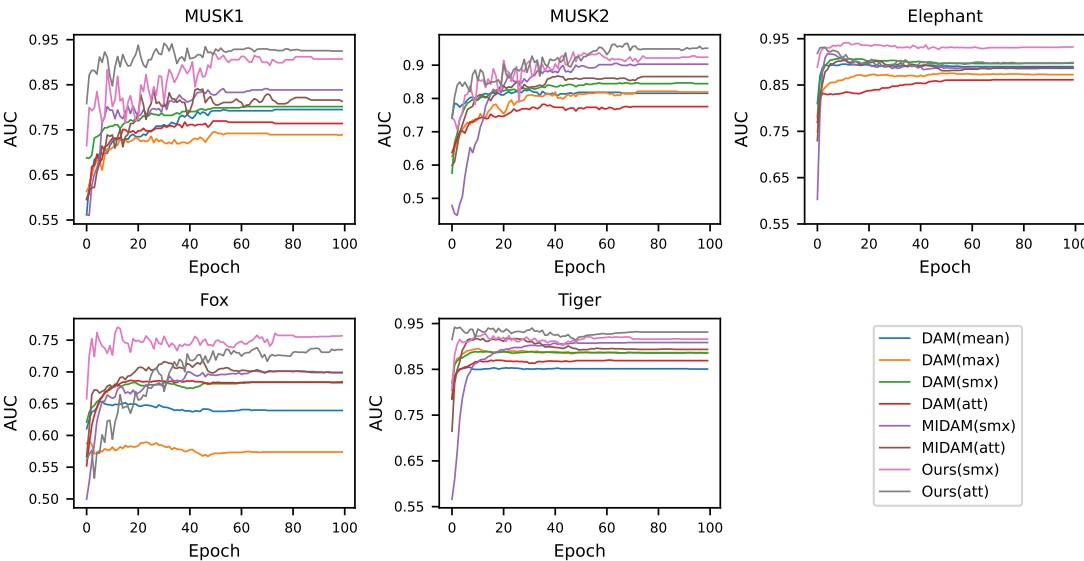

Figure 2: The test AUC score versus the number of epochs for Tabular Datasets.

operations; DAM(mean), DAM(max), DAM(smx), and DAM(att), all of which update the AUC loss with traditional PESG optimizer [34].

We conduct experiments on five commonly used tabular benchmark datasets [10, 1] for MIL tasks – MUSK1, MUSK2, Fox, Tiger, and Elephant – as well as one histopathological image dataset, namely Breast Cancer. For the tabular datasets, we use a two-layer feed-forward neural network with tanh activation and a sigmoid output for AUC loss normalization. For the Breast Cancer dataset, each image is divided into $32 \times 32$ patches and treated as a bag of 672 local patches to enable efficient multi-instance processing, using ResNet20 as the model. All datasets are randomly split into training and testing sets with a 0.9/0.1 ratio.

For the tabular datasets, we perform 5-fold cross-validation, repeating each run with three random seeds. For the image dataset, we use two random seeds. The learning rate for the primal variables is tuned within the set {1e-1, 1e-2, 1e-3}, while the learning rate for the dual variables is fixed at 1. We vary the value of $K$ from 1 to 5 and ultimately fix it at 3 to achieve more stable performance. We present the experimental results on the tabular datasets in Figure 2, and on

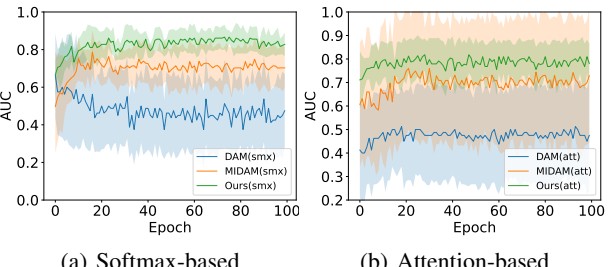

(a) Softmax-based     (b) Attention-based

Figure 3: The test AUC score versus the number of epochs for Breast Cancer Dataset.

the image dataset in Figure 3. For the tabular datasets, to ensure clearer visualizations, we omit error bars in the plots and instead report both the mean and standard deviation of the results in Table 2, as shown in Appendix A. For the image dataset, we focus our comparison on the softmax-based and attention-based methods. In both experimental settings, our proposed algorithms consistently outperform all baseline methods, demonstrating superior optimization behavior and generalization performance across a range of tasks and datasets.

## 7 Conclusion

In this work, we addressed the challenging problem of stochastic multi-level compositional mini-max optimization by proposing a smoothed variance-reduced algorithm. Our theoretical analysis demonstrates that the proposed smoothed method achieves a convergence rate of $O\left(\kappa^{3/2}/\epsilon^3\right)$ to an $(\epsilon, \epsilon/\sqrt{\kappa})$-stationary point. Furthermore, to bridge the gap between different stationarity measures, we developed a stage-wise algorithm under the two-sided PL condition, enabling a translation to an $\epsilon$-stationary point. Extensive experiments on deep AUC maximization and multi-instance learning tasks validate the superior performance of our approach.

## Acknowledgements

We thank anonymous reviewers for constructive comments. X. Zhang and H. Gao were partially supported by U.S. NSF CAREER 2339545, NSF IIS 2416607, NSF CNS 2107014.

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

# Contents

# A Applications and Experiments

## A.1 Deep AUC Maximization

AUC(Area under the ROC curve) is widely used to evaluate the classifiers for binary classification with imbalanced data. [33] reformulated the AUC maximization problem as the following two-level compositional minimax problem:

$$\min_{\tilde{w},a,b} \max_{\alpha} \mathcal{L}_{AUC}(\tilde{w}, a, b, \alpha; x, y)$$
$$s.t. \quad \tilde{w} = w - \tilde{\eta}\nabla_w \mathcal{L}_{CE}(w; x, y) \,, \tag{16}$$

where $w \in \mathbb{R}^d$ are model parameters while $(a, b, \alpha)$ are parameters for AUC loss, $(x, y)$ represents feature and label of a sample.

Here, $\mathcal{L}_{CE}$ indicates the standard cross-entropy loss function, $w - \tilde{\eta}\nabla_w \mathcal{L}_{CE}$ denotes using the gradient descent approach on cross-entropy loss to update the model parameters, where $\tilde{\eta} > 0$ is the learning rate. Then, the obtained model parameter $\tilde{w}$ can be optimized through the AUC loss. The following serves as a generic representation of Eq. (16) as a *two-level* compositional minimax optimization problem:

$$\min_{x \in \mathbb{R}^{d_x}} \max_{y \in \mathbb{R}^{d_y}} f(g(x), y) \,, \tag{17}$$

where $g$ denotes the inner-level function with one-step gradient descent and $f$ denotes the outer-level function. Inspired

Figure 4: Different $K$ on CIFAR10.

by the achievements in addressing the multi-level compositional minimization problem, we extend the one-step gradient descent for the inner-level function to a multi-step update. In detail, for $k \in \{1, \cdots, K\}$, the $k$-th inner-level function is defined as:

$$g^{(k)}(\cdot) \triangleq \begin{cases} \mathbb{E}[g^{(1)}(x; \xi^{(1)})] = \mathbb{E}[x - \tilde{\eta}\Delta(x; \xi^{(1)})], & k = 1, \\ \mathbb{E}[g^{(k)}(\tilde{g}; \xi^{(k)})] = \mathbb{E}[\tilde{g} - \tilde{\eta}\Delta(\tilde{g}; \xi^{(k)})], & k \neq 1, \end{cases} \tag{18}$$

where $\tilde{g}$ refers to $g^{(k-1)}(\cdot)$ when $k \in \{2, \cdots, K\}$, $\xi^{(k)}$ represents the data distribution for the $k$-th level function. The learning rate for the inner-level functions is denoted by $\tilde{\eta}$. Consequently, Eq. (17) can be reformulated as a *multi-level* compositional minimax optimization problem exactly as the Eq. (1).

## A.2 Multi-Instance Learning

Multi-instance learning [10] is designed for tasks with training data structured into bags containing many instances, with only bag-level labels known. The symmetric function, also known as the pooling operation, is a critical component of multi-instance learning. Diverse pooling strategies have been investigated, including mean pooling, max pooling, and softmax pooling [28] and attention-based pooling [21]. Then, to address memory concerns, [40] provided a class of variance-reduced stochastic pooling approaches by reformulating the AUC loss function with the pooled prediction as a *three-level compositional minimax function* as follows:

$$\min_{w,a,b} \max_{\alpha} \mathcal{F}(w, a, b, \alpha) := \mathbb{E}_{i \in \mathcal{D}_+}[(h(w; \mathcal{X}_i) - a)^2] + \mathbb{E}_{i \in \mathcal{D}_-}[(h(w; \mathcal{X}_i) - b)^2]$$

$$+ \alpha(c + \mathbb{E}_{i \in \mathcal{D}_-}[h(w; \mathcal{X}_i)] - \mathbb{E}_{i \in \mathcal{D}_+}[h(w; \mathcal{X}_i)]) - \frac{\alpha^2}{2} \,, \tag{19}$$

where $\mathcal{X}_i = \{x_i^1, \cdots, x_i^{n_i}\}$ denotes a bag of data instances, $\mathcal{D}_+$ represents only containing positive bags with label $y_i = 1$, $\mathcal{D}_-$ represents only containing negative bags with label $y_i = 0$. The pooled prediction $h(w; \mathcal{X}_i) = f_2(f_1(w; \mathcal{X}_i))$ denotes the predicted score of the bag $i$ over all its instance, which is a two-level compositional function. For example, for the log-sum-exp(smx) pooling, we have:

$$f_1(w; \mathcal{X}_i) = \frac{1}{|\mathcal{X}_i|} \sum_{x_i^j \in \mathcal{X}_i} \exp(\phi(w; x_i^j))/\tau), \quad f_2(s_i) = \tau \log(s_i). \tag{20}$$

For the attention-based (att) pooling, we have:

$$f_1(w; \mathcal{X}_i) = \begin{bmatrix} \frac{1}{|\mathcal{X}_i|} \exp(g(w; x_i^j)) w_c^T e(w_e; x_i^j) \\ \frac{1}{|\mathcal{X}_i|} \sum_{x_i^j \in \mathcal{X}_i} \exp(g(w; x_i^j)) \end{bmatrix}, \quad f_2(s_i) = \sigma\left(\frac{s_{i1}}{s_{i2}}\right). \tag{21}$$

Similarly, the three-level compositional minimax problem in Eq. (19) can be reformulated as a stochastic multi-level compositional minimax problem by integrating it with cross-entropy loss minimization, as in Eq. (16), after computing the predicted score $h(w; \mathcal{X}_i)$. In particular, applying $K$ inner gradient steps to optimize the cross-entropy loss results in a $K$-level inner function. Consequently, Eq.(19) can be expressed in the unified form of Eq. (1), which corresponds to a stochastic $(K+3)$-level compositional minimax problem.

### A.3 More Experimental Results

Here, we provide additional empirical results. Specifically, for the deep AUC maximization task, we perform experiments to evaluate the impact of the number of levels $K$ on performance. As shown in Figure 4, increasing the number of inner levels leads to further improvements in testing performance. For the multi-instance learning task, we report both the mean and standard deviation of the results on tabular datasets in Table 2.

Table 2: The test AUC score of different methods on all Tabular Datasets.

| Methods | MUSK1 | MUSK2 | Fox | Tiger | Elephant |
|---|---|---|---|---|---|
| Ours(att) | **0.942(0.039)** | **0.965(0.029)** | 0.738(0.018) | **0.942(0.017)** | 0.931(0.034) |
| Ours(smx) | 0.921(0.047) | 0.939(0.025) | **0.770(0.034)** | 0.928(0.026) | **0.942(0.026)** |
| MIDAM(att) | 0.841(0.142) | 0.868(0.087) | 0.718(0.078) | 0.918(0.030) | 0.919(0.029) |
| MIDAM(smx) | 0.841(0.142) | 0.905(0.117) | 0.702(0.056) | 0.909(0.031) | 0.903(0.039) |
| DAM(att) | 0.770(0.143) | 0.782(0.075) | 0.686(0.050) | 0.870(0.027) | 0.861(0.022) |
| DAM(smx) | 0.802(0.175) | 0.847(0.116) | 0.684(0.049) | 0.889(0.014) | 0.908(0.025) |
| DAM(max) | 0.745(0.112) | 0.822(0.123) | 0.591(0.082) | 0.895(0.047) | 0.875(0.028) |
| DAM(mean) | 0.795(0.138) | 0.826(0.072) | 0.653(0.103) | 0.855(0.021) | 0.895(0.020) |

## B  Appendix: Smoothed-SMCGDA-VR

To begin with, we introduce the following terminology to simplify the complex expressions, which will be useful in the subsequent analysis:

$$\nabla_x f(G(x_t), y_t) = \nabla g^{(1)}(x_t) \nabla g^{(2)}(G^{(1)}(x_t)) \cdots \nabla g^{(K)}(G^{(K-1)}(x_t)) \nabla_1 f(G^{(K)}(x_t), y_t),$$

$$\nabla_y f(G(x_t), y_t) = \nabla_2 f(G^{(K)}(x_t), y_t),$$

$$\nabla_x f(H(x_t), y_t) = \nabla g^{(1)}(x_t) \nabla g^{(2)}(h_t^{(1)}) \cdots \nabla g^{(K-1)}(h_t^{(K-2)}) \nabla g^{(K)}(h_t^{(K-1)}) \nabla_1 f(h_t^{(K)}, y_t),$$

$$\nabla_y f(H(x_t), y_t) = \nabla_2 f(h_t^{(K)}, y_t). \tag{22}$$

Therefore, for the smoothed loss, we have

$$\nabla_x f_\omega(H(x_t), y_t; z_t) = \nabla_x f(H(x_t), y_t) + \omega(x_t - z_t),$$
$$\nabla_y f_\omega(H(x_t), y_t; z_t) = \nabla_y f(H(x_t), y_t). \tag{23}$$

Moreover, we introduce $C_p^2$ as follows:

$$C_p^2 = \max\left\{ (K+1) C_g^{2(K-1)} (K C_f^2 + C_g^2), (K+1)\ell^2 \right\}. \tag{24}$$

Following [30], we introduce the following auxiliary functions for convergence analysis:

$$h_{\omega,d}(y; z) = \min_{x \in \mathbb{R}^{d_x}} f_\omega(G(x), y; z), \quad \text{dual function}$$

$$h_{\omega,p}(x; z) = \max_{y \in \mathbb{R}^{d_y}} f_\omega(G(x), y; z), \quad \text{primal function}$$

$$h(z) = \min_{x \in \mathbb{R}^{d_x}} \max_{y \in \mathbb{R}^{d_y}} f_\omega(G(x), y; z) ,$$

$$x^*(y, z) = \arg \min_{x \in \mathbb{R}^{d_x}} f_\omega(G(x), y; z) ,$$

$$x^*(z) = \arg \min_{x \in \mathbb{R}^{d_x}} h_{\omega, p}(x; z) ,$$

$$y^*(z) = \arg \max_{y \in \mathbb{R}^{d_y}} h_{\omega, d}(y; z) . \tag{25}$$

**Proof Structure.** Our proof consists of two key components. The first component, including Lemma B.3 and Lemma B.4, addresses the smoothing technique. The second component, comprising Lemma B.5, Lemma B.6, and Lemma B.7, deals with the multi-level compositional structure. In Section B.2, we complete the proof by carefully combining these two components while addressing their interdependence.

## B.1 Useful Lemmas

**Lemma B.1.** *Given Assumptions 3.1-3.3, we can know*

1. *$G^{(k)}(x)$ is $C_g^k$-Lipschitz continuous for $k \in \{1, \cdots, K-1\}$ and $G(x)$ is $C_G$-Lipschitz continuous where $C_G = C_g^K$;*

2. *$\nabla G(x)$ is $L_G$-Lipschitz continuous where $L_G = \sum_{j=0}^{K-1} L_g C_g^{K-1+j}$;*

3. *$\nabla_x f(G(x), y)$ is $\hat{L}$-Lipschitz continuous where $\hat{L} = C_G^2 L_f + C_f L_G$;*

4. *$\Phi(x) \triangleq \max_{y \in \mathbb{R}^{d_y}} f(G(x), y)$, $\Phi(x)$ is $L_\Phi$-Lipschitz continuous where $L_\Phi = \frac{2C_G^2 L_f^2}{\mu} + C_f L_G$.*

*Proof.* The first three properties follow from Lemma B.1. in [37]. The last property is based on Lemma A.3. in [30] and can be established by showing that $\|\Phi(x_2) - \Phi(x_1)\| \leq (\frac{2C_G^2 L_f^2}{\mu} + C_f L_G)\|x_2 - x_1\|$. $\square$

**Lemma B.2.** *[30] Given Assumptions 3.1-3.3, the following inequality holds:*

$$\|x^*(y_1, z) - x^*(y_2, z)\| \leq C_{x_{yz}^1} \|y_1 - y_2\| ,$$

$$\|x^*(y, z_1) - x^*(y, z_2)\| \leq C_{x_{yz}^2} \|z_1 - z_2\| ,$$

$$\|x^*(z_1) - x^*(z_2)\| \leq C_{x_z} \|z_1 - z_2\| , \tag{26}$$

*where $C_{x_{yz}^1} = \frac{\omega + \ell}{\omega - \ell}$, $C_{x_{yz}^2} = \frac{\omega}{\omega - \ell}$, and $C_{x_z} = \frac{\omega}{\omega - \ell}$ and $\omega > \ell$.*

**Lemma B.3.** *Given Assumptions 3.1-3.3, and $\gamma_z \eta \leq 1$, the following inequality holds:*

1. *The smoothed function $f_\omega(G(x_t), y_t; z_t)$ satisfies:*

$$f_\omega(G(x_{t+1}), y_{t+1}; z_t) - f_\omega(G(x_{t+1}), y_{t+1}; z_{t+1}) \geq \frac{\omega}{2\gamma_z \eta} \|z_{t+1} - z_t\|^2 . \tag{27}$$

2. *The dual function $\mathbb{E}[h_{\omega, d}(y_t; z_t)]$ satisfies:*

$$\mathbb{E}[h_{\omega, d}(y_{t+1}; z_{t+1})] \geq \mathbb{E}[h_{\omega, d}(y_t; z_t)] + \gamma_y \eta \mathbb{E}[\langle \nabla_y f_\omega(x^*(y_t, z_t), y_t; z_t), q_t \rangle]$$

$$- \frac{\gamma_y^2 \eta^2 L_{\omega, d}}{2} \mathbb{E}[\|q_t\|^2] + \frac{\omega}{2} \langle z_{t+1} - z_t, z_{t+1} + z_t - 2x^*(y_{t+1}; z_{t+1}) \rangle . \tag{28}$$

3. *The function $h(z_t)$ satisfies:*

$$h(z_{t+1}) - h(z_t) \leq \frac{\omega}{2} \langle z_{t+1} - z_t, z_{t+1} + z_t - 2x^*(y^*(z_{t+1}); z_t) \rangle . \tag{29}$$

*Proof. (1).* From the update rule $z_{t+1} = z_t + \gamma_z \eta(x_{t+1} - z_t)$, we obtain

$$f_\omega(G(x_{t+1}), y_{t+1}; z_t) - f_\omega(G(x_{t+1}), y_{t+1}; z_{t+1})$$

$$= \frac{\omega}{2}(\|x_{t+1} - z_t\|^2 - \|x_{t+1} - z_{t+1}\|^2)$$

$$= \frac{\omega}{2}(\frac{1}{\gamma_z^2 \eta^2}\|z_{t+1} - z_t\|^2 - \|(1 - \gamma_z \eta)(x_{t+1} - z_t)\|^2)$$

$$= \frac{\omega}{2}(\frac{1}{\gamma_z^2 \eta^2}\|z_{t+1} - z_t\|^2 - \frac{(1 - \gamma_z \eta)^2}{\gamma_z^2 \eta^2}\|z_{t+1} - z_t\|^2)$$

$$= \frac{\omega}{2}\frac{1 - 1 + 2\gamma_z \eta - \gamma_z^2 \eta^2}{\gamma_z^2 \eta^2}\|z_{t+1} - z_t\|^2$$

$$= \frac{\omega}{2}\frac{2 - \gamma_z \eta}{\gamma_z \eta}\|z_{t+1} - z_t\|^2$$

$$\geq \frac{\omega}{2\gamma_z \eta}\|z_{t+1} - z_t\|^2 , \tag{30}$$

where the last step holds uses the fact that $\gamma_z \eta \leq 1$. $\qquad\square$

*Proof. (2).* Since the dual function $h_{\omega,d}(y_t; z_t)$ is $L_{\omega,d}$-smooth, it satisfies that

$$\mathbb{E}[h_{\omega,d}(y_{t+1}; z_t)] \geq \mathbb{E}[h_{\omega,d}(y_t; z_t)] + \mathbb{E}[\langle \nabla_y h_{\omega,d}(y_t; z_t), y_{t+1} - y_t\rangle] - \frac{L_{\omega,d}}{2}\mathbb{E}[\|y_{t+1} - y_t\|^2]$$

$$= \mathbb{E}[h_{\omega,d}(y_t; z_t)] + \gamma_y \eta \mathbb{E}[\langle \nabla_y f_\omega(x^*(y_t, z_t), y_t; z_t), q_t\rangle] - \frac{\gamma_y^2 \eta^2 L_{\omega,d}}{2}\mathbb{E}[\|q_t\|^2] . \tag{31}$$

On the other hand, we have

$$h_{\omega,d}(y_{t+1}; z_{t+1}) - h_{\omega,d}(y_{t+1}; z_t)$$

$$= f_\omega(x^*(y_{t+1}; z_{t+1}), y_{t+1}; z_{t+1}) - f_\omega(x^*(y_{t+1}; z_{t+1}), y_{t+1}; z_t)$$

$$\geq f_\omega(x^*(y_{t+1}; z_{t+1}), y_{t+1}; z_{t+1}) - f_\omega(x^*(y_{t+1}; z_t), y_{t+1}; z_t)$$

$$= \frac{\omega}{2}(\|z_{t+1} - x^*(y_{t+1}; z_{t+1})\|^2 - \|z_t - x^*(y_{t+1}; z_{t+1})\|^2)$$

$$= \frac{\omega}{2}\langle z_{t+1} - z_t, z_{t+1} + z_t - 2x^*(y_{t+1}; z_{t+1})\rangle , \tag{32}$$

where we use the fact $\langle a - b, a + b\rangle = \|a\|^2 - \|b\|^2$ in the second-to-last step.

By combining the above two inequalities, the proof is complete. $\qquad\square$

*Proof. (3).* From the definition of $h(z_t)$, we obtain

$$h(z_{t+1}) - h(z_t)$$

$$= h_{\omega,d}(y^*(z_{t+1}); z_{t+1}) - h_{\omega,d}(y^*(z_t); z_t)$$

$$\leq h_{\omega,d}(y^*(z_{t+1}); z_{t+1}) - h_{\omega,d}(y^*(z_{t+1}); z_t)$$

$$= f_\omega(x^*(y^*(z_{t+1}); z_{t+1}), y^*(z_{t+1}); z_{t+1}) - f_\omega(x^*(y^*(z_{t+1}); z_t), y^*(z_{t+1}); z_t)$$

$$\leq f_\omega(x^*(y^*(z_{t+1}); z_t), y^*(z_{t+1}); z_{t+1}) - f_\omega(x^*(y^*(z_{t+1}); z_t), y^*(z_{t+1}); z_t)$$

$$= \frac{\omega}{2}(\|z_{t+1} - x^*(y^*(z_{t+1}); z_t)\|^2 - \|z_t - x^*(y^*(z_{t+1}); z_t)\|^2)$$

$$= \frac{\omega}{2}\langle z_{t+1} - z_t, z_{t+1} + z_t - 2x^*(y^*(z_{t+1}); z_t)\rangle , \tag{33}$$

where we use the fact $\langle a - b, a + b\rangle = \|a\|^2 - \|b\|^2$ in the last step. $\qquad\square$

**Lemma B.4.** *Given Assumptions 3.1-3.3, when $\eta \leq \frac{1}{2\gamma_x(\omega+\ell)}$, and $\gamma_z \eta \leq 1$, the following inequalities hold:*

$$\mathbb{E}[f_\omega(G(x_{t+1}), y_{t+1}; z_{t+1})] \leq \mathbb{E}[f_\omega(G(x_t), y_t; z_t)] - \frac{\gamma_x \eta}{2}\mathbb{E}[\|\nabla_x f_\omega(G(x_t), y_t; z_t)\|^2]$$

$$+ \frac{\gamma_y \eta}{2}\mathbb{E}[\|\nabla_y f_\omega(G(x_t), y_t; z_t)\|^2] - \frac{\omega}{2\gamma_z \eta}\mathbb{E}[\|z_{t+1} - z_t\|^2] + \gamma_x \eta \mathbb{E}[\|\nabla_x f_\omega(G(x_t), y_t; z_t) - p_t\|^2]$$

$$+ \gamma_x \eta K \sum_{k=1}^{K} A_k \mathbb{E}[\|g^{(k)}(h_t^{(k-1)}) - h_t^{(k)}\|^2] + \left(4\gamma_y \eta \gamma_x^2 \eta^2 \ell^2 - \frac{\gamma_x \eta}{4}\right) \mathbb{E}[\|p_t\|^2]$$

$$+ \left(\frac{3\gamma_y \eta}{4} + \frac{\omega + \ell}{2}\gamma_y^2 \eta^2\right) \mathbb{E}[\|q_t\|^2]. \tag{34}$$

*Proof.* First, from Lemma B.1, it follows that the smoothed loss function $f_\omega(G(x_t), y_t; z_t)$ is $(\omega + \ell)$-smooth with respect to $x$. Therefore, we have

$$\mathbb{E}[f_\omega(G(x_{t+1}), y_t; z_t)]$$

$$\leq \mathbb{E}[f_\omega(G(x_t), y_t; z_t)] + \mathbb{E}[\langle \nabla_x f_\omega(G(x_t), y_t; z_t), x_{t+1} - x_t \rangle] + \frac{\omega + \ell}{2} \mathbb{E}[\|x_{t+1} - x_t\|^2]$$

$$= \mathbb{E}[f_\omega(G(x_t), y_t; z_t)] - \gamma_x \eta \mathbb{E}[\langle \nabla_x f_\omega(G(x_t), y_t; z_t), p_t \rangle] + \frac{\omega + \ell}{2}\gamma_x^2 \eta^2 \mathbb{E}[\|p_t\|^2]$$

$$= \mathbb{E}[f_\omega(G(x_t), y_t; z_t)] - \frac{\gamma_x \eta}{2}\mathbb{E}[\|\nabla_x f_\omega(G(x_t), y_t; z_t)\|^2] + \frac{\gamma_x \eta}{2}\mathbb{E}[\|\nabla_x f_\omega(G(x_t), y_t; z_t) - p_t\|^2]$$

$$- \frac{\gamma_x \eta}{2}\mathbb{E}[\|p_t\|^2] + \frac{\omega + \ell}{2}\gamma_x^2 \eta^2 \mathbb{E}[\|p_t\|^2]$$

$$\leq \mathbb{E}[f_\omega(G(x_t), y_t; z_t)] - \frac{\gamma_x \eta}{2}\mathbb{E}[\|\nabla_x f_\omega(G(x_t), y_t; z_t)\|^2] + \gamma_x \eta \mathbb{E}[\|\nabla_x f_\omega(H(x_t), y_t; z_t) - p_t\|^2]$$

$$+ \gamma_x \eta \mathbb{E}[\|\nabla_x f_\omega(G(x_t), y_t; z_t) - \nabla_x f_\omega(H(x_t), y_t; z_t)\|^2] - \frac{\gamma_x \eta}{4}\mathbb{E}[\|p_t\|^2]$$

$$\leq \mathbb{E}[f_\omega(G(x_t), y_t; z_t)] - \frac{\gamma_x \eta}{2}\mathbb{E}[\|\nabla_x f_\omega(G(x_t), y_t; z_t)\|^2] + \gamma_x \eta \mathbb{E}[\|\nabla_x f_\omega(H(x_t), y_t; z_t) - p_t\|^2]$$

$$+ \gamma_x \eta K \sum_{k=1}^{K} A_k \mathbb{E}[\|g^{(k)}(h_t^{(k-1)}) - h_t^{(k)}\|^2] - \frac{\gamma_x \eta}{4}\mathbb{E}[\|p_t\|^2], \tag{35}$$

where the second-to-last step holds due to $\eta \leq \frac{1}{2\gamma_x(\omega+\ell)}$.

Similarly, since $f_\omega(G(x_t), y_t; z_t)$ is $(\omega + \ell)$-smooth with respect to $y$, we obtain

$$\mathbb{E}[f_\omega(G(x_{t+1}), y_{t+1}; z_t)]$$

$$\leq \mathbb{E}[f_\omega(G(x_{t+1}), y_t; z_t)] + \mathbb{E}[\langle \nabla_y f_\omega(G(x_{t+1}), y_t; z_t), y_{t+1} - y_t \rangle] + \frac{\omega + \ell}{2} \mathbb{E}[\|y_{t+1} - y_t\|^2]$$

$$= \mathbb{E}[f_\omega(G(x_{t+1}), y_t; z_t)] + \gamma_y \eta \mathbb{E}[\langle \nabla_y f_\omega(G(x_{t+1}), y_t; z_t) - \nabla_y f_\omega(G(x_t), y_t; z_t), q_t \rangle]$$

$$+ \gamma_y \eta \mathbb{E}[\langle \nabla_y f_\omega(G(x_t), y_t; z_t), q_t \rangle] + \frac{\omega + \ell}{2}\gamma_y^2 \eta^2 \mathbb{E}[\|q_t\|^2]$$

$$\leq \mathbb{E}[f_\omega(G(x_{t+1}), y_t; z_t)] + 4\gamma_y \eta \mathbb{E}[\|\nabla_y f_\omega(G(x_{t+1}), y_t; z_t) - \nabla_y f_\omega(G(x_t), y_t; z_t)\|^2] + \frac{\gamma_y \eta}{4}\mathbb{E}[\|q_t\|^2]$$

$$+ \frac{\gamma_y \eta}{2}\mathbb{E}[\|\nabla_y f_\omega(G(x_t), y_t; z_t)\|^2] + \frac{\gamma_y \eta}{2}\mathbb{E}[\|q_t\|^2] + \frac{\omega + \ell}{2}\gamma_y^2 \eta^2 \mathbb{E}[\|q_t\|^2]$$

$$\leq \mathbb{E}[f_\omega(G(x_{t+1}), y_t; z_t)] + \frac{\gamma_y \eta}{2}\mathbb{E}[\|\nabla_y f_\omega(G(x_t), y_t; z_t)\|^2] + 4\gamma_y \eta \gamma_x^2 \eta^2 \ell^2 \mathbb{E}[\|p_t\|^2]$$

$$+ \left(\frac{3\gamma_y \eta}{4} + \frac{\omega + \ell}{2}\gamma_y^2 \eta^2\right) \mathbb{E}[\|q_t\|^2]. \tag{36}$$

By combining the inequalities above with Lemma B.3, the proof is complete.

$\square$

**Lemma B.5.** *Given Assumptions 3.1-3.3, the following inequality holds:*

*1. The estimation error between $\nabla_x f(G(x_t), y_t)$ and $\nabla_x f(H(x_t), y_t)$ satisfies:*

$$\mathbb{E}[\|\nabla_x f(G(x_t), y_t) - \nabla_x f(H(x_t), y_t)\|^2] \leq K \sum_{k=1}^{K} A_k \mathbb{E}[\|g^{(k)}(h_t^{(k-1)}) - h_t^{(k)}\|^2], \tag{37}$$

*where $A_k = \left(C_g^{2(K-1)} C_f^2 L_g^2 \left(\sum_{j=k}^{K-1} C_g^{j-k}\right)^2 + C_g^{2K} L_f^2 C_g^{2(K-k)}\right).$*

2. *The bounded error between $\nabla_x f_\omega(H(x_t), y_t; z_t; \hat{\xi}_{t+1})$ and $\nabla_x f_\omega(H(x_t), y_t; z_t)$ satisfies:*

$$\mathbb{E}[\|\nabla_x f_\omega(H(x_t), y_t; z_t; \hat{\xi}_{t+1}) - \nabla_x f_\omega(H(x_t), y_t; z_t)\|^2] \leq C_p^2 \sigma^2 . \tag{38}$$

3. *For any $k \in \{1, \cdots, K\}$, the descent error between $h_{t+1}^{(k)}$ and $h_t^{(k)}$ satisfies:*

$$\mathbb{E}[\|h_{t+1}^{(k)} - h_t^{(k)}\|^2] \leq 2\alpha^2 \eta^4 \sum_{j=1}^{k} (2C_g^2)^{k-j} \mathbb{E}[\|h_t^{(j)} - g^{(j)}(h_t^{(j-1)})\|^2]$$

$$+ (2C_g^2)^k \gamma_x^2 \eta^2 \mathbb{E}[\|p_t\|^2] + 2\alpha^2 \eta^4 \sigma^2 \sum_{j=1}^{k} (2C_g^2)^{j-1} . \tag{39}$$

4. *For any $\lambda_k > 0$ where $k \in \{1, \cdots, K\}$, the estimation error between $g^{(k)}(h_t^{(k-1)})$ and $h_t^{(k)}$ satisfies:*

$$\sum_{k=1}^{K} \lambda_k \mathbb{E}[\|g^{(k)}(h_{t+1}^{(k-1)}) - h_{t+1}^{(k)}\|^2] \leq (1 - \alpha\eta^2) \sum_{k=1}^{K} \lambda_k \mathbb{E}[\|g^{(k)}(h_t^{(k-1)}) - h_t^{(k)}\|^2]$$

$$+ 2\alpha^2 \eta^4 \sum_{k=1}^{K} \Big( \sum_{j=k+1}^{K} \lambda_j (2C_g^2)^{j-k} \Big) \mathbb{E}[\|g^{(k)}(h_t^{(k-1)}) - h_t^{(k)}\|^2] + \gamma_x^2 \eta^2 \sum_{k=1}^{K} \lambda_k (2C_g^2)^k \mathbb{E}[\|p_t\|^2]$$

$$+ 2\alpha^2 \eta^4 \sigma^2 \sum_{k=1}^{K} \lambda_k \sum_{j=0}^{k-1} (2C_g^2)^j . \tag{40}$$

*Proof.* First, we have

$$\mathbb{E}[\|\nabla_x f_\omega(G(x_t), y_t; z_t) - \nabla_x f_\omega(H(x_t), y_t; z_t)\|^2]$$
$$= \mathbb{E}[\|\nabla_x f(G(x_t), y_t) + \omega(x_t - z_t) - \nabla_x f(H(x_t), y_t) - \omega(x_t - z_t)\|^2]$$
$$= \mathbb{E}[\|\nabla_x f(G(x_t), y_t) - \nabla_x f(H(x_t), y_t)\|^2]$$
$$\leq K \sum_{k=1}^{K} A_k \mathbb{E}[\|g^{(k)}(h_t^{(k-1)}) - h_t^{(k)}\|^2] , \tag{41}$$

where the last step follows from Lemma B.2, Eq. (25) in [37].

Similarly,

$$\mathbb{E}[\|\nabla_x f_\omega(H(x_t), y_t; z_t; \hat{\xi}_{t+1}) - \nabla_x f_\omega(H(x_t), y_t; z_t)\|^2]$$
$$= \mathbb{E}[\|\nabla g^{(1)}(h_t^{(0)}; \xi_{t+1}^{(1)}) \cdots \nabla g^{(K)}(h_t^{(K-1)}; \xi_{t+1}^{(K)}) \nabla_1 f(h_t^{(K)}, y_t; \zeta_{t+1}) + \omega(x_t - z_t)$$
$$\quad - \nabla g^{(1)}(h_t^{(0)}) \cdots \nabla g^{(K-1)}(h_t^{(K-2)}) \nabla g^{(K)}(h_t^{(K-1)}) \nabla_1 f(h_t^{(K)}, y_t) + \omega(x_t - z_t)\|^2]$$
$$= \mathbb{E}[\|\nabla g^{(1)}(h_t^{(0)}; \xi_{t+1}^{(1)}) \cdots \nabla g^{(K)}(h_t^{(K-1)}; \xi_{t+1}^{(K)}) \nabla_1 f(h_t^{(K)}, y_t; \zeta_{t+1})$$
$$\quad - \nabla g^{(1)}(h_t^{(0)}) \cdots \nabla g^{(K-1)}(h_t^{(K-2)}) \nabla g^{(K)}(h_t^{(K-1)}) \nabla_1 f(h_t^{(K)}, y_t)\|^2]$$
$$\leq (K+1) C_g^{2(K-1)} (KC_f^2 + C_g^2) \sigma^2 , \tag{42}$$

where the last step follows from Lemma B.2, Eq. (28) in [37]. From the definition of $C_p^2$, the proof is complete.

Subsequently, the remaining inequalities follow from Lemma B.4 and Lemma B.5 in [37]. $\square$

**Lemma B.6.** *Given Assumptions 3.1-3.3, we derive*

$$\mathbb{E}[\|p_{t+1} - \nabla_x f_\omega(H(x_{t+1}), y_{t+1}; z_{t+1})\|^2] \leq (1 - \rho_x \eta^2) \mathbb{E}[\|p_t - \nabla_x f_\omega(H(x_t), y_t; z_t)\|^2]$$

$$+ 2C_p^2 2\alpha^2 \eta^4 \sum_{k=1}^{K} \Big( \sum_{j=k}^{K} (2C_g^2)^{j-k} \Big) \mathbb{E}[\|h_t^{(k)} - g^{(k)}(h_t^{(k-1)})\|^2] + 2C_p^2 \sum_{k=0}^{K} (2C_g^2)^k \gamma_x^2 \eta^2 \mathbb{E}[\|p_t\|^2]$$

$$+ 2C_p^2 \gamma_y^2 \eta^2 \mathbb{E}[\|q_t\|^2] + 4C_p^2 \alpha^2 \eta^4 \sum_{k=1}^{K} \sum_{j=1}^{k} (2C_g^2)^{j-1} \sigma^2 + 2\rho_x^2 \eta^4 C_p^2 \sigma^2 . \tag{43}$$

*Proof.*

$$\mathbb{E}[\|p_{t+1} - \nabla_x f_\omega(H(x_{t+1}), y_{t+1}; z_{t+1})\|^2]$$

$$= \mathbb{E}[\|(1 - \rho_x\eta^2)(p_t - \nabla_x f_\omega(H(x_t), y_t; z_t; \hat{\xi}_{t+1})) + \nabla_x f_\omega(H(x_{t+1}), y_{t+1}; z_{t+1}; \hat{\xi}_{t+1})$$
$$- \nabla_x f_\omega(H(x_{t+1}), y_{t+1}; z_{t+1})\|^2]$$

$$= \mathbb{E}\Big[\Big\|(1 - \rho_x\eta^2)\Big(p_t - \nabla_x f_\omega(H(x_t), y_t; z_t)\Big) + \Big(\nabla_x f_\omega(H(x_{t+1}), y_{t+1}; z_{t+1}; \hat{\xi}_{t+1})$$
$$- \nabla_x f_\omega(H(x_t), y_t; z_t; \hat{\xi}_{t+1}) + \nabla_x f_\omega(H(x_t), y_t; z_t) - \nabla_x f_\omega(H(x_{t+1}), y_{t+1}; z_{t+1})\Big)$$
$$+ \rho_x\eta^2\Big(\nabla_x f_\omega(H(x_t), y_t; z_t; \hat{\xi}_{t+1}) - \nabla_x f_\omega(H(x_t), y_t; z_t)\Big)\Big\|^2\Big]$$

$$\leq (1 - \rho_x\eta^2)^2 \mathbb{E}[\|p_t - \nabla g^{(1)}(h_t^{(0)}) \cdots \nabla g^{(K)}(h_t^{(K-1)}) \nabla_1 f_\omega(h_t^{(K)}, y_t; z_t)\|^2]$$

$$+ 2\mathbb{E}[\|\nabla g^{(1)}(h_{t+1}^{(0)}; \xi_{t+1}^{(1)}) \cdots \nabla g^{(K)}(h_{t+1}^{(K-1)}; \xi_{t+1}^{(K)}) \nabla_1 f(h_{t+1}^{(K)}, y_{t+1}; \zeta_{t+1})$$
$$- \nabla g^{(1)}(h_t^{(0)}; \xi_{t+1}^{(1)}) \cdots \nabla g^{(K)}(h_t^{(K-1)}; \xi_{t+1}^{(K)}) \nabla_1 f(h_t^{(K)}, y_t; \zeta_{t+1})\|^2]$$

$$+ 2\rho_x^2\eta^4 \mathbb{E}[\|\nabla_x f_\omega(H(x_t), y_t; z_t; \hat{\xi}_{t+1}) - \nabla_x f_\omega(H(x_t), y_t; z_t)\|^2]$$

$$\leq (1 - \rho_x\eta^2)\mathbb{E}[\|p_t - \nabla_x f_\omega(H(x_t), y_t; z_t)\|^2] + 2T_1 + 2\rho_x^2\eta^4 C_p^2\sigma^2 , \qquad (44)$$

where the last step holds due to Lemma B.5 and third step holds due to the following inequality:

$$\mathbb{E}[\|\nabla_x f_\omega(H(x_{t+1}), y_{t+1}; z_{t+1}; \hat{\xi}_{t+1}) - \nabla_x f_\omega(H(x_t), y_t; z_t; \hat{\xi}_{t+1})$$
$$+ \nabla_x f_\omega(H(x_t), y_t; z_t) - \nabla_x f_\omega(H(x_{t+1}), y_{t+1}; z_{t+1})\|^2]$$

$$= \mathbb{E}[\|\nabla_x f(H(x_{t+1}), y_{t+1}; \hat{\xi}_{t+1}) + \omega(x_{t+1} - z_{t+1}) - \nabla_x f(H(x_t), y_t; \hat{\xi}_{t+1}) - \omega(x_t - z_t)$$
$$+ \nabla_x f(H(x_t), y_t) + \omega(x_t - z_t) - \nabla_x f(H(x_{t+1}), y_{t+1}) - \omega(x_{t+1} - z_{t+1})\|^2]$$

$$\leq \mathbb{E}[\|\nabla_x f(H(x_{t+1}), y_{t+1}; \hat{\xi}_{t+1}) - \nabla_x f(H(x_t), y_t; \hat{\xi}_{t+1})\|^2]$$

$$= \mathbb{E}[\|\nabla g^{(1)}(h_{t+1}^{(0)}; \xi_{t+1}^{(1)}) \cdots \nabla g^{(K)}(h_{t+1}^{(K-1)}; \xi_{t+1}^{(K)}) \nabla_1 f(h_{t+1}^{(K)}, y_{t+1}; \zeta_{t+1})$$
$$- \nabla g^{(1)}(h_t^{(0)}; \xi_{t+1}^{(1)}) \cdots \nabla g^{(K)}(h_t^{(K-1)}; \xi_{t+1}^{(K)}) \nabla_1 f(h_t^{(K)}, y_t; \zeta_{t+1})\|^2] . \qquad (45)$$

Next, we bound $T_1$ as follows:

$$T_1 = \mathbb{E}[\|\nabla g^{(1)}(h_{t+1}^{(0)}; \xi_{t+1}^{(1)}) \cdots \nabla g^{(K)}(h_{t+1}^{(K-1)}; \xi_{t+1}^{(K)}) \nabla_1 f(h_{t+1}^{(K)}, y_{t+1}; \zeta_{t+1})$$
$$- \nabla g^{(1)}(h_t^{(0)}; \xi_{t+1}^{(1)}) \cdots \nabla g^{(K)}(h_t^{(K-1)}; \xi_{t+1}^{(K)}) \nabla_1 f(h_t^{(K)}, y_t; \zeta_{t+1})\|^2]$$

$$\leq (K+1)C_g^{2K}L_f^2 \mathbb{E}[\|h_{t+1}^{(K)} - h_t^{(K)}\|^2] + (K+1)C_g^{2K}L_f^2 \mathbb{E}[\|y_{t+1} - y_t\|^2]$$

$$+ (K+1)C_g^{2(K-1)}C_f^2 L_g^2 \mathbb{E}[\|h_{t+1}^{(K-1)} - h_t^{(K-1)}\|^2] + \cdots + (K+1)C_g^{2(K-1)}C_f^2 L_g^2 \mathbb{E}[\|h_{t+1}^{(1)} - h_t^{(1)}\|^2]$$

$$+ (K+1)C_g^{2(K-1)}C_f^2 L_g^2 \mathbb{E}[\|x_{t+1} - x_t\|^2]$$

$$\leq C_p^2 \sum_{k=1}^K \mathbb{E}[\|h_{t+1}^{(k)} - h_t^{(k)}\|^2] + C_p^2 \mathbb{E}[\|x_{t+1} - x_t\|^2] + C_p^2 \mathbb{E}[\|y_{t+1} - y_t\|^2]$$

$$\leq C_p^2 2\alpha^2\eta^4 \sum_{k=1}^K \Big(\sum_{j=k}^K (2C_g^2)^{j-k}\Big) \mathbb{E}[\|h_t^{(k)} - g^{(k)}(h_t^{(k-1)})\|^2] + C_p^2 \sum_{k=0}^K (2C_g^2)^k \gamma_x^2\eta^2 \mathbb{E}[\|p_t\|^2]$$

$$+ C_p^2 \gamma_y^2\eta^2 \mathbb{E}[\|q_t\|^2] + 2C_p^2\alpha^2\eta^4 \sum_{k=1}^K \sum_{j=1}^k (2C_g^2)^{j-1}\sigma^2 . \qquad (46)$$

Combining this with the previous inequalities completes the proof. $\qquad \square$

**Lemma B.7.** *Given Assumption 3.1-3.3, we derive:*

$$\mathbb{E}[\|q_{t+1} - \nabla_y f_\omega(H(x_{t+1}), y_{t+1}; z_{t+1})\|^2] \leq (1 - \rho_y\eta^2)\mathbb{E}[\|q_t - \nabla_y f_\omega(H(x_t), y_t; z_t)\|^2]$$

$$+ 4\alpha^2\eta^4 L_f^2 \sum_{k=1}^K (2C_g^2)^{K-k} \mathbb{E}[\|h_t^{(k)} - g^{(k)}(h_t^{(k-1)})\|^2] + 2L_f^2(2C_g^2)^K \gamma_x^2\eta^2 \mathbb{E}[\|p_t\|^2]$$

$$+ 2L_f^2\gamma_y^2\eta^2\mathbb{E}[\|q_t\|^2] + 4\alpha^2\eta^4 L_f^2\sigma^2 \sum_{k=1}^{K}(2C_g^2)^{k-1} + 2\rho_y^2\eta^4\sigma^2 \ . \tag{47}$$

*Proof.*

$$\mathbb{E}[\|q_{t+1} - \nabla_y f_\omega(H(x_{t+1}), y_{t+1}; z_{t+1})\|^2]$$

$$\leq (1 - \rho_y\eta^2)\mathbb{E}[\|q_t - \nabla_y f_\omega(H(x_t), y_t; z_t)\|^2] + 2\mathbb{E}[\|\nabla_2 f(h_{t+1}^{(K)}, y_{t+1}; \zeta_{t+1}) - \nabla_2 f(h_t^{(K)}, y_t; \zeta_{t+1})\|^2]$$

$$+ 2\rho_y^2\eta^4\mathbb{E}[\|\nabla_2 f_\omega(h_t^{(K)}, y_t; z_t; \zeta_{t+1}) - \nabla_2 f_\omega(h_t^{(K)}, y_t; z_t)\|^2]$$

$$\leq (1 - \rho_y\eta^2)\mathbb{E}[\|q_t - \nabla_y f_\omega(H(x_t), y_t; z_t)\|^2] + 2L_f^2\mathbb{E}[\|h_{t+1}^{(K)} - h_t^{(K)}\|^2] + 2L_f^2\mathbb{E}[\|y_{t+1} - y_t\|^2]$$

$$+ 2\rho_y^2\eta^4\sigma^2 \ , \tag{48}$$

by applying Lemma B.5, the proof is complete. $\qquad\square$

## B.2  Proof of the Theorem 4.1

**Theorem B.8.** *(Restatement of Theorem 4.1) Given Assumptions 3.1-3.3, when $\rho_x > 0$, $\rho_y > 0$, $\alpha > 0$, $\omega = O(\ell)$, and the hyperparameter conditions are satisfied:*

$$\gamma_x \leq \min\left\{ \frac{\ell^2}{6\omega(\omega+\ell)^2} \ , \ \frac{64\ell}{(\omega-\ell)^2\sqrt{C_{x_{yz}^1}^2 + 1}} \ , \ \frac{1}{48\omega c_{\gamma_z} C_{x_{yz}^2}} \ , \ \frac{1}{8\sqrt{c_{\gamma_y}}\ell} \ , \ \frac{1}{16c_{\gamma_y}(2L_{\omega,d} + \omega + \ell)} \ , \right.$$

$$\frac{\sqrt{\rho_x}}{4C_p\sqrt{2\sum_{k=0}^{K}(2C_g^2)^k}} \ , \ \frac{\sqrt{\rho_y}}{4\sqrt{5c_{\gamma_y}(2C_g^2)^K L_f}} \ , \ \frac{\sqrt{\alpha}}{8\sqrt{\sum_{k=1}^{K} d_k(2C_g^2)^k}} \ , \ \frac{\sqrt{\rho_x}}{8\sqrt{c_{\gamma_y}}C_p} \ ,$$

$$\left. \frac{\sqrt{\rho_x}}{16C_p\sqrt{2\sum_{k=1}^{K}\sum_{j=k}^{K}(2C_g^2)^j}} \ , \ \frac{\sqrt{\rho_y}}{16\sqrt{5c_{\gamma_y}(2C_g^2)^K}} \ , \ \frac{\sqrt{\rho_y}}{4\sqrt{10c_{\gamma_y}}L_f} \right\}$$

$$\gamma_y = \gamma_x \underbrace{\frac{(\omega-\ell)^2}{64\ell^2}}_{c_{\gamma_y}} \ , \qquad \gamma_z = \gamma_x \underbrace{\frac{(\omega-\ell)^3\mu}{98304\omega\ell^2}}_{c_{\gamma_z}} \ ,$$

$$\eta \leq \min\left\{ \frac{1}{\sqrt{\rho_x}}, \frac{1}{\sqrt{\rho_y}}, \frac{1}{\sqrt{\alpha}}, \frac{1}{\gamma_z}, \frac{1}{2\gamma_x(\omega+\ell)} \ , \ \frac{1}{2}\sqrt{\frac{\hat{\lambda}_k}{\alpha\left(\sum_{j=k+1}^{K}\hat{\lambda}_j(2C_g^2)^{j-k}\right)}} \right\} \ , \tag{49}$$

*Algorithm 1 achieves the following convergence upper bound:*

$$\frac{1}{T}\sum_{t=0}^{T-1}\left(\mathbb{E}[\|\nabla_x f(G(x_t), y_t)\|^2] + \kappa\mathbb{E}[\|\nabla_y f(G(x_t), y_t)\|^2]\right)$$

$$\leq O\left(\frac{\kappa\mathcal{P}_0}{\gamma_x\eta T}\right) + O\left(\frac{\kappa\sigma^2}{\rho_x\eta^2 TS}\right) + O\left(\frac{\kappa\sigma^2}{\rho_y\eta^2 TS}\right) + O\left(\frac{\kappa\sigma^2}{\alpha\eta^2 TS}\right)$$

$$+ O\left(\kappa\frac{\alpha^2\eta^2\sigma^2}{\rho_x}\right) + O\left(\kappa\rho_x\eta^2\sigma^2\right) + O\left(\kappa\frac{\alpha^2\eta^2\sigma^2}{\rho_y}\right) + O\left(\kappa\rho_y\eta^2\sigma^2\right) + O\left(\kappa\alpha\eta^2\sigma^2\right) \ , \tag{50}$$

*where $\mathcal{P}_0 = f_\omega(G(x_0), y_0; z_0) - 2f_{\omega,d}(y_0; z_0) + 2g(z_0)$, with the definitions of the involved terms provided in Eq. (25).*

*Proof.* To establish the convergence rate of Algorithm 1, we propose a novel potential function as follows:

$$\mathcal{H}_t = \underbrace{f_\omega(G(x_t), y_t; z_t) - 2h_{\omega,d}(y_t; z_t) + 2h(z_t)}_{\mathcal{P}_t}$$

$$+ \nu_a\mathbb{E}[\|p_t - \nabla_x f_\omega(H(x_t), y_t; z_t)\|^2] + \nu_b\mathbb{E}[\|q_t - \nabla_y f_\omega(H(x_t), y_t; z_t)\|^2]$$

$$+ \sum_{k=1}^{K} \lambda_k \mathbb{E}[\|h_t^{(k)} - g^{(k)}(h_t^{(k-1)})\|^2] \,, \tag{51}$$

where the coefficient $\nu_a$, $\nu_b$ and $\{\lambda_k\}_{k=1}^{K}$ are positive.

### B.2.1 Bound $\mathcal{P}_{t+1} - \mathcal{P}_t$

First, we aim to derive an upper bound for $\mathcal{P}_{t+1} - \mathcal{P}_t$. To this end, we begin by applying Lemmas B.3 and B.4, from which we obtain

$$
\begin{aligned}
&\mathcal{P}_{t+1} - \mathcal{P}_t \\
&\leq f_\omega(G(x_{t+1}), y_{t+1}; z_{t+1}) - f_\omega(G(x_t), y_t; z_t) - 2\Big(h_{\omega,d}(y_{t+1}; z_{t+1}) - h_{\omega,d}(y_t; z_t)\Big) \\
&\quad + 2\Big(h(z_{t+1}) - h(z_t)\Big) \\
&\leq -\frac{\gamma_x \eta}{2}\mathbb{E}[\|\nabla_x f_\omega(G(x_t), y_t; z_t)\|^2] + \frac{\gamma_y \eta}{2}\mathbb{E}[\|\nabla_y f_\omega(G(x_t), y_t; z_t)\|^2] - \frac{\omega}{2\gamma_z \eta}\mathbb{E}[\|z_{t+1} - z_t\|^2] \\
&\quad + \gamma_x \eta \mathbb{E}[\|\nabla_x f_\omega(G(x_t), y_t; z_t) - p_t\|^2] + \gamma_x \eta K \sum_{k=1}^{K} A_k \mathbb{E}[\|g^{(k)}(h_t^{(k-1)}) - h_t^{(k)}\|^2] \\
&\quad + \Big(4\gamma_y \eta \gamma_x^2 \eta^2 \ell^2 - \frac{\gamma_x \eta}{4}\Big)\mathbb{E}[\|p_t\|^2] + \Big(\frac{3\gamma_y \eta}{4} + \frac{\omega + \ell}{2}\gamma_y^2 \eta^2\Big)\mathbb{E}[\|q_t\|^2] \\
&\quad - 2\gamma_y \eta \mathbb{E}\langle \nabla_y f_\omega(x^*(y_t, z_t), y_t; z_t), q_t\rangle] + \gamma_y^2 \eta^2 L_{\omega,d}\mathbb{E}[\|q_t\|^2] \\
&\quad - \omega\langle z_{t+1} - z_t, z_{t+1} + z_t - 2x^*(y_{t+1}; z_{t+1})\rangle + \omega\langle z_{t+1} - z_t, z_{t+1} + z_t - 2x^*(y^*(z_{t+1}); z_t)\rangle \\
&= -\frac{\gamma_x \eta}{2}\mathbb{E}[\|\nabla_x f_\omega(G(x_t), y_t; z_t)\|^2] + \frac{\gamma_y \eta}{2}\mathbb{E}[\|\nabla_y f_\omega(G(x_t), y_t; z_t)\|^2] - \frac{\omega}{2\gamma_z \eta}\mathbb{E}[\|z_{t+1} - z_t\|^2] \\
&\quad + \gamma_x \eta \mathbb{E}[\|\nabla_x f_\omega(G(x_t), y_t; z_t) - p_t\|^2] + \gamma_x \eta K \sum_{k=1}^{K} A_k \mathbb{E}[\|g^{(k)}(h_t^{(k-1)}) - h_t^{(k)}\|^2] \\
&\quad + \Big(4\gamma_y \eta \gamma_x^2 \eta^2 \ell^2 - \frac{\gamma_x \eta}{4}\Big)\mathbb{E}[\|p_t\|^2] + \Big(\frac{3\gamma_y \eta}{4} + \frac{\omega + \ell}{2}\gamma_y^2 \eta^2 + \gamma_y^2 \eta^2 L_{\omega,d}\Big)\mathbb{E}[\|q_t\|^2] \\
&\quad - 2\gamma_y \eta \mathbb{E}\langle \nabla_y f_\omega(x^*(y_t, z_t), y_t; z_t), q_t\rangle] + 2\omega\langle z_{t+1} - z_t, x^*(y_{t+1}; z_{t+1}) - x^*(y^*(z_{t+1}); z_t)\rangle \,.
\end{aligned}
\tag{52}
$$

Next, we derive

$$
\begin{aligned}
&2\langle z_{t+1} - z_t, x^*(y_{t+1}; z_{t+1}) - x^*(y^*(z_{t+1}); z_t)\rangle \\
&= 2\langle z_{t+1} - z_t, x^*(y_{t+1}; z_{t+1}) - x^*(y^*(z_{t+1}); z_{t+1})\rangle \\
&\quad + 2\langle z_{t+1} - z_t, x^*(y^*(z_{t+1}); z_{t+1}) - x^*(y^*(z_{t+1}); z_t)\rangle \\
&\leq \frac{1}{6\gamma_z \eta}\|z_{t+1} - z_t\|^2 + 6\gamma_z \eta\|x^*(y_{t+1}; z_{t+1}) - x^*(y^*(z_{t+1}); z_{t+1})\|^2 \\
&\quad + 2\|z_{t+1} - z_t\|\|x^*(y^*(z_{t+1}); z_{t+1}) - x^*(y^*(z_{t+1}); z_t)\| \\
&\leq \frac{1}{6\gamma_z \eta}\|z_{t+1} - z_t\|^2 + 6\gamma_z \eta\|x^*(y_{t+1}; z_{t+1}) - x^*(y^*(z_{t+1}); z_{t+1})\|^2 + 2C_{x_{yz}^2}\|z_{t+1} - z_t\|^2 \\
&= \Big(\frac{1}{6\gamma_z \eta} + 2C_{x_{yz}^2}\Big)\|z_{t+1} - z_t\|^2 + 6\gamma_z \eta\|x^*(y_{t+1}; z_{t+1}) - x^*(y^*(z_{t+1}); z_{t+1})\|^2 \,, \tag{53}
\end{aligned}
$$

where the third step follows from Lemma B.2.

Additionally, we have

$$
\begin{aligned}
&- 2\gamma_y \eta \mathbb{E}[\langle \nabla_y f_\omega(x^*(y_t, z_t), y_t; z_t), q_t\rangle] \\
&= -2\gamma_y \eta \mathbb{E}[\langle \nabla_y f_\omega(x^*(y_t, z_t), y_t; z_t) - \nabla_y f_\omega(G(x_t), y_t; z_t), q_t\rangle] - 2\gamma_y \eta \mathbb{E}[\langle \nabla_y f_\omega(G(x_t), y_t; z_t), q_t\rangle] \\
&= -2\gamma_y \eta \mathbb{E}[\langle \nabla_y f_\omega(x^*(y_t, z_t), y_t; z_t) - \nabla_y f_\omega(G(x_t), y_t; z_t), q_t\rangle] \\
&\quad - \gamma_y \eta \mathbb{E}[\|\nabla_y f_\omega(G(x_t), y_t; z_t)\|^2] - \gamma_y \eta \mathbb{E}[\|q_t\|^2] + \gamma_y \eta \mathbb{E}[\|\nabla_y f_\omega(G(x_t), y_t; z_t) - q_t\|^2]
\end{aligned}
$$

$$
\begin{aligned}
&\leq \gamma_y\eta\frac{1}{c}\mathbb{E}[\|\nabla_y f_\omega(x^*(y_t,z_t),y_t;z_t) - \nabla_y f_\omega(G(x_t),y_t;z_t)\|^2] + \gamma_y\eta c\mathbb{E}[\|q_t\|^2] \\
&\quad - \gamma_y\eta\mathbb{E}[\|\nabla_y f_\omega(G(x_t),y_t;z_t)\|^2] - \gamma_y\eta\mathbb{E}[\|q_t\|^2] + 2\gamma_y\eta\mathbb{E}[\|\nabla_y f_\omega(G(x_t),y_t;z_t) - \nabla_y f_\omega(H(x_t),y_t;z_t)\|^2] \\
&\quad + 2\gamma_y\eta\mathbb{E}[\|\nabla_y f_\omega(H(x_t),y_t;z_t) - q_t\|^2] \\
&\leq \gamma_y\eta\frac{1}{c}\mathbb{E}[\|\nabla_y f_\omega(x^*(y_t,z_t),y_t;z_t) - \nabla_y f_\omega(G(x_t),y_t;z_t)\|^2] - \gamma_y\eta\mathbb{E}[\|\nabla_y f_\omega(G(x_t),y_t;z_t)\|^2] \\
&\quad + 2\gamma_y\eta L_f^2 K \sum_{k=1}^{K} C_g^{2(K-k)}\mathbb{E}[\|g^{(k)}(h_t^{(k-1)}) - h_t^{(k)}\|^2] + 2\gamma_y\eta\mathbb{E}[\|\nabla_y f_\omega(H(x_t),y_t;z_t) - q_t\|^2] \\
&\quad - (1-c)\gamma_y\eta\mathbb{E}[\|q_t\|^2]\,,
\end{aligned}
\tag{54}
$$

where $c > 0$ is a constant, and the last step follows from

$$
\begin{aligned}
&\mathbb{E}[\|\nabla_y f_\omega(G(x_t),y_t;z_t) - \nabla_y f_\omega(H(x_t),y_t;z_t)\|^2] \\
&= \mathbb{E}[\|\nabla_y f(G(x_t),y_t) - \nabla_y f(H(x_t),y_t)\|^2] \\
&\leq L_f^2\mathbb{E}[\|G^{(K)}(x_t) - h_t^{(K)}\|^2] \\
&\leq L_f^2 K\sum_{k=1}^{K} C_g^{2(K-k)}\mathbb{E}[\|g^{(k)}(h_t^{(k-1)}) - h_t^{(k)}\|^2]\,.
\end{aligned}
\tag{55}
$$

Setting $c = \frac{1}{8}$, we obtain

$$
\begin{aligned}
&\mathcal{P}_{t+1} - \mathcal{P}_t \\
&\leq -\frac{\gamma_x\eta}{2}\mathbb{E}[\|\nabla_x f_\omega(G(x_t),y_t;z_t)\|^2] + \frac{\gamma_y\eta}{2}\mathbb{E}[\|\nabla_y f_\omega(G(x_t),y_t;z_t)\|^2] - \frac{\omega}{2\gamma_z\eta}\mathbb{E}[\|z_{t+1} - z_t\|^2] \\[4pt]
&\quad + \gamma_x\eta\mathbb{E}[\|\nabla_x f_\omega(G(x_t),y_t;z_t) - p_t\|^2] + \gamma_x\eta K\sum_{k=1}^{K} A_k\mathbb{E}[\|g^{(k)}(h_t^{(k-1)}) - h_t^{(k)}\|^2] \\
&\quad + \Big(4\gamma_y\eta\gamma_x^2\eta^2\ell^2 - \frac{\gamma_x\eta}{4}\Big)\mathbb{E}[\|p_t\|^2] + \Big(\frac{3\gamma_y\eta}{4} + \frac{\omega+\ell}{2}\gamma_y^2\eta^2 + \gamma_y^2\eta^2 L_{\omega,d} - \frac{7}{8}\gamma_y\eta\Big)\mathbb{E}[\|q_t\|^2] \\
&\quad + 8\gamma_y\eta\mathbb{E}[\|\nabla_y f_\omega(x^*(y_t,z_t),y_t;z_t) - \nabla_y f_\omega(G(x_t),y_t;z_t)\|^2] - \gamma_y\eta\mathbb{E}[\|\nabla_y f_\omega(G(x_t),y_t;z_t)\|^2] \\
&\quad + 2\gamma_y\eta L_f^2 K\sum_{k=1}^{K} C_g^{2(K-k)}\mathbb{E}[\|g^{(k)}(h_t^{(k-1)}) - h_t^{(k)}\|^2] + 2\gamma_y\eta\mathbb{E}[\|\nabla_y f_\omega(H(x_t),y_t;z_t) - q_t\|^2] \\
&\quad + \omega\Big(\frac{1}{6\gamma_z\eta} + 2C_{x_{yz}^2}\Big)\mathbb{E}[\|z_{t+1} - z_t\|^2] + 6\omega\gamma_z\eta\mathbb{E}[\|x^*(y_{t+1};z_{t+1}) - x^*(y^*(z_{t+1});z_{t+1})\|^2] \\[4pt]
&\leq -\frac{\gamma_x\eta}{2}\mathbb{E}[\|\nabla_x f_\omega(G(x_t),y_t;z_t)\|^2] - \frac{\gamma_y\eta}{2}\mathbb{E}[\|\nabla_y f_\omega(G(x_t),y_t;z_t)\|^2] + \omega\Big(2C_{x_{yz}^2} - \frac{1}{3\gamma_z\eta}\Big)\mathbb{E}[\|z_{t+1} - z_t\|^2] \\
&\quad + \gamma_x\eta\mathbb{E}[\|\nabla_x f_\omega(G(x_t),y_t;z_t) - p_t\|^2] + 2\gamma_y\eta\mathbb{E}[\|\nabla_y f_\omega(H(x_t),y_t;z_t) - q_t\|^2] \\
&\quad + \sum_{k=1}^{K}\Big(\gamma_x\eta K A_k + 2\gamma_y\eta L_f^2 K C_g^{2(K-k)}\Big)\mathbb{E}[\|g^{(k)}(h_t^{(k-1)}) - h_t^{(k)}\|^2] \\
&\quad + \Big(4\gamma_y\eta\gamma_x^2\eta^2\ell^2 - \frac{\gamma_x\eta}{4}\Big)\mathbb{E}[\|p_t\|^2] + \Big(\frac{3\gamma_y\eta}{4} + \frac{\omega+\ell}{2}\gamma_y^2\eta^2 + \gamma_y^2\eta^2 L_{\omega,d} - \frac{7}{8}\gamma_y\eta\Big)\mathbb{E}[\|q_t\|^2] \\
&\quad + 8\gamma_y\eta\ell^2\mathbb{E}[\|x^*(y_t,z_t) - x_t\|^2] + 6\omega\gamma_z\eta\mathbb{E}[\|x^*(y_{t+1};z_{t+1}) - x^*(y^*(z_{t+1});z_{t+1})\|^2]\,.
\end{aligned}
\tag{56}
$$

Furthermore, due to the strong convexity of $f_\omega(G(x_t),y_t;z_t)$ regarding $x$, we obtain

$$
\mathbb{E}[\|x^*(y_t,z_t) - x_t\|^2] \leq \frac{1}{(\omega-\ell)^2}\mathbb{E}[\|\nabla_x f_\omega(G(x_t),y_t;z_t)\|^2]\,.
\tag{57}
$$

In addition, by introducing

$$
y^+(z_t) = y_t + \gamma_y\eta\nabla_y f_\omega(x^*(y_t,z_t),y_t;z_t)\,,
\tag{58}
$$

we obtain

$$
\begin{aligned}
&\mathbb{E}[\|x^*(y_{t+1}; z_{t+1}) - x^*(y^*(z_{t+1}); z_{t+1})\|^2] \\
&= \mathbb{E}[\|x^*(y_{t+1}; z_{t+1}) - x^*(z_{t+1})\|^2] \\
&\leq 4\mathbb{E}[\|x^*(z_{t+1}) - x^*(z_t)\|^2] + 4\mathbb{E}[\|x^*(z_t) - x^*(y^+(z_t); z_t)\|^2] \\
&\quad + 4\mathbb{E}[\|x^*(y^+(z_t); z_t) - x^*(y_{t+1}; z_t)\|^2] + 4\mathbb{E}[\|x^*(y_{t+1}; z_t) - x^*(y_{t+1}; z_{t+1})\|^2] \\
&\leq 4C_{x_z}^2 \mathbb{E}[\|z_{t+1} - z_t\|^2] + 4\mathbb{E}[\|x^*(z_t) - x^*(y^+(z_t); z_t)\|^2] + 4C_{x_{yz}^1}^2 \mathbb{E}[\|y^+(z_t) - y_{t+1}\|^2] \\
&\quad + 4C_{x_{yz}^2}^2 \mathbb{E}[\|z_t - z_{t+1}\|^2] \\
&= 4(C_{x_z}^2 + C_{x_{yz}^2}^2)\mathbb{E}[\|z_{t+1} - z_t\|^2] + 4\mathbb{E}[\|x^*(z_t) - x^*(y^+(z_t); z_t)\|^2] \\
&\quad + 4\gamma_y^2\eta^2 C_{x_{yz}^1}^2 \mathbb{E}[\|\nabla_y f_\omega(x^*(y_t, z_t), y_t; z_t) - q_t\|^2] \\
&\leq 4(C_{x_z}^2 + C_{x_{yz}^2}^2)\mathbb{E}[\|z_{t+1} - z_t\|^2] + 4\mathbb{E}[\|x^*(z_t) - x^*(y^+(z_t); z_t)\|^2] \\
&\quad + 8\gamma_y^2\eta^2 C_{x_{yz}^1}^2 \mathbb{E}[\|\nabla_y f_\omega(x^*(y_t, z_t), y_t; z_t) - \nabla_y f_\omega(G(x_t), y_t; z_t)\|^2] \\
&\quad + 8\gamma_y^2\eta^2 C_{x_{yz}^1}^2 \mathbb{E}[\|\nabla_y f_\omega(G(x_t), y_t; z_t) - q_t\|^2] \\
&\leq 4(C_{x_z}^2 + C_{x_{yz}^2}^2)\mathbb{E}[\|z_{t+1} - z_t\|^2] + 4\mathbb{E}[\|x^*(z_t) - x^*(y^+(z_t); z_t)\|^2] \\
&\quad + 8\gamma_y^2\eta^2 C_{x_{yz}^1}^2 \ell^2 \mathbb{E}[\|x^*(y_t, z_t) - G(x_t)\|^2] + 16\gamma_y^2\eta^2 C_{x_{yz}^1}^2 \mathbb{E}[\|\nabla_y f_\omega(H(x_t), y_t; z_t) - q_t\|^2] \\
&\quad + 16\gamma_y^2\eta^2 C_{x_{yz}^1}^2 \mathbb{E}[\|\nabla_y f_\omega(G(x_t), y_t; z_t) - \nabla_y f_\omega(H(x_t), y_t; z_t)\|^2] \\
&\leq 4(C_{x_z}^2 + C_{x_{yz}^2}^2)\mathbb{E}[\|z_{t+1} - z_t\|^2] + 4\mathbb{E}[\|x^*(z_t) - x^*(y^+(z_t); z_t)\|^2] \\
&\quad + 8\gamma_y^2\eta^2 C_{x_{yz}^1}^2 \ell^2 \mathbb{E}[\|x^*(y_t, z_t) - x_t\|^2] + 16\gamma_y^2\eta^2 C_{x_{yz}^1}^2 \mathbb{E}[\|\nabla_y f_\omega(H(x_t), y_t; z_t) - q_t\|^2] \\
&\quad + 16\gamma_y^2\eta^2 C_{x_{yz}^1}^2 L_f^2 K \sum_{k=1}^{K} C_g^{2(K-k)} \mathbb{E}[\|g^{(k)}(h_t^{(k-1)}) - h_t^{(k)}\|^2] .
\end{aligned}
\tag{59}
$$

Moreover, we derive

$$
\begin{aligned}
&\|x^*(z_t) - x^*(y^+(z_t); z_t)\|^2 \\
&\leq \frac{2}{\omega - \ell}(h_{\omega,p}(x^*(y^+(z_t); z_t); z_t) - h_{\omega,p}(x^*(z_t); z_t)) \\
&\leq \frac{2}{\omega - \ell}(h_{\omega,p}(x^*(y^+(z_t); z_t); z_t) - f_\omega(x^*(y^+(z_t); z_t), y^+(z_t); z_t) \\
&\qquad + f_\omega(x^*(y^+(z_t); z_t), y^+(z_t); z_t) - h_{\omega,p}(x^*(z_t); z_t)) \\
&\leq \frac{1}{(\omega - \ell)\mu}\|\nabla_y f_\omega(x^*(y^+(z_t); z_t), y^+(z_t); z_t)\|^2 \\
&\leq \frac{2}{(\omega - \ell)\mu}\|\nabla_y f_\omega(x^*(y^+(z_t); z_t), y^+(z_t); z_t) - \nabla_y f_\omega(x^*(y_t, z_t), y_t; z_t)\|^2 \\
&\quad + \frac{2}{(\omega - \ell)\mu}\|\nabla_y f_\omega(x^*(y_t, z_t), y_t; z_t)\|^2 \\
&\leq \frac{2\ell^2 C_{x_{yz}^1}^2}{(\omega - \ell)\mu}\|y^+(z_t) - y_t\|^2 + \frac{2\ell^2}{(\omega - \ell)\mu}\|y^+(z_t) - y_t\|^2 + \frac{2}{(\omega - \ell)\mu}\|\nabla_y f_\omega(x^*(y_t, z_t), y_t; z_t)\|^2 \\
&\leq \frac{2(1 + \gamma_y^2\eta^2\ell^2 C_{x_{yz}^1}^2 + \gamma_y^2\eta^2\ell^2)}{(\omega - \ell)\mu}\|\nabla_y f_\omega(x^*(y_t, z_t), y_t; z_t)\|^2 \\
&\leq \frac{4(1 + \gamma_y^2\eta^2\ell^2 C_{x_{yz}^1}^2 + \gamma_y^2\eta^2\ell^2)}{(\omega - \ell)\mu}\|\nabla_y f_\omega(x^*(y_t, z_t), y_t; z_t) - \nabla_y f_\omega(G(x_t), y_t; z_t)\|^2 \\
&\quad + \frac{4(1 + \gamma_y^2\eta^2\ell^2 C_{x_{yz}^1}^2 + \gamma_y^2\eta^2\ell^2)}{(\omega - \ell)\mu}\|\nabla_y f_\omega(G(x_t), y_t; z_t)\|^2
\end{aligned}
$$

$$
\leq \frac{4(1 + \gamma_y^2\eta^2\ell^2 C_{x_{yz}^1}^2 + \gamma_y^2\eta^2\ell^2)\ell^2}{(\omega - \ell)\mu}\|x^*(y_t, z_t) - x_t\|^2
$$
$$
+ \frac{4(1 + \gamma_y^2\eta^2\ell^2 C_{x_{yz}^1}^2 + \gamma_y^2\eta^2\ell^2)}{(\omega - \ell)\mu}\|\nabla_y f_\omega(G(x_t), y_t; z_t)\|^2
$$
$$
\leq \frac{4(1 + \gamma_y^2\eta^2\ell^2 C_{x_{yz}^1}^2 + \gamma_y^2\eta^2\ell^2)\ell^2}{(\omega - \ell)^3\mu}\|\nabla_x f_\omega(G(x_t), y_t; z_t)\|^2
$$
$$
+ \frac{4(1 + \gamma_y^2\eta^2\ell^2 C_{x_{yz}^1}^2 + \gamma_y^2\eta^2\ell^2)}{(\omega - \ell)\mu}\|\nabla_y f_\omega(G(x_t), y_t; z_t)\|^2 . \tag{60}
$$

Therefore, we obtain the following upper bound for $\mathcal{P}_{t+1} - \mathcal{P}_t$:

$$
\mathcal{P}_{t+1} - \mathcal{P}_t
$$
$$
\leq -\frac{\gamma_x\eta}{2}\mathbb{E}[\|\nabla_x f_\omega(G(x_t), y_t; z_t)\|^2] - \frac{\gamma_y\eta}{2}\mathbb{E}[\|\nabla_y f_\omega(G(x_t), y_t; z_t)\|^2] + \omega(2C_{x_{yz}^2}^2 - \frac{1}{3\gamma_z\eta})\mathbb{E}[\|z_{t+1} - z_t\|^2]
$$
$$
+ \gamma_x\eta\mathbb{E}[\|\nabla_x f_\omega(G(x_t), y_t; z_t) - p_t\|^2] + 2\gamma_y\eta\mathbb{E}[\|\nabla_y f_\omega(H(x_t), y_t; z_t) - q_t\|^2]
$$
$$
+ \sum_{k=1}^{K}\left(\gamma_x\eta K A_k + 2\gamma_y\eta L_f^2 K C_g^{2(K-k)}\right)\mathbb{E}[\|g^{(k)}(h_t^{(k-1)}) - h_t^{(k)}\|^2] + 24\omega\gamma_z\eta(C_{x_z}^2 + C_{x_{yz}^2}^2)\mathbb{E}[\|z_{t+1} - z_t\|^2]
$$
$$
+ \left(4\gamma_y\eta\gamma_x^2\eta^2\ell^2 - \frac{\gamma_x\eta}{4}\right)\mathbb{E}[\|p_t\|^2] + \left(\frac{3\gamma_y\eta}{4} + \frac{\omega + \ell}{2}\gamma_y^2\eta^2 + \gamma_y^2\eta^2 L_{\omega,d} - \frac{7}{8}\gamma_y\eta\right)\mathbb{E}[\|q_t\|^2]
$$
$$
+ \left(\frac{8\gamma_y\eta\ell^2 + 48\omega\gamma_z\gamma_y^2\eta^3 C_{x_{yz}^1}^2\ell^2}{(\omega - \ell)^2} + \frac{96\omega\gamma_z\eta(1 + \gamma_y^2\eta^2\ell^2 C_{x_{yz}^1}^2 + \gamma_y^2\eta^2\ell^2)\ell^2}{(\omega - \ell)^3\mu}\right)\mathbb{E}[\|\nabla_x f_\omega(G(x_t), y_t; z_t)\|^2]
$$
$$
+ 24\omega\gamma_z\eta\frac{4(1 + \gamma_y^2\eta^2\ell^2 C_{x_{yz}^1}^2 + \gamma_y^2\eta^2\ell^2)}{(\omega - \ell)\mu}\mathbb{E}[\|\nabla_y f_\omega(G(x_t), y_t; z_t)\|^2]
$$
$$
+ 96\omega\gamma_z\gamma_y^2\eta^3 C_{x_{yz}^1}^2\mathbb{E}[\|\nabla_y f_\omega(H(x_t), y_t; z_t) - q_t\|^2]
$$
$$
+ 96\omega\gamma_z\gamma_y^2\eta^3 C_{x_{yz}^1}^2 L_f^2 K\sum_{k=1}^{K} C_g^{2(K-k)}\mathbb{E}[\|g^{(k)}(h_t^{(k-1)}) - h_t^{(k)}\|^2]
$$
$$
\leq \left(\frac{8\gamma_y\eta\ell^2 + 48\omega\gamma_z\gamma_y^2\eta^3 C_{x_{yz}^1}^2\ell^2}{(\omega - \ell)^2} + \frac{96\omega\gamma_z\eta(1 + \gamma_y^2\eta^2\ell^2 C_{x_{yz}^1}^2 + \gamma_y^2\eta^2\ell^2)\ell^2}{(\omega - \ell)^3\mu} - \frac{\gamma_x\eta}{2}\right)\mathbb{E}[\|\nabla_x f_\omega(G(x_t), y_t; z_t)\|^2]
$$
$$
+ \left(24\omega\gamma_z\eta\frac{4(1 + \gamma_y^2\eta^2\ell^2 C_{x_{yz}^1}^2 + \gamma_y^2\eta^2\ell^2)}{(\omega - \ell)\mu} - \frac{\gamma_y\eta}{2}\right)\mathbb{E}[\|\nabla_y f_\omega(G(x_t), y_t; z_t)\|^2]
$$
$$
+ \omega\left(24\gamma_z\eta(C_{x_z}^2 + C_{x_{yz}^2}^2) + 2C_{x_{yz}^2}^2 - \frac{1}{3\gamma_z\eta}\right)\mathbb{E}[\|z_{t+1} - z_t\|^2]
$$
$$
+ \gamma_x\eta\mathbb{E}[\|\nabla_x f_\omega(G(x_t), y_t; z_t) - p_t\|^2] + \left(2\gamma_y\eta + 96\omega\gamma_z\gamma_y^2\eta^3 C_{x_{yz}^1}^2\right)\mathbb{E}[\|\nabla_y f_\omega(H(x_t), y_t; z_t) - q_t\|^2]
$$
$$
+ \sum_{k=1}^{K}\left(\gamma_x\eta K A_k + 2\gamma_y\eta L_f^2 K C_g^{2(K-k)} + 96\omega\gamma_z\gamma_y^2\eta^3 C_{x_{yz}^1}^2 L_f^2 K C_g^{2(K-k)}\right)\mathbb{E}[\|g^{(k)}(h_t^{(k-1)}) - h_t^{(k)}\|^2]
$$
$$
+ \left(4\gamma_y\eta\gamma_x^2\eta^2\ell^2 - \frac{\gamma_x\eta}{4}\right)\mathbb{E}[\|p_t\|^2] + \left(\frac{3\gamma_y\eta}{4} + \frac{\omega + \ell}{2}\gamma_y^2\eta^2 + \gamma_y^2\eta^2 L_{\omega,d} - \frac{7}{8}\gamma_y\eta\right)\mathbb{E}[\|q_t\|^2] . \tag{61}
$$

### B.2.2  Bound $\mathcal{H}_{t+1} - \mathcal{H}_t$

In the following, we aim to derive an upper bound for $\mathcal{H}_{t+1} - \mathcal{H}_t$:

$$
\mathcal{H}_{t+1} - \mathcal{H}_t
$$
$$
\leq \left(\frac{8\gamma_y\eta\ell^2 + 48\omega\gamma_z\gamma_y^2\eta^3 C_{x_{yz}^1}^2\ell^2}{(\omega - \ell)^2} + \frac{96\omega\gamma_z\eta(1 + \gamma_y^2\eta^2\ell^2 C_{x_{yz}^1}^2 + \gamma_y^2\eta^2\ell^2)\ell^2}{(\omega - \ell)^3\mu} - \frac{\gamma_x\eta}{2}\right)\mathbb{E}[\|\nabla_x f_\omega(G(x_t), y_t; z_t)\|^2]
$$
$$
+ \left(24\omega\gamma_z\eta\frac{4(1 + \gamma_y^2\eta^2\ell^2 C_{x_{yz}^1}^2 + \gamma_y^2\eta^2\ell^2)}{(\omega - \ell)\mu} - \frac{\gamma_y\eta}{2}\right)\mathbb{E}[\|\nabla_y f_\omega(G(x_t), y_t; z_t)\|^2]
$$

$$+ \omega \left( 24\gamma_z \eta (C_{x_z}^2 + C_{x_{yz}^2}^2) + 2C_{x_{yz}^2}^2 - \frac{1}{3\gamma_z \eta} \right) \mathbb{E}[\|z_{t+1} - z_t\|^2]$$

$$+ \left( \gamma_x \eta - \rho_x \eta^2 \nu_a \right) \mathbb{E}[\|\nabla_x f_\omega (G(x_t), y_t; z_t) - p_t\|^2]$$

$$+ \left( 2\gamma_y \eta + 96\omega\gamma_z \gamma_y^2 \eta^3 C_{x_{yz}^1}^2 - \rho_y \eta^2 \nu_b \right) \mathbb{E}[\|\nabla_y f_\omega (H(x_t), y_t; z_t) - q_t\|^2]$$

$$+ \sum_{k=1}^K \Bigg( \gamma_x \eta K A_k + 2\gamma_y \eta L_f^2 K C_g^{2(K-k)} + 96\omega\gamma_z \gamma_y^2 \eta^3 C_{x_{yz}^1}^2 L_f^2 K C_g^{2(K-k)} + 2C_p^2 2\alpha^2 \eta^4 \nu_a \Big( \sum_{j=k}^K (2C_g^2)^{j-k} \Big)$$

$$+ 4\alpha^2 \eta^4 \nu_b L_f^2 (2C_g^2)^{K-k} + 2\alpha^2 \eta^4 \Big( \sum_{j=k+1}^K \lambda_j (2C_g^2)^{j-k} \Big) - \alpha\eta^2 \lambda_k \Bigg) \mathbb{E}[\|g^{(k)}(h_t^{(k-1)}) - h_t^{(k)}\|^2]$$

$$+ \left( 4\gamma_y \eta \gamma_x^2 \eta^2 \ell^2 + 2C_p^2 \sum_{k=0}^K (2C_g^2)^k \gamma_x^2 \eta^2 \nu_a + 2L_f^2 (2C_g^2)^K \gamma_x^2 \eta^2 \nu_b + \gamma_x^2 \eta^2 \sum_{k=1}^K \lambda_k (2C_g^2)^k - \frac{\gamma_x \eta}{4} \right) \mathbb{E}[\|p_t\|^2]$$

$$+ \left( \frac{3\gamma_y \eta}{4} + \frac{\omega + \ell}{2} \gamma_y^2 \eta^2 + \gamma_y^2 \eta^2 L_{\omega,d} + 2C_p^2 \gamma_y^2 \eta^2 \nu_a + 2L_f^2 \gamma_y^2 \eta^2 \nu_b - \frac{7}{8}\gamma_y \eta \right) \mathbb{E}[\|q_t\|^2]$$

$$+ 4C_p^2 \alpha^2 \eta^4 \sigma^2 \nu_a \sum_{k=1}^K \sum_{j=1}^k (2C_g^2)^{j-1} + 2\rho_x^2 \eta^4 \nu_a C_p^2 \sigma^2 + 4\alpha^2 \eta^4 \nu_b L_f^2 \sigma^2 \sum_{k=1}^K (2C_g^2)^{k-1}$$

$$+ 2\rho_y^2 \eta^4 \nu_b \sigma^2 + 2\alpha^2 \eta^4 \sigma^2 \sum_{k=1}^K \lambda_k \sum_{j=0}^{k-1} (2C_g^2)^j \ . \tag{62}$$

We consider the following choice for bounding $\mathbb{E}[\|\nabla_x f_\omega (G(x_t), y_t; z_t)\|^2]$ and $\mathbb{E}[\|\nabla_y f_\omega (G(x_t), y_t; z_t)\|^2]$:

$$\frac{8\gamma_y \eta \ell^2}{(\omega - \ell)^2} - \frac{\gamma_x \eta}{8} \le 0 \ , \qquad \frac{48\omega\gamma_z \gamma_y^2 \eta^3 C_{x_{yz}^1}^2 \ell^2}{(\omega - \ell)^2} - \frac{\gamma_x \eta}{512} \le 0 \ ,$$

$$\frac{96\omega\gamma_z \eta (1 + \gamma_y^2 \eta^2 \ell^2 C_{x_{yz}^1}^2 + \gamma_y^2 \eta^2 \ell^2)\ell^2}{(\omega - \ell)^3 \mu} - \frac{\gamma_x \eta}{512} \le 0 \ ,$$

$$24\omega\gamma_z \eta \frac{4(1 + \gamma_y^2 \eta^2 \ell^2 C_{x_{yz}^1}^2 + \gamma_y^2 \eta^2 \ell^2)}{(\omega - \ell)\mu} - \frac{\gamma_y \eta}{8} \le 0 \ . \tag{63}$$

Since $\gamma_z \eta \le 1$ and $C_{x_{yz}^1} = \frac{\omega + \ell}{\omega - \ell}$, we set

$$\gamma_y = \gamma_x \underbrace{\frac{(\omega - \ell)^2}{64\ell^2}}_{c_{\gamma_y}} \ , \qquad \gamma_z = \gamma_x \underbrace{\frac{(\omega - \ell)^3 \mu}{98304\omega\ell^2}}_{c_{\gamma_z}} \ ,$$

$$\gamma_x \le \min\{ \frac{\ell^2}{6\omega(\omega + \ell)^2} \ , \quad \frac{64\ell}{(\omega - \ell)^2 \sqrt{C_{x_{yz}^1}^2 + 1}} \} \ . \tag{64}$$

Additionally, we consider the following choice for bounding $\mathbb{E}[\|z_{t+1} - z_t\|^2]$, we set

$$\omega \left( 2C_{x_{yz}^2}^2 + 24\gamma_z \eta (C_{x_z}^2 + C_{x_{yz}^2}^2) - \frac{1}{3\gamma_z \eta} \right) \le -\frac{\omega}{4\gamma_z \eta} \ . \tag{65}$$

Specifically, we enforce

$$2\omega C_{x_{yz}^2}^2 \le \frac{\omega}{24\gamma_z \eta} \ , \qquad 24\omega\gamma_z \eta (C_{x_z}^2 + C_{x_{yz}^2}^2) \le \frac{\omega}{24\gamma_z \eta} \ . \tag{66}$$

Then, based on Eq. (64), from $C_{x_z} = C_{x_{yz}^2}$ and $\eta < 1$, we obtain

$$\gamma_x \le \frac{1}{48\omega c_{\gamma_z} C_{x_{yz}^2}^2} \ . \tag{67}$$

To remove the term $\mathbb{E}[\|\nabla_x f_\omega (H(x_t), y_t; z_t) - p_t\|^2]$, we impose

$$\gamma_x \eta - \rho_x \eta^2 \nu_a \le 0 \ . \tag{68}$$

From this, we obtain the parameter choice

$$\nu_a = \frac{\gamma_x}{\rho_x \eta} \; . \tag{69}$$

Similarly, to remove the term $\mathbb{E}[\|\nabla_y f_\omega(H(x_t), y_t; z_t) - q_t\|^2]$, we impose

$$2\gamma_y \eta + 96\omega\gamma_z\gamma_y^2\eta^3 C_{x_{yz}^1}^2 - \rho_y \eta^2 \nu_b \le 0 \; . \tag{70}$$

From the second inequality in Eq. (63) and definition of $c_{\gamma_y}$, we have

$$96\omega\gamma_z\gamma_y^2\eta^3 C_{x_{yz}^1}^2 \le \frac{2\gamma_x\eta}{512}\frac{(\omega-\ell)^2}{\ell^2} = \frac{2\gamma_x\eta}{512}64c_{\gamma_y} \le \frac{1}{2}\gamma_y\eta \; . \tag{71}$$

As a result, we require

$$\frac{5}{2}\gamma_y\eta \le \rho_y\eta^2\nu_b \; , \tag{72}$$

which leads to the parameter choice

$$\nu_b = \frac{5\gamma_y}{2\rho_y\eta} \; . \tag{73}$$

Then, for any $k \in \{1, \cdots, K\}$, to remove the term $\mathbb{E}[\|g^{(k)}(h_t^{(k-1)}) - h_t^{(k)}\|^2]$, we set

$$\gamma_x\eta K A_k + 2\gamma_y\eta L_f^2 K C_g^{2(K-k)} + 96\omega\gamma_z\gamma_y^2\eta^3 C_{x_{yz}^1}^2 L_f^2 K C_g^{2(K-k)} + 4\alpha^2\eta^4 C_p^2 \nu_a \Big(\sum_{j=k}^{K}(2C_g^2)^{j-k}\Big)$$

$$+ 4\alpha^2\eta^4\nu_b L_f^2 (2C_g^2)^{K-k} + 2\alpha^2\eta^4 \sum_{k=1}^{K}\Big(\sum_{j=k+1}^{K}\lambda_j(2C_g^2)^{j-k}\Big) - \alpha\eta^2\lambda_k \le 0 \; . \tag{74}$$

Plugging the value of $\nu_a$ and $\nu_b$, we obtain

$$\gamma_x\eta K A_k + 2\gamma_y\eta L_f^2 K C_g^{2(K-k)} + 96\omega\gamma_z\gamma_y^2\eta^3 C_{x_{yz}^1}^2 L_f^2 K C_g^{2(K-k)} + 4\alpha^2\eta^4 C_p^2 \frac{\gamma_x}{\rho_x\eta}\Big(\sum_{j=k}^{K}(2C_g^2)^{j-k}\Big)$$

$$+ 4\alpha^2\eta^4\frac{5\gamma_y}{2\rho_y\eta} L_f^2 (2C_g^2)^{K-k} + 2\alpha^2\eta^4\Big(\sum_{j=k+1}^{K}\lambda_j(2C_g^2)^{j-k}\Big) - \alpha\eta^2\lambda_k \le 0 \; . \tag{75}$$

To analyze this, we first simplify the expression:

$$\gamma_x\eta K A_k + 2\gamma_y\eta L_f^2 K C_g^{2(K-k)} + 96\omega\gamma_z\gamma_y^2\eta^3 C_{x_{yz}^1}^2 L_f^2 K C_g^{2(K-k)} + 4\alpha^2\eta^4 C_p^2 \frac{\gamma_x}{\rho_x\eta}\Big(\sum_{j=k}^{K}(2C_g^2)^{j-k}\Big)$$

$$+ 4\alpha^2\eta^4\frac{5\gamma_y}{2\rho_y\eta} L_f^2 (2C_g^2)^{K-k} - \frac{1}{2}\alpha\eta^2\lambda_k$$

$$\le \gamma_x\eta K A_k + 2\gamma_y\eta L_f^2 K C_g^{2(K-k)} + \frac{1}{2}\gamma_y\eta L_f^2 K C_g^{2(K-k)} + 4\alpha^2\eta^4 C_p^2 \frac{\gamma_x}{\rho_x\eta}\Big(\sum_{j=k}^{K}(2C_g^2)^{j-k}\Big)$$

$$+ 4\alpha^2\eta^4\frac{5\gamma_y}{2\rho_y\eta} L_f^2 (2C_g^2)^{K-k} - \frac{1}{2}\alpha\eta^2\lambda_k$$

$$\le \alpha\eta^2\Big[\frac{1}{\alpha\eta}\gamma_x K A_k + \frac{1}{\alpha\eta}\frac{5}{2}\gamma_y L_f^2 K C_g^{2(K-k)} + 4\alpha\eta C_p^2 \frac{\gamma_x}{\rho_x}\Big(\sum_{j=k}^{K}(2C_g^2)^{j-k}\Big)$$

$$+ 10\alpha\eta\frac{\gamma_y}{\rho_y} L_f^2 (2C_g^2)^{K-k} - \frac{1}{2}\lambda_k\Big] \; . \tag{76}$$

Due to $\alpha\eta^2 \le 1$, we enforce the following to be non-positive:

$$\lambda_k \ge \frac{\gamma_x}{\alpha\eta}\Big[2K A_k + 5c_{\gamma_y} L_f^2 K C_g^{2(K-k)}\Big] + \gamma_x\alpha\eta\Big[8C_p^2 \frac{1}{\rho_x}\Big(\sum_{j=k}^{K}(2C_g^2)^{j-k}\Big) + 20\frac{c_{\gamma_y}}{\rho_y} L_f^2 (2C_g^2)^{K-k}\Big]$$

$$= \frac{\gamma_x}{\alpha\eta}\Big[2KA_k + 5c_{\gamma_y}L_f^2 KC_g^{2(K-k)}\Big] + \alpha^2\eta^2\frac{\gamma_x}{\alpha\eta}\Big[8C_p^2\frac{1}{\rho_x}\Big(\sum_{j=k}^{K}(2C_g^2)^{j-k}\Big) + 20\frac{c_{\gamma_y}}{\rho_y}L_f^2(2C_g^2)^{K-k}\Big] \,.$$

$$(77)$$

Therefore, we obtain the parameter choice for any $\lambda_k$ where $k \in \{1, \cdots, K\}$:

$$\lambda_k = \frac{\gamma_x}{\alpha\eta}\Big[2KA_k + 5c_{\gamma_y}L_f^2 KC_g^{2(K-k)} + 8C_p^2\frac{\alpha}{\rho_x}\Big(\sum_{j=k}^{K}(2C_g^2)^{j-k}\Big) + 20\frac{\alpha c_{\gamma_y}}{\rho_y}L_f^2(2C_g^2)^{K-k}\Big]$$

$$\triangleq \frac{\gamma_x}{\alpha\eta}\hat{\lambda}_k \,. \qquad (78)$$

Moreover, we enforce

$$2\alpha^2\eta^4\Big(\sum_{j=k+1}^{K}\lambda_j(2C_g^2)^{j-k}\Big) - \alpha\eta^2\lambda_k \le -\frac{1}{2}\alpha\eta^2\lambda_k \,, \qquad (79)$$

for $k \in \{1, \cdots, K\}$, which leads to

$$\eta \le \frac{1}{2}\sqrt{\frac{\hat{\lambda}_k}{\alpha\Big(\sum_{j=k+1}^{K}\hat{\lambda}_j(2C_g^2)^{j-k}\Big)}} \,. \qquad (80)$$

To guarantee that $\mathbb{E}[\|p_t\|^2]$ cancels out, we enforce

$$4\gamma_y\eta\gamma_x^2\eta^2\ell^2 + 2C_p^2\sum_{k=0}^{K}(2C_g^2)^k\gamma_x^2\eta^2\nu_a + 2L_f^2(2C_g^2)^K\gamma_x^2\eta^2\nu_b + \gamma_x^2\eta^2\sum_{k=1}^{K}\lambda_k(2C_g^2)^k - \frac{\gamma_x\eta}{4} \le 0 \,.$$

$$(81)$$

This is equivalent to enforce

$$4\gamma_y\eta\gamma_x^2\eta^2\ell^2 + 2C_p^2\gamma_x^2\eta^2\frac{\gamma_x}{\rho_x\eta}\sum_{k=0}^{K}(2C_g^2)^k + \gamma_x^2\eta^2\frac{5\gamma_y}{2\rho_y\eta}2L_f^2(2C_g^2)^K + \gamma_x^2\eta^2\sum_{k=1}^{K}\frac{\gamma_x}{\alpha\eta}\hat{\lambda}_k(2C_g^2)^k - \frac{\gamma_x\eta}{4} \le 0 \,.$$

$$(82)$$

Specifically, we enforce

$$4\gamma_y\eta\gamma_x^2\eta^2\ell^2 \le \frac{\gamma_x\eta}{16} \,, \qquad 2C_p^2\gamma_x^2\eta^2\frac{\gamma_x}{\rho_x\eta}\sum_{k=0}^{K}(2C_g^2)^k \le \frac{\gamma_x\eta}{16} \,,$$

$$\gamma_x^2\eta^2\frac{5\gamma_y}{2\rho_y\eta}2L_f^2(2C_g^2)^K \le \frac{\gamma_x\eta}{16} \,, \qquad \gamma_x^2\eta^2\sum_{k=1}^{K}\frac{\gamma_x}{\alpha\eta}\hat{\lambda}_k(2C_g^2)^k \le \frac{\gamma_x\eta}{16} \,. \qquad (83)$$

To solve the first third inequalities, we obtain

$$\gamma_x \le \left\{\frac{1}{8\sqrt{c_{\gamma_y}}\ell} \,, \frac{\sqrt{\rho_x}}{4C_p\sqrt{2\sum_{k=0}^{K}(2C_g^2)^k}} \,, \frac{\sqrt{\rho_y}}{4\sqrt{5c_{\gamma_y}(2C_g^2)^K L_f}}\right\} \,. \qquad (84)$$

For the last inequality, it is equivalent to enforce

$$\gamma_x^2\eta^2\sum_{k=1}^{K}\frac{\gamma_x}{\alpha\eta}\Big(d_k + 8C_p^2\frac{\alpha}{\rho_x}\Big(\sum_{j=k}^{K}(2C_g^2)^{j-k}\Big) + 20\frac{\alpha c_{\gamma_y}}{\rho_y}L_f^2(2C_g^2)^{K-k}\Big)(2C_g^2)^k \le \frac{\gamma_x\eta}{16} \,, \qquad (85)$$

where

$$d_k = 2KA_k + 5c_{\gamma_y}L_f^2 KC_g^{2(K-k)} \,. \qquad (86)$$

Specifically, we enforce

$$\gamma_x^2 \eta^2 \sum_{k=1}^{K} \frac{\gamma_x}{\alpha\eta} d_k (2C_g^2)^k \leq \frac{\gamma_x \eta}{64} ,$$

$$\gamma_x^2 \eta^2 \sum_{k=1}^{K} \frac{\gamma_x}{\alpha\eta} \frac{8C_p^2 \alpha}{\rho_x} \sum_{j=k}^{K} (2C_g^2)^j \leq \frac{\gamma_x \eta}{64} ,$$

$$\gamma_x^2 \eta^2 \sum_{k=1}^{K} \frac{\gamma_x}{\alpha\eta} 20\alpha c_{\gamma_y} \frac{(2C_g^2)^K}{\rho_y} \leq \frac{\gamma_x \eta}{64} . \tag{87}$$

For the first inequality, since $d_k$ is independent of hyperparameters, we obtain

$$\gamma_x \leq \frac{\sqrt{\alpha}}{8\sqrt{\sum_{k=1}^{K} d_k (2C_g^2)^k}} . \tag{88}$$

For the remaining inequalities, we obtain

$$\gamma_x \leq \left\{ \frac{\sqrt{\rho_x}}{16C_p \sqrt{2\sum_{k=1}^{K}\sum_{j=k}^{K}(2C_g^2)^j}} , \frac{\sqrt{\rho_y}}{16\sqrt{5c_{\gamma_y}(2C_g^2)^K}} \right\} . \tag{89}$$

As for $\mathbb{E}[\|q_t\|^2]$, we enforce

$$\frac{3\gamma_y \eta}{4} + \frac{\omega+\ell}{2}\gamma_y^2\eta^2 + \gamma_y^2\eta^2 L_{\omega,d} + 2C_p^2\gamma_y^2\eta^2\nu_a + 2L_f^2\gamma_y^2\eta^2\nu_b - \frac{7}{8}\gamma_y\eta \leq 0 . \tag{90}$$

This is equivalent to enforce

$$\frac{3\gamma_y \eta}{4} + \frac{\omega+\ell}{2}\gamma_y^2\eta^2 + \gamma_y^2\eta^2 L_{\omega,d} + 2C_p^2\gamma_y^2\eta^2\frac{\gamma_x}{\rho_x\eta} + 2L_f^2\gamma_y^2\eta^2\frac{5\gamma_y}{2\rho_y\eta} - \frac{7}{8}\gamma_y\eta \leq 0 . \tag{91}$$

Specifically, we enforce

$$\gamma_y^2\eta^2 L_{\omega,d} + \frac{\gamma_y^2\eta^2(\omega+\ell)}{2} \leq \frac{\gamma_y\eta}{32} ,$$

$$2C_p^2\gamma_y^2\eta^2\frac{\gamma_x}{\rho_x\eta} \leq \frac{\gamma_y\eta}{32} , \qquad 2L_f^2\gamma_y^2\eta^2\frac{5\gamma_y}{2\rho_y\eta} \leq \frac{\gamma_y\eta}{32} . \tag{92}$$

To solve these inequalities, we obtain

$$\gamma_x \leq \left\{ \frac{1}{16c_{\gamma_y}(2L_{\omega,d}+\omega+\ell)} , \frac{\sqrt{\rho_x}}{8\sqrt{c_{\gamma_y}}C_p} , \frac{\sqrt{\rho_y}}{4\sqrt{10c_{\gamma_y}}L_f} \right\} . \tag{93}$$

In summary, by setting

$$\gamma_x \leq \min \left\{ \frac{\ell^2}{6\omega(\omega+\ell)^2} , \frac{64\ell}{(\omega-\ell)^2\sqrt{C_{x_{yz}^1}^2+1}} , \frac{1}{48\omega c_{\gamma_z}C_{x_{yz}^2}} , \frac{1}{8\sqrt{c_{\gamma_y}}\ell} , \frac{1}{16c_{\gamma_y}(2L_{\omega,d}+\omega+\ell)} , \right.$$

$$\frac{\sqrt{\rho_x}}{4C_p\sqrt{2\sum_{k=0}^{K}(2C_g^2)^k}} , \frac{\sqrt{\rho_y}}{4\sqrt{5c_{\gamma_y}(2C_g^2)^K}L_f} , \frac{\sqrt{\alpha}}{8\sqrt{\sum_{k=1}^{K}d_k(2C_g^2)^k}} , \frac{\sqrt{\rho_x}}{8\sqrt{c_{\gamma_y}}C_p} ,$$

$$\left. \frac{\sqrt{\rho_x}}{16C_p\sqrt{2\sum_{k=1}^{K}\sum_{j=k}^{K}(2C_g^2)^j}} , \frac{\sqrt{\rho_y}}{16\sqrt{5c_{\gamma_y}(2C_g^2)^K}} , \frac{\sqrt{\rho_y}}{4\sqrt{10c_{\gamma_y}}L_f} \right\} ,$$

$$\gamma_y = \gamma_x \underbrace{\frac{(\omega-\ell)^2}{64\ell^2}}_{c_{\gamma_y}} , \qquad \gamma_z = \gamma_x \underbrace{\frac{(\omega-\ell)^3\mu}{98304\omega\ell^2}}_{c_{\gamma_z}} ,$$

$$\eta \le \min\left\{ \frac{1}{\sqrt{\rho_x}}, \frac{1}{\sqrt{\rho_y}}, \frac{1}{\sqrt{\alpha}}, \frac{1}{\gamma_z}, \frac{1}{2\gamma_x(\omega+\ell)}, \frac{1}{2}\sqrt{\frac{\hat{\lambda}_k}{\alpha\left(\sum_{j=k+1}^{K}\hat{\lambda}_j(2C_g^2)^{j-k}\right)}} \right\}, \tag{94}$$

we obtain

$$\mathcal{H}_{t+1} - \mathcal{H}_t$$
$$\le -\frac{\gamma_x\eta}{4}\mathbb{E}[\|\nabla_x f_\omega(G(x_t), y_t; z_t)\|^2] - \frac{\gamma_y\eta}{4}\mathbb{E}[\|\nabla_y f_\omega(G(x_t), y_t; z_t)\|^2] - \frac{\omega}{4\gamma_z\eta}\mathbb{E}[\|z_{t+1} - z_t\|^2]$$
$$+ 4C_p^2\alpha^2\eta^4\sigma^2\frac{\gamma_x}{\rho_x\eta}\sum_{k=1}^{K}\sum_{j=1}^{k}(2C_g^2)^{j-1} + 2\rho_x^2\eta^4\frac{\gamma_x}{\rho_x\eta}C_p^2\sigma^2 + 4\alpha^2\eta^4\sigma^2\frac{5\gamma_y}{2\rho_y\eta}L_f^2\sum_{k=1}^{K}(2C_g^2)^{k-1}$$
$$+ 2\rho_y^2\eta^4\frac{5\gamma_y}{2\rho_y\eta}\sigma^2 + 2\alpha^2\eta^4\sigma^2\sum_{k=1}^{K}\frac{\gamma_x}{\alpha\eta}\hat{\lambda}_k\sum_{j=0}^{k-1}(2C_g^2)^j$$
$$\le -\frac{\gamma_x\eta}{4}\mathbb{E}[\|\nabla_x f_\omega(G(x_t), y_t; z_t)\|^2] - c_{\gamma_y}\frac{\gamma_x\eta}{4}\mathbb{E}[\|\nabla_y f_\omega(G(x_t), y_t; z_t)\|^2] - c_{\gamma_z}\omega\frac{\gamma_x\eta}{4}\mathbb{E}[\|x_t - z_t\|^2]$$
$$+ 4C_p^2\alpha^2\eta^4\sigma^2\frac{\gamma_x}{\rho_x\eta}\sum_{k=1}^{K}\sum_{j=1}^{k}(2C_g^2)^{j-1} + 2\rho_x^2\eta^4\frac{\gamma_x}{\rho_x\eta}C_p^2\sigma^2 + 4\alpha^2\eta^4\sigma^2\frac{5\gamma_x c_{\gamma_y}}{2\rho_y\eta}L_f^2\sum_{k=1}^{K}(2C_g^2)^{k-1}$$
$$+ 2\rho_y^2\eta^4\frac{5\gamma_x c_{\gamma_y}}{2\rho_y\eta}\sigma^2 + 2\alpha^2\eta^4\sigma^2\sum_{k=1}^{K}\frac{\gamma_x}{\alpha\eta}\hat{\lambda}_k\sum_{j=0}^{k-1}(2C_g^2)^j \ . \tag{95}$$

From

$$\|\nabla_x f(G(x_t), y_t)\|^2 \le 2\|\nabla_x f_\omega(G(x_t), y_t; z_t)\|^2 + 2\omega^2\|x_t - z_t\|^2 \ ,$$
$$\|\nabla_y f(G(x_t), y_t)\|^2 = \|\nabla_y f_\omega(G(x_t), y_t; z_t)\|^2 \ , \tag{96}$$

by summing over $t$ from 0 to $T-1$ and reformulate it, we obtain

$$\frac{1}{T}\sum_{t=0}^{T-1}\left(\mathbb{E}[\|\nabla_x f(G(x_t), y_t)\|^2] + \kappa\mathbb{E}[\|\nabla_y f(G(x_t), y_t)\|^2]\right)$$
$$\le \frac{1}{T}\sum_{t=0}^{T-1}\left(2\mathbb{E}[\|\nabla_x f_\omega(G(x_t), y_t; z_t)\|^2] + 2\kappa\mathbb{E}[\|\nabla_y f_\omega(G(x_t), y_t; z_t)\|^2] + 2\omega^2\mathbb{E}[\|x_t - z_t\|^2]\right)$$
$$\le \max\left\{ \frac{8}{\gamma_x\eta}, \frac{8\kappa}{\gamma_x\eta c_{\gamma_y}}, \frac{8\omega}{\gamma_x\eta c_{\gamma_z}} \right\}\left( \frac{\mathcal{H}_0 - \mathcal{H}_T}{T} + 4C_p^2\frac{\alpha^2\eta^2}{\rho_x}\sum_{k=1}^{K}\sum_{j=1}^{k}(2C_g^2)^{j-1}\sigma^2 + 2\rho_x\eta^2 C_p^2\sigma^2 \right.$$
$$\left. + 10c_{\gamma_y}\frac{\alpha^2\eta^2}{\rho_y}L_f^2\sum_{k=1}^{K}(2C_g^2)^{k-1}\sigma^2 + 5\rho_y\eta^2 c_{\gamma_y}\sigma^2 + 2\alpha\eta^2\sigma^2\sum_{k=1}^{K}\hat{\lambda}_k\sum_{j=0}^{k-1}(2C_g^2)^j \right) \ . \tag{97}$$

When $t = 0$, we derive

$$\mathcal{H}_0 = \mathcal{P}_0 + \frac{\gamma_x}{\rho_x\eta}\mathbb{E}[\|p_0 - \nabla_x f_\omega(H(x_0), y_0; z_0)\|^2] + \frac{5\gamma_y}{2\rho_y\eta}\mathbb{E}[\|q_0 - \nabla_y f_\omega(H(x_0), y_0; z_0)\|^2]$$
$$+ \sum_{k=1}^{K}\frac{\gamma_x}{\alpha\eta}\hat{\lambda}_k\mathbb{E}[\|h_0^{(k)} - g^{(k)}(h_0^{(k-1)})\|^2]$$
$$\le \mathcal{P}_0 + \frac{\gamma_x}{\rho_x\eta}\frac{\sigma^2}{S} + \frac{5\gamma_x c_{\gamma_y}}{2\rho_y\eta}\frac{\sigma^2}{S} + \sum_{k=1}^{K}\frac{\gamma_x}{\alpha\eta}\hat{\lambda}_k\frac{\sigma^2}{S} \ . \tag{98}$$

Finally, we have

$$\frac{1}{T}\sum_{t=0}^{T-1}\left(\mathbb{E}[\|\nabla_x f(G(x_t), y_t)\|^2] + \kappa\mathbb{E}[\|\nabla_y f(G(x_t), y_t)\|^2]\right)$$

$$\leq O\Big(\frac{\kappa \mathcal{P}_0}{\gamma_x \eta T}\Big) + O\Big(\frac{\kappa \sigma^2}{\rho_x \eta^2 TS}\Big) + O\Big(\frac{\kappa \sigma^2}{\rho_y \eta^2 TS}\Big) + O\Big(\frac{\kappa \sigma^2}{\alpha \eta^2 TS}\Big)$$
$$+ O\Big(\kappa \frac{\alpha^2 \eta^2 \sigma^2}{\rho_x}\Big) + O\Big(\kappa \rho_x \eta^2 \sigma^2\Big) + O\Big(\kappa \frac{\alpha^2 \eta^2 \sigma^2}{\rho_y}\Big) + O\Big(\kappa \rho_y \eta^2 \sigma^2\Big) + O\Big(\kappa \alpha \eta^2 \sigma^2\Big) . \quad (99)$$

$\square$

## C Appendix: Stagewise-SMCGDA-VR

Note that in this section, we provide a general algorithm for the multi-level compositional minimax optimization problem satisfying the two-sided PL condition. Specifically, we aim to solve the following problem:

$$\min_{x \in \mathbb{R}^{d_x}} \max_{y \in \mathbb{R}^{d_y}} f(G(x), y) , \qquad (100)$$

where $f(\cdot, \cdot)$ satisfies the two-sided PL condition. To simplify the analysis, we further assume that the loss function satisfies the same continuity, smoothness, and bounded variance assumptions as stated in the main text.

In this general setting, we can obtain two results which may be of independent interest beyond their use for the translation phase.

First, when the number of stages is just one, i.e., $R = 1$, we can obtain the convergence rate for the multi-level compositional minimax optimization problem satisfying the nonconvex-PL assumption in Theorem C.1.

**Theorem C.1.** *Given Assumption 3.1-3.3, when $R = 1$, by setting $\eta_{y,1} = O(\epsilon/\kappa)$, $\eta_{x,1} = O(\epsilon/\kappa^3)$, and the initial batch size as $O(\kappa/\epsilon)$, after running Algorithm 3 for $T_1 = O(\kappa^3/\epsilon^3)$ total iterations, we have $\frac{1}{T_1} \sum_{t=0}^{T_1-1} \mathbb{E}[\|\nabla \Phi(x_{1,t})\|^2] \leq \epsilon^2$.*

Note that in the proof of Theorem C.1, we do not use the PL condition with respect to $x$. Therefore, the result provides a convergence rate for the nonconvex-PL minimax problem. In addition, this convergence rate corresponds to the standard compositional minimax algorithm without the use of the smoothing technique. Therefore, in Table C, we compare the convergence rate and learning rate with and without the use of the smoothing technique. It can be seen that we should use a smaller learning rate for $x$ compared to $y$ when not using the smoothing technique, as the condition number $\kappa > 1$.

Table 3: The comparison of the convergence rate and learning rate with and without the use of the smoothing technique. **LR-$x$** denotes the learning rate for $x$, and **LR-$y$** denotes that for $y$.

| Algorithms | Convergence Rate | LR-$x$ | LR-$y$ | LR-$x$/LR-$y$ |
|---|---|---|---|---|
| Smoothed-SMCGDA-VR (Thm. 4.1) | $O(\kappa^{3/2}/\epsilon^3)$ | $O(\epsilon/\kappa^{1/2})$ | $O(\epsilon/\kappa^{1/2})$ | $O(1)$ |
| Onestage-SMCGDA-VR (Thm. C.1) | $O(\kappa^3/\epsilon^3)$ | $O(\epsilon/\kappa^3)$ | $O(\epsilon/\kappa)$ | $O(1/\kappa^2)$ |

Second, when the number of stages is greater than one, i.e., $R > 1$, we can obtain the convergence rate for the multi-level compositional minimax optimization problem satisfying the two-sided PL condition in Theorem C.2.

**Theorem C.2.** *Given Assumption 3.1-3.4, by setting $c_0 = \frac{25L_f^2}{\mu^2}$, $\rho_x = 6400c_0 L_\beta^2$, $\rho_y = 640L_\beta^2$, $\alpha = 640c_0 L_\beta^2$, $\eta_{y,0} = \frac{1}{20L_\beta}$, $T_0 = \max\{225, \frac{16\mathcal{V}_0}{L_\beta \sigma^2}\}$, and for $r \geq 1$, $\eta_{x,r} = O(\mu^2/(\sqrt{2^{r-1}}L_\beta))$, $\eta_{y,r} = O(1/(\sqrt{2^{r-1}}L_\beta))$, $T_r = O(c_0/(\mu \times 2^{r-1}))$, after running Algorithm 3 for $O(\kappa^6/\epsilon)$ total iterations, we can get $\mathbb{E}[\Phi(\tilde{x}_R) - \Phi(x^*)] \leq \epsilon$.*

### C.1 Useful Lemmas

**Lemma C.3.** *Given Assumptions 3.1-3.3 and $\eta_{x,r} \leq \frac{1}{2L_\Phi}$, we know*

$$\mathbb{E}[\Phi(x_{r,t+1})] \leq \mathbb{E}[\Phi(x_{r,t})] - \frac{\eta_{x,r}}{2} \mathbb{E}[\|\nabla \Phi(x_{r,t})\|^2] - \frac{\eta_{x,r}}{4} \mathbb{E}[\|p_{r,t}\|^2]$$

**Algorithm 3** Stagewise Stochastic Multi-level Compositional Gradient Descent Ascent with Variance Reduced Algorithm (Stagewise-SMCGDA-VR)

---

**Input:** $\rho_x > 0$, $\rho_y > 0$, $\alpha > 0$, $\eta_{x,r} > 0$, $\eta_{y,r} > 0$.

$\tilde{h}_0^{(0)} = x_0$ and $\tilde{h}_0^{(k)} = g^{(k)}(\tilde{h}_0^{(k-1)}; \xi_{0,0}^{(k)})$, for $k \in \{1, \cdots, K\}$.

$\tilde{p}_0 = \nabla g^{(1)}(x_0; \xi_{r,0}^{(1)}) \cdots \nabla g^{(K)}(\tilde{h}_0^{(K-1)}; \xi_{r,0}^{(K)}) \nabla_1 f(\tilde{h}_0^{(K)}, y_0; \zeta_{0,0})$,

$\tilde{q}_0 = \nabla_2 f(\tilde{h}_0^{(K)}, y_0; \zeta_{0,0})$,

1: **for** $r = 0, \cdots, R - 1$ **do**
2:    $x_{r,0} = \tilde{x}_r$, $y_{r,0} = \tilde{y}_r$, $h_{r,0}^{(k)} = \tilde{h}_r^{(k)}$ for $k \in \{0, \cdots, K - 1\}$,
   $p_{r,0} = \tilde{p}_r$, $q_{r,0} = \tilde{q}_r$.
3:    **for** $t = 0, \cdots, T_r - 1$, **do**
4:       $x_{r,t+1} = x_{r,t} - \eta_{x,r} p_{r,t}$,
      $y_{r,t+1} = y_{r,t} + \eta_{y,r} q_{r,t}$.
5:       $h_{r,t+1}^{(0)} = x_{r,t+1}$,
6:       **for** $k = 1, \cdots, K$ **do**
7:          Compute $k$-th inner-level function:
         $h_{r,t+1}^{(k)} = g^{(k)}(h_{r,t+1}^{(k-1)}; \xi_{r,t+1}^{(k)}) + (1 - \alpha \eta_{x,r}^2)(h_{r,t}^{(k)} - g^{(k)}(h_{r,t}^{(k-1)}; \xi_{r,t+1}^{(k)}))$,
8:       **end for**
9:       Compute stochastic compositional gradient $u_{r,t+1}$ and $v_{r,t+1}$:
      $u_{r,t+1;t+1} = \nabla g^{(1)}(h_{r,t+1}^{(0)}; \xi_{r,t+1}^{(1)}) \cdots \nabla g^{(K-1)}(h_{r,t+1}^{(K-2)}; \xi_{r,t+1}^{(K-1)}) \nabla g^{(K)}(h_{r,t+1}^{(K-1)}; \xi_{r,t+1}^{(K)}) \times$
      $\nabla_1 f(h_{r,t+1}^{(K)}, y_{r,t+1}; \zeta_{r,t+1})$ ,
      $v_{r,t+1;t+1} = \nabla_2 f(h_{r,t+1}^{(K)}, y_{r,t+1}; \zeta_{r,t})$ ,
10:      Compute variance-reduced gradient $p_{r,t+1}$ and $q_{r,t+1}$:
      $p_{r,t+1} = u_{r,t+1;t+1} + (1 - \rho_x \eta_{x,r}^2)(p_{r,t} - u_{r,t;t+1})$,
      $q_{r,t+1} = v_{r,t+1;t+1} + (1 - \rho_y \eta_{y,r}^2)(q_{r,t} - v_{r,t;t+1})$,
11:    **end for**
12:    Randomly select $(\tilde{x}_{r+1}, \tilde{y}_{r+1}, \tilde{h}_{r+1}^{(k)}, \tilde{p}_{r+1}, \tilde{q}_{r+1})$ from $\{(x_{r,t}, y_{r,t}, h_{r,t}^{(k)}, p_{r,t}, q_{r,t})\}_{t=0}^{T_r - 1}$.
13: **end for**

---

$$+ \eta_{x,r} \mathbb{E}[\|\nabla \Phi(x_{r,t}) - \nabla_x f(G(x_{r,t}), y_{r,t})\|^2] + 2\eta_{x,r} K \sum_{k=1}^{K} A_k \mathbb{E}[\|g^{(k)}(h_{r,t}^{(k-1)}) - h_{r,t}^{(k)}\|^2]$$

$$+ 2\eta_{x,r} \mathbb{E}[\|\nabla_x f(H(x_{r,t}), y_{r,t}) - p_{r,t}\|^2] \, . \tag{101}$$

*Proof.*

$$\mathbb{E}[\Phi(x_{r,t+1})] \leq \mathbb{E}[\Phi(x_{r,t})] + \mathbb{E}[\langle \nabla \Phi(x_{r,t}), x_{r,t+1} - x_{r,t} \rangle] + \frac{L_\Phi}{2} \mathbb{E}[\|x_{r,t+1} - x_{r,t}\|^2]$$

$$= \mathbb{E}[\Phi(x_{r,t})] - \eta_{x,r} \mathbb{E}[\langle \nabla \Phi(x_{r,t}), p_{r,t} \rangle] + \frac{\eta_{x,r}^2 L_\Phi}{2} \mathbb{E}[\|p_{r,t}\|^2]$$

$$= \mathbb{E}[\Phi(x_{r,t})] - \frac{\eta_{x,r}}{2} \mathbb{E}[\|\nabla \Phi(x_{r,t})\|^2] - \frac{\eta_{x,r}}{2} \mathbb{E}[\|p_{r,t}\|^2] + \frac{\eta_{x,r}}{2} \mathbb{E}[\|\nabla \Phi(x_{r,t}) - p_{r,t}\|^2] + \frac{\eta_{x,r}^2 L_\Phi}{2} \mathbb{E}[\|p_{r,t}\|^2]$$

$$\leq \mathbb{E}[\Phi(x_{r,t})] - \frac{\eta_{x,r}}{2} \mathbb{E}[\|\nabla \Phi(x_{r,t})\|^2] + (\frac{\eta_{x,r}^2 L_\Phi}{2} - \frac{\eta_{x,r}}{2}) \mathbb{E}[\|p_{r,t}\|^2]$$

$$+ \eta_{x,r} \mathbb{E}[\|\nabla \Phi(x_{r,t}) - \nabla_x f(G(x_{r,t}), y_{r,t})\|^2] + \eta_{x,r} \mathbb{E}[\|\nabla_x f(G(x_{r,t}), y_{r,t}) - p_{r,t}\|^2]$$

$$\leq \mathbb{E}[\Phi(x_{r,t})] - \frac{\eta_{x,r}}{2} \mathbb{E}[\|\nabla \Phi(x_{r,t})\|^2] - \frac{\eta_{x,r}}{4} \mathbb{E}[\|p_{r,t}\|^2] + \eta_{x,r} \mathbb{E}[\|\nabla \Phi(x_{r,t}) - \nabla_x f(G(x_{r,t}), y_{r,t})\|^2]$$

$$+ 2\eta_{x,r} \mathbb{E}[\|\nabla_x f(G(x_{r,t}), y_{r,t}) - \nabla_x f(H(x_{r,t}), y_{r,t})\|^2] + 2\eta_{x,r} \mathbb{E}[\|\nabla_x f(H(x_{r,t}), y_{r,t}) - p_{r,t}\|^2]$$

$$\leq \mathbb{E}[\Phi(x_{r,t})] - \frac{\eta_{x,r}}{2} \mathbb{E}[\|\nabla \Phi(x_{r,t})\|^2] - \frac{\eta_{x,r}}{4} \mathbb{E}[\|p_{r,t}\|^2] + \eta_{x,r} \mathbb{E}[\|\nabla \Phi(x_{r,t}) - \nabla_x f(G(x_{r,t}), y_{r,t})\|^2]$$

$$+ 2\eta_{x,r} K \sum_{k=1}^{K} A_k \mathbb{E}[\|g^{(k)}(h_{r,t}^{(k-1)}) - h_{r,t}^{(k)}\|^2] + 2\eta_{x,r} \mathbb{E}[\|\nabla_x f(H(x_{r,t}), y_{r,t}) - p_{r,t}\|^2] \, , \tag{102}$$

where the second-to-last step holds due to $\eta_{x,r} \leq \frac{1}{2L_\Phi}$. $\qquad\qquad\square$

**Lemma C.4.** *Given Assumption 3.1-3.3 , $\eta_{y,r} \leq \frac{1}{\ell}$, we have*

$$\mathbb{E}[f(G(x_{r,t}), y_{r,t})]$$

$$\leq \mathbb{E}[f(G(x_{r,t+1}), y_{r,t+1})] + \frac{3\eta_{x,r}}{2}\mathbb{E}[\|\nabla_x f(G(x_{r,t}), y_{r,t})\|^2] - \frac{\eta_{y,r}}{4}\mathbb{E}[\|\nabla_y f(G(x_{r,t}), y_{r,t})\|^2]$$

$$+ \eta_{x,r}K\sum_{k=1}^{K} A_k \mathbb{E}[\|g^{(k)}(h_{r,t}^{(k-1)}) - h_{r,t}^{(k)}\|^2] + 2\eta_{y,r}L_f^2 K \sum_{k=1}^{K} C_g^{2(K-k)} \mathbb{E}[\|g^{(k)}(h_{r,t}^{(k-1)}) - h_{r,t}^{(k)}\|^2]$$

$$+ \eta_{x,r}\mathbb{E}[\|\nabla_x f(H(x_{r,t}), y_{r,t}) - p_{r,t}\|^2] + 2\eta_{y,r}\mathbb{E}[\|\nabla_y f(H(x_{r,t}), y_{r,t}) - q_{r,t}\|^2]$$

$$+ 2\eta_{x,r}^2\ell\mathbb{E}[\|p_{r,t}\|^2] - \frac{\eta_{y,r}}{4}\mathbb{E}[\|q_{r,t}\|^2] . \tag{103}$$

*Proof.* First, from Lemma B.1, we obtain

$$\mathbb{E}[f(G(x_{r,t}), y_{r,t})]$$

$$\leq \mathbb{E}[f(G(x_{r,t+1}), y_{r,t})] - \mathbb{E}[\langle \nabla_x f(G(x_{r,t}), y_{r,t}), x_{r,t+1} - x_{r,t}\rangle] + \frac{\ell}{2}\mathbb{E}[\|x_{r,t+1} - x_{r,t}\|^2]$$

$$= \mathbb{E}[f(G(x_{r,t+1}), y_{r,t})] + \eta_{x,r}\mathbb{E}[\langle \nabla_x f(G(x_{r,t}), y_{r,t}), p_{r,t}\rangle] + \frac{\eta_{x,r}^2\ell}{2}\mathbb{E}[\|p_{r,t}\|^2]$$

$$= \mathbb{E}[f(G(x_{r,t+1}), y_{r,t})] + \eta_{x,r}\mathbb{E}[\|\nabla_x f(G(x_{r,t}), y_{r,t})\|^2]$$

$$\quad + \eta_{x,r}\mathbb{E}[\langle \nabla_x f(G(x_{r,t}), y_{r,t}), p_{r,t} - \nabla_x f(G(x_{r,t}), y_{r,t})\rangle] + \frac{\eta_{x,r}^2\ell}{2}\mathbb{E}[\|p_{r,t}\|^2]$$

$$\leq \mathbb{E}[f(G(x_{r,t+1}), y_{r,t})] + \frac{3\eta_{x,r}}{2}\mathbb{E}[\|\nabla_x f(G(x_{r,t}), y_{r,t})\|^2] + \frac{\eta_{x,r}}{2}\mathbb{E}[\|\nabla_x f(G(x_{r,t}), y_{r,t}) - p_{r,t}\|^2]$$

$$\quad + \frac{\eta_{x,r}^2\ell}{2}\mathbb{E}[\|p_{r,t}\|^2]$$

$$\leq \mathbb{E}[f(G(x_{r,t+1}), y_{r,t})] + \frac{3\eta_{x,r}}{2}\mathbb{E}[\|\nabla_x f(G(x_{r,t}), y_{r,t})\|^2] + \frac{\eta_{x,r}^2\ell}{2}\mathbb{E}[\|p_{r,t}\|^2]$$

$$\quad + \eta_{x,r}\mathbb{E}[\|\nabla_x f(G(x_{r,t}), y_{r,t}) - \nabla_x f(H(x_{r,t}), y_{r,t})\|^2] + \eta_{x,r}\mathbb{E}[\|\nabla_x f(H(x_{r,t}), y_{r,t}) - p_{r,t}\|^2]$$

$$\leq \mathbb{E}[f(G(x_{r,t+1}), y_{r,t})] + \frac{3\eta_{x,r}}{2}\mathbb{E}[\|\nabla_x f(G(x_{r,t}), y_{r,t})\|^2] + \frac{\eta_{x,r}^2\ell}{2}\mathbb{E}[\|p_{r,t}\|^2]$$

$$\quad + \eta_{x,r}K\sum_{k=1}^{K} A_k \mathbb{E}[\|g^{(k)}(h_{r,t}^{(k-1)}) - h_{r,t}^{(k)}\|^2] + \eta_{x,r}\mathbb{E}[\|\nabla_x f(H(x_{r,t}), y_{r,t}) - p_{r,t}\|^2] . \tag{104}$$

Moreover, we obtain

$$\mathbb{E}[f(G(x_{r,t+1}), y_{r,t})]$$

$$\leq \mathbb{E}[f(G(x_{r,t+1}), y_{r,t+1})] - \mathbb{E}[\langle \nabla_y f(G(x_{r,t+1}), y_{r,t}), y_{r,t+1} - y_{r,t}\rangle] + \frac{\ell}{2}\mathbb{E}[\|y_{r,t+1} - y_{r,t}\|^2]$$

$$= \mathbb{E}[f(G(x_{r,t+1}), y_{r,t+1})] - \eta_{y,r}\mathbb{E}[\langle \nabla_y f(G(x_{r,t+1}), y_{r,t}), q_{r,t}\rangle] + \frac{\eta_{y,r}^2\ell}{2}\mathbb{E}[\|q_{r,t}\|^2]$$

$$\leq \mathbb{E}[f(G(x_{r,t+1}), y_{r,t+1})] - \frac{\eta_{y,r}}{2}\mathbb{E}[\|\nabla_y f(G(x_{r,t+1}), y_{r,t})\|^2] + \frac{\eta_{y,r}}{2}\mathbb{E}[\|\nabla_y f(G(x_{r,t+1}), y_{r,t}) - q_{r,t}\|^2]$$

$$\quad + (\frac{\eta_{y,r}^2\ell}{2} - \frac{\eta_{y,r}}{2})\mathbb{E}[\|q_{r,t}\|^2]$$

$$\leq \mathbb{E}[f(G(x_{r,t+1}), y_{r,t+1})] - \frac{\eta_{y,r}}{2}\mathbb{E}[\|\nabla_y f(G(x_{r,t+1}), y_{r,t})\|^2] + (\frac{\eta_{y,r}^2\ell}{2} - \frac{\eta_{y,r}}{2})\mathbb{E}[\|q_{r,t}\|^2]$$

$$\quad + \eta_{y,r}\mathbb{E}[\|\nabla_y f(G(x_{r,t+1}), y_{r,t}) - \nabla_y f(G(x_{r,t}), y_{r,t})\|^2] + \eta_{y,r}\mathbb{E}[\|\nabla_y f(G(x_{r,t}), y_{r,t}) - q_{r,t}\|^2]$$

$$\leq \mathbb{E}[f(G(x_{r,t+1}), y_{r,t+1})] - \frac{\eta_{y,r}}{4}\mathbb{E}[\|\nabla_y f(G(x_{r,t}), y_{r,t})\|^2] + (\frac{\eta_{y,r}^2\ell}{2} - \frac{\eta_{y,r}}{2})\mathbb{E}[\|q_{r,t}\|^2]$$

$$+ \frac{3}{2}\eta_{y,r}\mathbb{E}[\|\nabla_y f(G(x_{r,t+1}), y_{r,t}) - \nabla_y f(G(x_{r,t}), y_{r,t})\|^2] + \eta_{y,r}\mathbb{E}[\|\nabla_y f(G(x_{r,t}), y_{r,t}) - q_{r,t}\|^2]$$

$$\leq \mathbb{E}[f(G(x_{r,t+1}), y_{r,t+1})] - \frac{\eta_{y,r}}{4}\mathbb{E}[\|\nabla_y f(G(x_{r,t}), y_{r,t})\|^2] + (\frac{\eta_{y,r}^2\ell}{2} - \frac{\eta_{y,r}}{2})\mathbb{E}[\|q_{r,t}\|^2]$$

$$+ \frac{3}{2}\eta_{y,r}\ell^2\mathbb{E}[\|x_{r,t+1} - x_{r,t}\|^2] + \eta_{y,r}\mathbb{E}[\|\nabla_y f(G(x_{r,t}), y_{r,t}) - q_{r,t}\|^2]$$

$$\leq \mathbb{E}[f(G(x_{r,t+1}), y_{r,t+1})] - \frac{\eta_{y,r}}{4}\mathbb{E}[\|\nabla_y f(G(x_{r,t}), y_{r,t})\|^2] - \frac{\eta_{y,r}}{4}\mathbb{E}[\|q_{r,t}\|^2] + \frac{3}{2}\eta_{x,r}^2\ell\mathbb{E}[\|p_{r,t}\|^2]$$

$$+ 2\eta_{y,r}\mathbb{E}[\|\nabla_y f(G(x_{r,t}), y_{r,t}) - \nabla_y f(H(x_{r,t}), y_{r,t})\|^2] + 2\eta_{y,r}\mathbb{E}[\|\nabla_y f(H(x_{r,t}), y_{r,t}) - q_{r,t}\|^2]$$

$$\leq \mathbb{E}[f(G(x_{r,t+1}), y_{r,t+1})] - \frac{\eta_{y,r}}{4}\mathbb{E}[\|\nabla_y f(G(x_{r,t}), y_{r,t})\|^2] - \frac{\eta_{y,r}}{4}\mathbb{E}[\|q_{r,t}\|^2] + \frac{3}{2}\eta_{x,r}^2\ell\mathbb{E}[\|p_{r,t}\|^2]$$

$$+ 2\eta_{y,r}L_f^2\mathbb{E}[\|G(x_{r,t}) - h_{r,t}^{(K)}\|^2] + 2\eta_{y,r}\mathbb{E}[\|\nabla_y f(H(x_{r,t}), y_{r,t}) - q_{r,t}\|^2]$$

$$\leq \mathbb{E}[f(G(x_{r,t+1}), y_{r,t+1})] - \frac{\eta_{y,r}}{4}\mathbb{E}[\|\nabla_y f(G(x_{r,t}), y_{r,t})\|^2] - \frac{\eta_{y,r}}{4}\mathbb{E}[\|q_{r,t}\|^2] + \frac{3}{2}\eta_{x,r}^2\ell\mathbb{E}[\|p_{r,t}\|^2]$$

$$+ 2\eta_{y,r}L_f^2 K \sum_{k=1}^{K} C_g^{2(K-k)}\mathbb{E}[\|g^{(k)}(h_{r,t}^{(k-1)}) - h_{r,t}^{(k)}\|^2] + 2\eta_{y,r}\mathbb{E}[\|\nabla_y f(H(x_{r,t}), y_{r,t}) - q_{r,t}\|^2] \,,$$

$$\tag{105}$$

where the sixth step holds due to $\eta_{y,r} \leq \frac{1}{\ell}$, the fourth step follows from the following inequality:

$$- \|\nabla_y f(G(x_{r,t+1}), y_{r,t})\|^2$$
$$\leq -\frac{1}{2}\|\nabla_y f(G(x_{r,t}), y_{r,t})\|^2 + \|\nabla_y f(G(x_{r,t+1}), y_{r,t}) - \nabla_y f(G(x_{r,t}), y_{r,t})\|^2 \,. \tag{106}$$

By combining these two inequalities, the proof is complete. $\qquad\square$

**Lemma C.5.** *Given Assumption 3.1-3.3 , $\eta_{x,r} \leq \frac{1}{16\ell}$, we have*

$$\mathbb{E}[\Phi(x_{r,t+1}) - f(G(x_{r,t+1}), y_{r,t+1})] - \mathbb{E}[\Phi(x_{r,t}) - f(G(x_{r,t}), y_{r,t})]$$

$$\leq \frac{5\eta_{x,r}}{2}\mathbb{E}[\|\nabla\Phi(x_{r,t})\|^2] - \frac{\eta_{y,r}}{4}\mathbb{E}[\|\nabla_y f(G(x_{r,t}), y_{r,t})\|^2] + 4\eta_{x,r}\mathbb{E}[\|\nabla\Phi(x_{r,t}) - \nabla_x f(G(x_{r,t}), y_{r,t})\|^2]$$

$$+ 3\eta_{x,r}K\sum_{k=1}^{K} A_k\mathbb{E}[\|g^{(k)}(h_{r,t}^{(k-1)}) - h_{r,t}^{(k)}\|^2] + 2\eta_{y,r}L_f^2 K\sum_{k=1}^{K} C_g^{2(K-k)}\mathbb{E}[\|g^{(k)}(h_{r,t}^{(k-1)}) - h_{r,t}^{(k)}\|^2]$$

$$+ 3\eta_{x,r}\mathbb{E}[\|\nabla_x f(H(x_{r,t}), y_{r,t}) - p_{r,t}\|^2] + 2\eta_{y,r}\mathbb{E}[\|\nabla_y f(H(x_{r,t}), y_{r,t}) - q_{r,t}\|^2]$$

$$- \frac{\eta_{x,r}}{8}\mathbb{E}[\|p_{r,t}\|^2] - \frac{\eta_{y,r}}{4}\mathbb{E}[\|q_{r,t}\|^2] \,. \tag{107}$$

*Proof.* In terms of Lemma C.3 and Lemma C.4, we obtain

$$\mathbb{E}[\Phi(x_{r,t+1}) - f(G(x_{r,t+1}), y_{r,t+1})] - \mathbb{E}[\Phi(x_{r,t}) - f(G(x_{r,t}), y_{r,t})]$$

$$\leq -\frac{\eta_{x,r}}{2}\mathbb{E}[\|\nabla\Phi(x_{r,t})\|^2] - \frac{\eta_{x,r}}{4}\mathbb{E}[\|p_{r,t}\|^2] + \eta_{x,r}\mathbb{E}[\|\nabla\Phi(x_{r,t}) - \nabla_x f(G(x_{r,t}), y_{r,t})\|^2]$$

$$+ 2\eta_{x,r}K\sum_{k=1}^{K} A_k\mathbb{E}[\|g^{(k)}(h_{r,t}^{(k-1)}) - h_{r,t}^{(k)}\|^2] + 2\eta_{x,r}\mathbb{E}[\|\nabla_x f(H(x_{r,t}), y_{r,t}) - p_{r,t}\|^2]$$

$$+ \frac{3\eta_{x,r}}{2}\mathbb{E}[\|\nabla_x f(G(x_{r,t}), y_{r,t})\|^2] - \frac{\eta_{y,r}}{4}\mathbb{E}[\|\nabla_y f(G(x_{r,t}), y_{r,t})\|^2]$$

$$+ \eta_{x,r}K\sum_{k=1}^{K} A_k\mathbb{E}[\|g^{(k)}(h_{r,t}^{(k-1)}) - h_{r,t}^{(k)}\|^2] + 2\eta_{y,r}L_f^2 K\sum_{k=1}^{K} C_g^{2(K-k)}\mathbb{E}[\|g^{(k)}(h_{r,t}^{(k-1)}) - h_{r,t}^{(k)}\|^2]$$

$$+ \eta_{x,r}\mathbb{E}[\|\nabla_x f(H(x_{r,t}), y_{r,t}) - p_{r,t}\|^2] + 2\eta_{y,r}\mathbb{E}[\|\nabla_y f(H(x_{r,t}), y_{r,t}) - q_{r,t}\|^2]$$

$$+ 2\eta_{x,r}^2\ell\mathbb{E}[\|p_{r,t}\|^2] - \frac{\eta_{y,r}}{4}\mathbb{E}[\|q_{r,t}\|^2]$$

$$\leq -\frac{\eta_{x,r}}{2}\mathbb{E}[\|\nabla\Phi(x_{r,t})\|^2] - \frac{\eta_{x,r}}{4}\mathbb{E}[\|p_{r,t}\|^2] + \eta_{x,r}\mathbb{E}[\|\nabla\Phi(x_{r,t}) - \nabla_x f(G(x_{r,t}), y_{r,t})\|^2]$$

$$+ 3\eta_{x,r}K\sum_{k=1}^{K}A_k\mathbb{E}[\|g^{(k)}(h_{r,t}^{(k-1)}) - h_{r,t}^{(k)}\|^2] + 2\eta_{y,r}L_f^2K\sum_{k=1}^{K}C_g^{2(K-k)}\mathbb{E}[\|g^{(k)}(h_{r,t}^{(k-1)}) - h_{r,t}^{(k)}\|^2]$$

$$+ \frac{3\eta_{x,r}}{2}\mathbb{E}[\|\nabla_x f(G(x_{r,t}), y_{r,t})\|^2] - \frac{\eta_{y,r}}{4}\mathbb{E}[\|\nabla_y f(G(x_{r,t}), y_{r,t})\|^2]$$

$$+ 3\eta_{x,r}\mathbb{E}[\|\nabla_x f(H(x_{r,t}), y_{r,t}) - p_{r,t}\|^2] + 2\eta_{y,r}\mathbb{E}[\|\nabla_y f(H(x_{r,t}), y_{r,t}) - q_{r,t}\|^2]$$

$$+ 2\eta_{x,r}^2\ell\mathbb{E}[\|p_{r,t}\|^2] - \frac{\eta_{y,r}}{4}\mathbb{E}[\|q_{r,t}\|^2]$$

$$\leq \frac{5\eta_{x,r}}{2}\mathbb{E}[\|\nabla\Phi(x_{r,t})\|^2] - \frac{\eta_{y,r}}{4}\mathbb{E}[\|\nabla_y f(G(x_{r,t}), y_{r,t})\|^2] + 4\eta_{x,r}\mathbb{E}[\|\nabla\Phi(x_{r,t}) - \nabla_x f(G(x_{r,t}), y_{r,t})\|^2]$$

$$+ 3\eta_{x,r}K\sum_{k=1}^{K}A_k\mathbb{E}[\|g^{(k)}(h_{r,t}^{(k-1)}) - h_{r,t}^{(k)}\|^2] + 2\eta_{y,r}L_f^2K\sum_{k=1}^{K}C_g^{2(K-k)}\mathbb{E}[\|g^{(k)}(h_{r,t}^{(k-1)}) - h_{r,t}^{(k)}\|^2]$$

$$+ 3\eta_{x,r}\mathbb{E}[\|\nabla_x f(H(x_{r,t}), y_{r,t}) - p_{r,t}\|^2] + 2\eta_{y,r}\mathbb{E}[\|\nabla_y f(H(x_{r,t}), y_{r,t}) - q_{r,t}\|^2]$$

$$- \frac{\eta_{x,r}}{8}\mathbb{E}[\|p_{r,t}\|^2] - \frac{\eta_{y,r}}{4}\mathbb{E}[\|q_{r,t}\|^2] \,, \tag{108}$$

where the last step follows from $\|\nabla_x f(G(x_{r,t}), y_{r,t})\|^2 \leq 2\|\nabla\Phi(x_{r,t})\|^2 + 2\|\nabla_x f(G(x_{r,t}), y_{r,t}) - \nabla\Phi(x_{r,t})\|^2$ and $\eta_{x,r} \leq \frac{1}{16\ell}$. $\qquad\square$

**Lemma C.6.** *Given Assumption 3.1-3.3, by setting*

$$\eta_{x,r} \leq \min\left\{\frac{1}{2L_\Phi}, \frac{1}{16\ell}, \frac{1}{2}\sqrt{\frac{\tilde{\lambda}_k}{\alpha(\sum_{j=k+1}^{K}\tilde{\lambda}_j(2C_g^2)^{j-k})}}\right\}, \eta_{y,r} \leq \min\left\{\frac{1}{2\ell}\right\},$$

$$\rho_y = 640L_\beta^2\,, \rho_x = 6400c_0L_\beta^2, \alpha = 640c_0L_\beta^2\,, c_0 = \frac{25\ell^2}{\mu^2}\,. \tag{109}$$

*where $\tilde{\lambda}_k$ is defined in Eq. (116), $L_\beta$ is defined in Eq. (132), such that $\eta_{x,r} = \frac{\eta_{y,r}}{10c_0}$, we have*

$$\frac{1}{T_r}\sum_{t=0}^{T_r-1}\left(\mathbb{E}[\|\nabla\Phi(x_{r,t})\|^2] + \frac{c_0\eta_{x,r}}{\eta_{y,r}}\mathbb{E}[\|\nabla_y f(G(x_{r,t}), y_{r,t})\|^2]\right)$$

$$\leq \frac{40c_0\mathcal{V}_{r,0}}{\eta_{y,r}T_r} + \frac{160c_0}{\rho_y\eta_{y,r}^2T_r}(\sigma_{r,0}^x + \sigma_{r,0}^y + 56\sigma_{r,0}^h) + 330c_0L_\beta^2\rho_y\eta_{y,r}^2\sigma^2\,. \tag{110}$$

*Proof.* We first propose a novel Lyapunov function as follows:

$$\mathcal{H}_{r,t+1} = \mathbb{E}[\Phi(x_{r,t+1})] - \Phi(x_*) + \frac{c_0\eta_{x,r}}{\eta_{y,r}}(\mathbb{E}[\Phi(x_{r,t+1})] - \mathbb{E}[f(G(x_{r,t+1}), y_{r,t+1})])$$

$$+ \frac{4}{\rho_x\eta_{x,r}}\mathbb{E}[\|\nabla_x f(H(x_{r,t+1}), y_{r,t+1}) - p_{r,t+1}\|^2] + \frac{4}{\rho_y\eta_{y,r}}\mathbb{E}[\|\nabla_y f(H(x_{r,t+1}), y_{r,t+1}) - q_{r,t+1}\|^2]$$

$$+ \sum_{k=1}^{K}\lambda_k\mathbb{E}[\|g^{(k)}(h_{r,t+1}^{(k-1)}) - h_{r,t+1}^{(k)}\|^2]\,, \tag{111}$$

where $\eta_{x,r} = \frac{\eta_{y,r}}{10c_0}$. Then, from Lemma C.5, B.5, B.6 and B.7, we obtain

$$\mathcal{H}_{r,t+1} - \mathcal{H}_{r,t} \leq -\frac{\eta_{x,r}}{4}\mathbb{E}[\|\nabla\Phi(x_{r,t})\|^2] - \frac{c_0\eta_{x,r}}{4}\mathbb{E}[\|\nabla_y f(G(x_{r,t}), y_{r,t})\|^2]$$

$$+ \left(\eta_{x,r} + \frac{4c_0\eta_{x,r}^2}{\eta_{y,r}}\right)\mathbb{E}[\|\nabla\Phi(x_{r,t}) - \nabla_x f(G(x_{r,t}), y_{r,t})\|^2]$$

$$+ \left(2\eta_{x,r} + \frac{3c_0\eta_{x,r}^2}{\eta_{y,r}} - 4\eta_{x,r}\right)\mathbb{E}[\|\nabla_x f(H(x_{r,t}), y_{r,t}) - p_{r,t}\|^2]$$

$$+ (2c_0\eta_{x,r} - 4\eta_{y,r})\mathbb{E}[\|\nabla_y f(H(x_{r,t}), y_{r,t}) - q_{r,t}\|^2]$$

$$+ \left(\eta_{x,r}^2\sum_{k=1}^{K}\lambda_k(2C_g^2)^k + \frac{8\eta_{x,r}}{\rho_x}C_p^2\sum_{k=0}^{K}(2C_g^2)^k + 2L_f^2(2C_g^2)^K\frac{4\eta_{x,r}^2}{\rho_y\eta_{y,r}} - \frac{\eta_{x,r}}{4} - \frac{c_0\eta_{x,r}}{\eta_{y,r}}\frac{\eta_{x,r}}{8}\right)\mathbb{E}[\|p_{r,t}\|^2]$$

$$+ \left( \frac{8\eta_{y,r}^2}{\rho_x \eta_{x,r}} C_p^2 + \frac{8\eta_{y,r}}{\rho_y} L_f^2 - \frac{c_0 \eta_{x,r}}{4} \right) \mathbb{E}[\|q_{r,t}\|^2]$$

$$+ \sum_{k=1}^{K} \left( 2\eta_{x,r} K A_k + \frac{3c_0 \eta_{x,r}^2}{\eta_{y,r}} K A_k + 2c_0 \eta_{x,r} L_f^2 K C_g^{2(K-k)} + \frac{16\alpha^2 \eta_{x,r}^4}{\rho_x \eta_{x,r}} C_p^2 \Big( \sum_{j=k}^{K} (2C_g^2)^{j-k} \Big) \right.$$

$$\left. + \frac{16\alpha^2 \eta_{x,r}^4}{\rho_y \eta_{y,r}} L_f^2 (2C_g^2)^{K-k} + 2\alpha^2 \eta_{x,r}^4 \Big( \sum_{j=k+1}^{K} \lambda_j (2C_g^2)^{j-k} \Big) - \alpha \eta_{x,r}^2 \lambda_k \right) \mathbb{E}[\|h_{r,t}^{(k)} - g^{(k)}(h_{r,t}^{(k-1)})\|^2]$$

$$+ \frac{16\alpha^2 \eta_{x,r}^3}{\rho_x} C_p^2 \sigma^2 \sum_{k=1}^{K} \sum_{j=1}^{k} (2C_g^2)^{j-1} + 8\rho_x \eta_{x,r}^3 C_p^2 \sigma^2 + 2\alpha^2 \eta_{x,r}^4 \sigma^2 \sum_{k=1}^{K} \lambda_k \sum_{j=0}^{k-1} (2C_g^2)^j$$

$$+ \frac{16\alpha^2 \eta_{x,r}^4}{\rho_y \eta_{y,r}} L_f^2 \sigma^2 \sum_{k=1}^{K} (2C_g^2)^{k-1} + 8\rho_y \eta_{y,r}^3 \sigma^2 . \tag{112}$$

To begin with, for any $k \in \{1, \cdots, K\}$, to remove the term $\mathbb{E}[\|h_{r,t}^{(k)} - g^{(k)}(h_{r,t}^{(k-1)})\|^2]$, we set

$$2\eta_{x,r} K A_k + \frac{3c_0 \eta_{x,r}^2}{\eta_{y,r}} K A_k + 2c_0 \eta_{x,r} L_f^2 K C_g^{2(K-k)} + \frac{16\alpha^2 \eta_{x,r}^4}{\rho_x \eta_{x,r}} C_p^2 \Big( \sum_{j=k}^{K} (2C_g^2)^{j-k} \Big)$$

$$+ \frac{16\alpha^2 \eta_{x,r}^4}{\rho_y \eta_{y,r}} L_f^2 (2C_g^2)^{K-k} + 2\alpha^2 \eta_{x,r}^4 \Big( \sum_{j=k+1}^{K} \lambda_j (2C_g^2)^{j-k} \Big) - \alpha \eta_{x,r}^2 \lambda_k \le 0 . \tag{113}$$

Since $\eta_{x,r} = \frac{\eta_{y,r}}{10c_0}$, we enforce

$$3\eta_{x,r} K A_k + 2c_0 \eta_{x,r} L_f^2 K C_g^{2(K-k)} + \frac{16\alpha^2 \eta_{x,r}^4}{\rho_x \eta_{x,r}} C_p^2 \sum_{j=k}^{K} (2C_g^2)^{j-k}$$

$$+ \frac{16\alpha^2 \eta_{x,r}^4}{\rho_y \eta_{y,r}} L_f^2 (2C_g^2)^{K-k} - \frac{1}{2} \alpha \eta_{x,r}^2 \lambda_k \le 0 . \tag{114}$$

By solving this, we obtain

$$\lambda_k = \frac{8K A_k}{\alpha \eta_{x,r}} + \frac{8c_0 L_f^2 K C_g^{2(K-k)}}{\alpha \eta_{x,r}} + \frac{32}{\rho_x \eta_{x,r}} C_p^2 \sum_{j=k}^{K} (2C_g^2)^{j-k} + \frac{32}{\rho_y \eta_{y,r}} L_f^2 (2C_g^2)^{K-k}$$

$$\triangleq \frac{8\lambda_{k,1}}{\alpha \eta_{x,r}} + \frac{8c_0 \lambda_{k,2}}{\alpha \eta_{x,r}} + \frac{32\lambda_{k,3}}{\rho_x \eta_{x,r}} + \frac{32\lambda_{k,4}}{\rho_y \eta_{y,r}} , \tag{115}$$

where $k \in \{1, \cdots, K\}$ and $\lambda_k' = \max\{\lambda_{k,1}, \lambda_{k,2}, \lambda_{k,3}, \lambda_{k,4}\}$. Given that $\lambda_k$ can be organized as:

$$\lambda_k = \frac{1}{\eta_{x,r}} \Big( \frac{8\lambda_{k,1}}{\alpha} + \frac{8c_0 \lambda_{k,2}}{\alpha} + \frac{32\lambda_{k,3}}{\rho_x} + \frac{32\lambda_{k,4}}{10c_0 \rho_y} \Big) \triangleq \frac{1}{\eta_{x,r}} \tilde{\lambda}_k . \tag{116}$$

Moreover, we enforce

$$2\alpha^2 \eta_{x,r}^4 \sum_{j=k+1}^{K} \lambda_j (2C_g^2)^{j-k} - \alpha \eta_{x,r}^2 \lambda_k \le -\frac{1}{2} \alpha \eta_{x,r}^2 \lambda_k , \tag{117}$$

for $k \in \{1, \cdots, K\}$, which leads to

$$\eta_{x,r} \le \frac{1}{2} \sqrt{ \frac{\tilde{\lambda}_k}{\alpha (\sum_{j=k+1}^{K} \tilde{\lambda}_j (2C_g^2)^{j-k})} } . \tag{118}$$

Additionally, by plugging the value of $\lambda_k$, we obtain

$$2\eta_{x,r} K \sum_{k=1}^{K} A_k + \frac{3c_0 \eta_{x,r}^2}{\eta_{y,r}} K \sum_{k=1}^{K} A_k + 2c_0 \eta_{x,r} L_f^2 K \sum_{k=1}^{K} C_g^{2(K-k)} + 2\alpha^2 \eta_{x,r}^4 \sum_{k=1}^{K} \Big( \sum_{j=k+1}^{K} \lambda_j (2C_g^2)^{j-k} \Big)$$

$$+ \frac{16\alpha^2 \eta_{x,r}^4}{\rho_x \eta_{x,r}} C_p^2 \sum_{k=1}^{K} \Big( \sum_{j=k}^{K} (2C_g^2)^{j-k} \Big) + \frac{16\alpha^2 \eta_{x,r}^4}{\rho_y \eta_{y,r}} L_f^2 \sum_{k=1}^{K} (2C_g^2)^{K-k} - \alpha \eta_{x,r}^2 \sum_{k=1}^{K} \lambda_k$$

$$\leq \frac{23}{10} \eta_{x,r} K \sum_{k=1}^{K} A_k + 2c_0 \eta_{x,r} L_f^2 K \sum_{k=1}^{K} C_g^{2(K-k)} + \frac{16\alpha^2 \eta_{x,r}^4}{\rho_x \eta_{x,r}} C_p^2 \sum_{k=1}^{K} \Big( \sum_{j=k}^{K} (2C_g^2)^{j-k} \Big) + \frac{16\alpha^2 \eta_{x,r}^4}{\rho_y \eta_{y,r}} L_f^2 \sum_{k=1}^{K} (2C_g^2)^{K-k}$$

$$- \frac{1}{2} \alpha \eta_{x,r}^2 \sum_{k=1}^{K} \Big( \frac{8KA_k}{\alpha \eta_{x,r}} + \frac{8c_0 L_f^2 K C_g^{2(K-k)}}{\alpha \eta_{x,r}} + \frac{32}{\rho_x \eta_{x,r}} C_p^2 \sum_{j=k}^{K} (2C_g^2)^{j-k} + \frac{32}{\rho_y \eta_{y,r}} L_f^2 (2C_g^2)^{K-k} \Big)$$

$$\leq \frac{23}{10} \eta_{x,r} K \sum_{k=1}^{K} A_k + 2c_0 \eta_{x,r} L_f^2 K \sum_{k=1}^{K} C_g^{2(K-k)} + \frac{16\alpha \eta_{x,r}^2}{\rho_x \eta_{x,r}} C_p^2 \sum_{k=1}^{K} \Big( \sum_{j=k}^{K} (2C_g^2)^{j-k} \Big) + \frac{16\alpha \eta_{x,r}^2}{\rho_y \eta_{y,r}} L_f^2 \sum_{k=1}^{K} (2C_g^2)^{K-k}$$

$$- \sum_{k=1}^{K} \Big( 4KA_k \eta_{x,r} + 4c_0 \eta_{x,r} L_f^2 K C_g^{2(K-k)} + \frac{16\alpha \eta_{x,r}^2}{\rho_x \eta_{x,r}} C_p^2 \sum_{j=k}^{K} (2C_g^2)^{j-k} + \frac{16\alpha \eta_{x,r}^2}{\rho_y \eta_{y,r}} L_f^2 (2C_g^2)^{K-k} \Big)$$

$$= -\frac{17}{10} \eta_{x,r} K \sum_{k=1}^{K} A_k - 2c_0 \eta_{x,r} L_f^2 K \sum_{k=1}^{K} C_g^{2(K-k)} . \tag{119}$$

To guarantee that $\mathbb{E}[\|p_{r,t}\|^2]$ cancels out, we enforce

$$\eta_{x,r}^2 \sum_{k=1}^{K} \lambda_k (2C_g^2)^k + \frac{8\eta_{x,r}}{\rho_x} C_p^2 \sum_{k=0}^{K} (2C_g^2)^k + 2L_f^2 (2C_g^2)^K \frac{4\eta_{x,r}^2}{\rho_y \eta_{y,r}} - \frac{\eta_{x,r}}{4} - \frac{c_0 \eta_{x,r}}{\eta_{y,r}} \frac{\eta_{x,r}}{8} \leq 0 . \tag{120}$$

This can be done by setting

$$\eta_{x,r} \sum_{k=1}^{K} \tilde{\lambda}_k (2C_g^2)^k - \frac{\eta_{x,r}}{16} \leq 0 ,$$

$$\frac{8\eta_{x,r}}{\rho_x} C_p^2 \sum_{k=0}^{K} (2C_g^2)^k - \frac{\eta_{x,r}}{16} \leq 0 ,$$

$$2L_f^2 (2C_g^2)^K \frac{4\eta_{x,r}^2}{\rho_y \eta_{y,r}} - \frac{\eta_{x,r}}{16} \leq 0 . \tag{121}$$

For the first inequality, we enforce

$$\sum_{k=1}^{K} \frac{8\lambda_{k,1}}{\alpha} (2C_g^2)^k - \frac{1}{64} \leq 0 , \quad \sum_{k=1}^{K} \frac{8c_0 \lambda_{k,2}}{\alpha} (2C_g^2)^k - \frac{1}{64} \leq 0 ,$$

$$\sum_{k=1}^{K} \frac{32\lambda_{k,3}}{\rho_x} (2C_g^2)^k - \frac{1}{64} \leq 0 , \quad \sum_{k=1}^{K} \frac{32\lambda_{k,4}}{10c_0 \rho_y} (2C_g^2)^k - \frac{1}{64} \leq 0 . \tag{122}$$

It is easy to obtain

$$\alpha \geq \max\{ \sum_{k=1}^{K} 512\lambda_{k,1} (2C_g^2)^k , \sum_{k=1}^{K} 512c_0 \lambda_{k,2} (2C_g^2)^k \} ,$$

$$\rho_x \geq \sum_{k=1}^{K} 2048\lambda_{k,3} (2C_g^2)^k , \quad \rho_y \geq \sum_{k=1}^{K} \frac{1024\lambda_{k,4}}{5c_0} (2C_g^2)^k . \tag{123}$$

For the second and third inequalities, we obtain

$$\rho_x \geq 128C_p^2 \sum_{k=0}^{K} (2C_g^2)^k , \quad \rho_y \geq \frac{64}{5c_0} L_f^2 (2C_g^2)^K . \tag{124}$$

Similarly, to guarantee that $\mathbb{E}[\|q_{r,t}\|^2]$ cancels out, we enforce

$$\frac{8\eta_{y,r}^2}{\rho_x \eta_{x,r}} C_p^2 + \frac{8\eta_{y,r}}{\rho_y} L_f^2 - \frac{c_0 \eta_{x,r}}{4} \leq 0 . \tag{125}$$

With $\eta_{x,r} = \frac{\eta_{y,r}}{10c_0}$, we obtain

$$\frac{80c_0\eta_{y,r}}{\rho_x}C_p^2 + \frac{8\eta_{y,r}}{\rho_y}L_f^2 - \frac{\eta_{y,r}}{40} \leq 0 \ . \tag{126}$$

To solve this inequality, we enforce

$$\frac{80c_0}{\rho_x}C_p^2 \leq \frac{1}{80} \ , \quad \frac{8}{\rho_y}L_f^2 \leq \frac{1}{80} \ . \tag{127}$$

We obtain

$$\rho_x \geq 6400c_0C_p^2, \quad \rho_y \geq 640L_f^2 \ . \tag{128}$$

In summary, the hyperparameters should be set as follows:

$$\eta_{x,r} \leq \min\left\{\frac{1}{2L_\Phi}, \frac{1}{16\ell}, \frac{1}{2}\sqrt{\frac{\tilde{\lambda}_k}{\alpha(\sum_{j=k+1}^{K}\tilde{\lambda}_j(2C_g^2)^{j-k})}}\right\}, \eta_{y,r} \leq \min\left\{\frac{1}{\ell}\right\},$$

$$\rho_x \geq \max\{\sum_{k=1}^{K}2048\lambda_{k,3}(2C_g^2)^k , 128C_p^2\sum_{k=0}^{K}(2C_g^2)^k , 6400c_0C_p^2\} \ ,$$

$$\rho_y \geq \max\{\sum_{k=1}^{K}\frac{1024\lambda_{k,4}}{5c_0}(2C_g^2)^k , \frac{64}{5c_0}L_f^2(2C_g^2)^K , 640L_f^2\} \ ,$$

$$\alpha \geq \max\{\sum_{k=1}^{K}512\lambda_{k,1}(2C_g^2)^k , \sum_{k=1}^{K}512c_0\lambda_{k,2}(2C_g^2)^k\} \ . \tag{129}$$

Moreover, by setting $c_0 = \frac{25\ell^2}{\mu^2}$, we get

$$\left(\eta_{x,r} + \frac{4c_0\eta_{x,r}^2}{\eta_{y,r}}\right)\frac{\ell^2}{\mu^2} \leq \frac{c_0\eta_{x,r}}{16} \ . \tag{130}$$

Then from Lemma A.7 in [4] , we have $(\eta_{x,r} + \frac{4c_0\eta_{x,r}^2}{\eta_{y,r}})\mathbb{E}[\|\nabla\Phi(x_{r,t}) - \nabla_x f(G(x_{r,t}), y_{r,t})\|^2] \leq \frac{c_0\eta_{x,r}}{16}\mathbb{E}[\|\nabla_y f(G(x_{r,t}), y_{r,t})\|^2]$.

As a result, we have

$$\mathcal{H}_{r,t+1} - \mathcal{H}_{r,t}$$
$$\leq -\frac{\eta_{x,r}}{4}\mathbb{E}[\|\nabla\Phi(x_{r,t})\|^2] - \frac{c_0\eta_{x,r}}{4}\mathbb{E}[\|\nabla_y f(G(x_{r,t}), y_{r,t})\|^2] + \frac{c_0\eta_{x,r}}{16}\mathbb{E}[\|\nabla_y f(G(x_{r,t}), y_{r,t})\|^2]$$
$$- \frac{17\eta_{x,r}}{10}\mathbb{E}[\|\nabla_x f(H(x_{r,t}), y_{r,t}) - p_{r,t}\|^2] - \frac{19\eta_{y,r}}{5}\mathbb{E}[\|\nabla_y f(H(x_{r,t}), y_{r,t}) - q_{r,t}\|^2]$$
$$- \left(\frac{17}{10}\eta_{x,r}K\sum_{k=1}^{K}A_k + 2c_0\eta_{x,r}L_f^2K\sum_{k=1}^{K}C_g^{2(K-k)}\right)\mathbb{E}[\|h_{r,t}^{(k)} - g^{(k)}(h_{r,t}^{(k-1)})\|^2]$$
$$+ \frac{16\alpha^2\eta_{x,r}^3}{\rho_x}C_p^2\sigma^2\sum_{k=1}^{K}\sum_{j=1}^{k}(2C_g^2)^{j-1} + 8\rho_x\eta_{x,r}^3 C_p^2\sigma^2 + 2\alpha^2\eta_{x,r}^4\sigma^2\sum_{k=1}^{K}\lambda_k\sum_{j=0}^{k-1}(2C_g^2)^j$$
$$+ \frac{16\alpha^2\eta_{x,r}^4}{\rho_y\eta_{y,r}}L_f^2\sigma^2\sum_{k=1}^{K}(2C_g^2)^{k-1} + 8\rho_y\eta_{y,r}^3\sigma^2$$
$$\leq -\frac{\eta_{x,r}}{4}\mathbb{E}[\|\nabla\Phi(x_{r,t})\|^2] - \frac{\eta_{x,r}}{4}\frac{c_0\eta_{x,r}}{\eta_{y,r}}\mathbb{E}[\|\nabla_y f(G(x_{r,t}), y_{r,t})\|^2] - \frac{c_0\eta_{x,r}}{16}\mathbb{E}[\|\nabla_y f(G(x_{r,t}), y_{r,t})\|^2]$$
$$- \frac{17\eta_{x,r}}{10}\mathbb{E}[\|\nabla_x f(H(x_{r,t}), y_{r,t}) - p_{r,t}\|^2] - \frac{19\eta_{y,r}}{5}\mathbb{E}[\|\nabla_y f(H(x_{r,t}), y_{r,t}) - q_{r,t}\|^2]$$
$$- \left(\frac{17}{10}\eta_{x,r}K\sum_{k=1}^{K}A_k + 2c_0\eta_{x,r}L_f^2K\sum_{k=1}^{K}C_g^{2(K-k)}\right)\mathbb{E}[\|h_{r,t}^{(k)} - g^{(k)}(h_{r,t}^{(k-1)})\|^2]$$

$$+ \frac{16\alpha^2\eta_{x,r}^3}{\rho_x}C_p^2\sigma^2\sum_{k=1}^{K}\sum_{j=1}^{k}(2C_g^2)^{j-1} + 8\rho_x\eta_{x,r}^3C_p^2\sigma^2 + 2\alpha^2\eta_{x,r}^4\sigma^2\sum_{k=1}^{K}\lambda_k\sum_{j=0}^{k-1}(2C_g^2)^j$$

$$+ \frac{16\alpha^2\eta_{x,r}^4}{\rho_y\eta_{y,r}}L_f^2\sigma^2\sum_{k=1}^{K}(2C_g^2)^{k-1} + 8\rho_y\eta_{y,r}^3\sigma^2 , \tag{131}$$

where the last step holds due to $\alpha\eta_{x,r}^2 \le 1$, and $-\frac{\eta_{x,r}}{4} \ge -\frac{\eta_{y,r}}{8}$ since $\eta_{x,r} = \frac{\eta_{y,r}}{10c_0}$, $c_0 = \frac{25\ell^2}{\mu^2}$.

Then, we set

$$L_\beta^2 = \max\{1 , \sum_{k=1}^{K}\lambda_k'\sum_{j=0}^{k}(2C_g^2)^j + C_p^2\sum_{k=1}^{K}\sum_{j=0}^{k}(2C_g^2)^j\} ,$$

$$\rho_y = 640L_\beta^2 , \rho_x = 6400c_0L_\beta^2, \alpha = 640c_0L_\beta^2 , \tag{132}$$

where $\lambda_k'$ is defined in Eq.(115). It is easy to verify that the conditions in Eq. (129) are satisfied. Meanwhile, this indicates $\rho_x = 10c_0\rho_y, \alpha = c_0\rho_y$.

By summing $t$ from 0 to $T_r - 1$, we obtain

$$\frac{1}{T_r}\sum_{t=0}^{T_r-1}\left(\mathbb{E}[\|\nabla\Phi(x_{r,t})\|^2] + \frac{c_0\eta_{x,r}}{\eta_{y,r}}\mathbb{E}[\|\nabla_y f(G(x_{r,t}), y_{r,t})\|^2]\right)$$

$$\le \frac{4(\mathcal{H}_{r,0} - \mathcal{H}_{t,T_r})}{\eta_{x,r}T_r} + \frac{64\alpha^2\eta_{x,r}^2}{\rho_x}C_p^2\sigma^2\sum_{k=1}^{K}\sum_{j=1}^{k}(2C_g^2)^{j-1} + 32\rho_x\eta_{x,r}^2C_p^2\sigma^2$$

$$+ 8\alpha^2\eta_{x,r}^3\sigma^2\sum_{k=1}^{K}\lambda_k\sum_{j=0}^{k-1}(2C_g^2)^j + \frac{64\alpha^2\eta_{x,r}^3}{\rho_y\eta_{y,r}}L_f^2\sigma^2\sum_{k=1}^{K}(2C_g^2)^{k-1} + \frac{32\rho_y\eta_{y,r}^3}{\eta_{x,r}}\sigma^2$$

$$\le \frac{4\mathcal{V}_{r,0}}{\eta_{x,r}T_r} + \frac{16\sigma_{r,0}^x}{\rho_x\eta_{x,r}^2T_r} + \frac{16\sigma_{r,0}^y}{\rho_y\eta_{x,r}\eta_{y,r}T_r} + \frac{4\sigma_{r,0}^{h,k}}{\eta_{x,r}T_r}\left(\frac{8\lambda_{k,1}}{\alpha\eta_{x,r}} + \frac{8c_0\lambda_{k,2}}{\alpha\eta_{x,r}} + \frac{32\lambda_{k,3}}{\rho_x\eta_{x,r}} + \frac{32\lambda_{k,4}}{\rho_y\eta_{y,r}}\right)$$

$$+ \frac{64\alpha^2\eta_{x,r}^2}{\rho_x}C_p^2\sigma^2\sum_{k=1}^{K}\sum_{j=1}^{k}(2C_g^2)^{j-1} + 32\rho_x\eta_{x,r}^2C_p^2\sigma^2 + \frac{64\alpha^2\eta_{x,r}^3}{\rho_y\eta_{y,r}}L_f^2\sigma^2\sum_{k=1}^{K}(2C_g^2)^{k-1}$$

$$+ 8\alpha^2\eta_{x,r}^3\sigma^2\sum_{k=1}^{K}\left(\frac{8\lambda_{k,1}}{\alpha\eta_{x,r}} + \frac{8c_0\lambda_{k,2}}{\alpha\eta_{x,r}} + \frac{32\lambda_{k,3}}{\rho_x\eta_{x,r}} + \frac{32\lambda_{k,4}}{\rho_y\eta_{y,r}}\right)\sum_{j=0}^{k-1}(2C_g^2)^j + \frac{32\rho_y\eta_{y,r}^3}{\eta_{x,r}}\sigma^2$$

$$\le \frac{40c_0\mathcal{V}_{r,0}}{\eta_{y,r}T_r} + \frac{160\sigma_{r,0}^x}{\rho_y\eta_{y,r}^2T_r} + \frac{160c_0\sigma_{r,0}^y}{\rho_y\eta_{y,r}^2T_r} + \frac{8960c_0\sigma_{r,0}^h}{\rho_y\eta_{y,r}^2T_r}$$

$$+ \frac{8}{125}\rho_y\eta_{y,r}^2C_p^2\sigma^2\sum_{k=1}^{K}\sum_{j=1}^{k}(2C_g^2)^{j-1} + \frac{32}{10}\rho_y\eta_{y,r}^2C_p^2\sigma^2 + \frac{8}{125}c_0\rho_y\eta_{y,r}^2L_f^2\sigma^2\sum_{k=1}^{K}(2C_g^2)^{k-1}$$

$$+ 8\lambda_k'\sigma^2\sum_{k=1}^{K}\left(\frac{8}{100}\rho_y\eta_{y,r}^2 + \frac{8}{100}c_0\rho_y\eta_{y,r}^2 + \frac{32\rho_y\eta_{y,r}^2}{1000} + \frac{32c_0\rho_y\eta_{y,r}^2}{1000}\right)\sum_{j=0}^{k-1}(2C_g^2)^j + 320c_0\rho_y\eta_{y,r}^2\sigma^2$$

$$\le \frac{40c_0\mathcal{V}_{r,0}}{\eta_{y,r}T_r} + \frac{160c_0}{\rho_y\eta_{y,r}^2T_r}(\sigma_{r,0}^x + \sigma_{r,0}^y + 56\sigma_{r,0}^h) + 330c_0L_\beta^2\rho_y\eta_{y,r}^2\sigma^2 , \tag{133}$$

where $\mathcal{V}_{r,0} = \mathbb{E}[\Phi(x_{r,0})] - \Phi(x_*) + \frac{c_0\eta_{x,r}}{\eta_{y,r}}(\mathbb{E}[\Phi(x_{r,0})] - \mathbb{E}[f(g(x_{r,0}), y_{r,0})])$, $\sigma_{r,0}^x = \mathbb{E}[\|\nabla_x f(H(x_{r,0}), y_{r,0}) - p_{r,0}\|^2]$, $\sigma_{r,0}^y = \mathbb{E}[\|\nabla_y f(H(x_{r,0}), y_{r,0}) - q_{r,0}\|^2]$ and $\sigma_{r,0}^{h,k} = \mathbb{E}[\|g^{(k)}(h_{r,0}^{(k-1)}) - h_{r,0}^{(k)}\|^2]$, $\sigma_{r,0}^h = \sum_{k=1}^{K}\lambda_k'\sigma_{r,0}^{h,k}$. $\qquad\square$

## C.2 Proof of the Theorem C.1

*Proof.* Based on Lemma C.6, we have

$$\frac{1}{T_r} \sum_{t=0}^{T_r-1} \left( \mathbb{E}[\|\nabla\Phi(x_{r,t})\|^2] + \frac{c_0\eta_{x,r}}{\eta_{y,r}} \mathbb{E}[\|\nabla_y f(G(x_{r,t}), y_{r,t})\|^2] \right)$$

$$\leq \frac{40c_0\mathcal{V}_{r,0}}{\eta_{y,r}T_r} + \frac{160c_0}{\rho_y\eta_{y,r}^2 T_r}(\sigma_{r,0}^x + \sigma_{r,0}^y + 56\sigma_{r,0}^h) + 330c_0 L_\beta^2 \rho_y \eta_{y,r}^2 \sigma^2 . \tag{134}$$

Since $c_0 = O(\kappa^2)$, then it is easy to verify that by setting by setting $\eta_{y,1} = O(\epsilon/\kappa)$, $\eta_{x,1} = O(\epsilon/\kappa^3)$, $T_1 = O(\kappa^3/\epsilon^3)$, and the initial batch size as $O(\kappa/\epsilon)$, we have

$$\frac{40c_0\mathcal{V}_{r,0}}{\eta_{y,r}T_r} \leq O(\epsilon^2) ,$$

$$\frac{160c_0}{\rho_y\eta_{y,r}^2 T_r}(\sigma_{r,0}^x + \sigma_{r,0}^y + 56\sigma_{r,0}^h) \leq O(\epsilon^2) ,$$

$$330c_0 L_\beta^2 \rho_y \eta_{y,r}^2 \sigma^2 \leq O(\epsilon^2) . \tag{135}$$

As a result, we can conclude $\frac{1}{T_1} \sum_{t=0}^{T_1-1} \mathbb{E}[\|\nabla\Phi(x_{1,t})\|^2] \leq \epsilon^2$.

$\square$

## C.3 Proof of the Theorem C.2

**Lemma C.7.** *Assumption 3.1-3.4 , we have*

$$\sigma_{r+1,0}^x + \sigma_{r+1,0}^y + 56\sigma_{r+1,0}^h \leq \frac{320c_0}{\rho_y\eta_{y,r}^2 T_r}\left(\sigma_{r,0}^x + \sigma_{r,0}^y + 56\sigma_{r,0}^h\right) + \frac{20c_0\mathcal{V}_{r,0}}{\eta_{y,r}T_r} + 338c_0\rho_y\eta_{y,r}^2 L_\beta^2\sigma^2 . \tag{136}$$

*Proof.* In the following, we will bound $\sigma_{r,0}^x + \sigma_{r,0}^y + 56\sigma_{r,0}^h$. At first, from Lemma B.5, we get

$$\frac{1}{T_r} \sum_{t=0}^{T_r-1} \sum_{k=1}^K \lambda_k' \mathbb{E}[\|h_{r,t}^{(k)} - g^{(k)}(h_{r,t}^{(k-1)})\|^2]$$

$$\leq \frac{1}{\alpha\eta_{x,r}^2 T_r} \sum_{k=1}^K \lambda_k' \mathbb{E}[\|h_{r,0}^{(k)} - g^{(k)}(h_{r,0}^{(k-1)})\|^2] + \frac{2\alpha\eta_{x,r}^2}{T_r} \sum_{k=1}^K \sum_{j=k+1}^K \lambda_j'(2C_g^2)^{j-k} \mathbb{E}[\|h_{r,0}^{(k)} - g^{(k)}(h_{r,0}^{(k-1)})\|^2]$$

$$+ \frac{1}{\alpha} \sum_{k=1}^K \lambda_k'(2C_g^2)^k \frac{1}{T_r} \sum_{t=0}^{T_r-1} \mathbb{E}[\|p_{r,t}\|^2] + 2\alpha\eta_{x,r}^2\sigma^2 \sum_{k=1}^K \lambda_k' \sum_{j=0}^{k-1}(2C_g^2)^j$$

$$\leq \frac{1}{\alpha\eta_{x,r}^2 T_r} \sum_{k=1}^K \lambda_k' \mathbb{E}[\|h_{r,0}^{(k)} - g^{(k)}(h_{r,0}^{(k-1)})\|^2] + \frac{2\alpha^2\eta_{x,r}^4}{\alpha\eta_{x,r}^2 T_r} \sum_{k=1}^K \sum_{j=k+1}^K \lambda_j'(2C_g^2)^{j-k} \mathbb{E}[\|h_{r,0}^{(k)} - g^{(k)}(h_{r,0}^{(k-1)})\|^2]$$

$$+ \frac{1}{\alpha} \sum_{k=1}^K \lambda_k'(2C_g^2)^k \frac{1}{T_r} \sum_{t=0}^{T_r-1} \mathbb{E}[\|p_{r,t}\|^2] + 2\alpha\eta_{x,r}^2\sigma^2 \sum_{k=1}^K \lambda_k' \sum_{j=0}^{k-1}(2C_g^2)^j$$

$$\leq \frac{100}{\rho_y\eta_{y,r}^2 T_r} \sum_{k=1}^K \lambda_k' \mathbb{E}[\|h_{r,0}^{(k)} - g^{(k)}(h_{r,0}^{(k-1)})\|^2] + \frac{50}{\rho_y\eta_{y,r}^2 T_r} \sum_{k=1}^K \lambda_k' \mathbb{E}[\|h_{r,0}^{(k)} - g^{(k)}(h_{r,0}^{(k-1)})\|^2]$$

$$+ \frac{L_\beta^2}{c_0^2\rho_y} \frac{1}{T_r} \sum_{t=0}^{T_r-1} \mathbb{E}[\|p_{r,t}\|^2] + \frac{\rho_y\eta_{y,r}^2}{50}\sigma^2 \sum_{k=1}^K \lambda_k' \sum_{j=0}^{k-1}(2C_g^2)^j , \tag{137}$$

where the last step holds due to $\alpha\eta_{x,r}^2 \leq 1$. Then, according to the random sampling operation, we have

$$\sigma_{r+1,0}^h = \sum_{k=1}^K \lambda_k' \mathbb{E}[\|g^{(k)}(x_{r+1,0}) - h_{r+1,0}^{(k)}\|^2] = \frac{1}{T_r} \sum_{t=0}^{T_r-1} \sum_{k=1}^K \lambda_k' \mathbb{E}[\|g^{(k)}(h_{r,t}^{(k-1)}) - h_{r,t}^{(k)}\|^2]$$

$$
\leq \frac{150}{\rho_y \eta_{y,r}^2 T_r} \sum_{k=1}^{K} \lambda_k' \sigma_{r,0}^{h,k} + \frac{L_\beta^2}{c_0^2 \rho_y} \frac{1}{T_r} \sum_{t=0}^{T_r-1} \mathbb{E}[\|p_{r,t}\|^2] + \frac{\rho_y \eta_{y,r}^2}{50} \sigma^2 \sum_{k=1}^{K} \lambda_k' \sum_{j=0}^{k-1} (2C_g^2)^j
$$

$$
\leq \frac{150}{\rho_y \eta_{y,r}^2 T_r} \sigma_{r,0}^h + \frac{L_\beta^2}{\rho_y} \left( \frac{1}{T_r} \sum_{t=0}^{T_r-1} \mathbb{E}[\|p_{r,t}\|^2] + \frac{1}{10} \frac{1}{T_r} \sum_{t=0}^{T_r-1} \mathbb{E}[|q_{r,t}\|^2] \right) + \frac{\rho_y \eta_{y,r}^2}{50} \sigma^2 \sum_{k=1}^{K} \lambda_k' \sum_{j=0}^{k-1} (2C_g^2)^j
$$

$$
\leq \frac{150 \sigma_{r,0}^h}{\rho_y \eta_{y,r}^2 T_r} + \frac{1}{640} \left( \frac{1}{T_r} \sum_{t=0}^{T_r-1} \mathbb{E}[\|p_{r,t}\|^2] + \frac{1}{10} \frac{1}{T_r} \sum_{t=0}^{T_r-1} \mathbb{E}[|q_{r,t}\|^2] \right) + \frac{1}{50} \rho_y \eta_{y,r}^2 L_\beta^2 \sigma^2 , \tag{138}
$$

where the last step holds due to $\rho_y = 640 L_\beta^2$.

Then, based on Lemma B.6, we have

$$
\frac{1}{T_r} \sum_{t=0}^{T_r-1} \mathbb{E}[\|\nabla_x f(H(x_{r,t}), y_{r,t}) - p_{r,t}\|^2]
$$

$$
\leq \frac{1}{\rho_x \eta_{x,r}^2 T_r} \mathbb{E}[\|\nabla_x f(H(x_{r,0}), y_{r,0}) - p_{r,0}\|^2] + \frac{2\eta_{y,r}^2}{\rho_x \eta_{x,r}^2} C_p^2 \frac{1}{T_r} \sum_{t=0}^{T_r-1} \mathbb{E}[\|q_{r,t}\|^2]
$$

$$
+ \frac{4\alpha^2 \eta_{x,r}^2}{\rho_x} C_p^2 \frac{1}{T_r} \sum_{t=0}^{T_r-1} \sum_{k=1}^{K} \left( \sum_{j=k}^{K} (2C_g^2)^{j-k} \right) \mathbb{E}[\|h_{r,t}^{(k)} - g^{(k)}(h_{r,t}^{(k-1)})\|^2] + \frac{2}{\rho_x} C_p^2 \sum_{k=0}^{K} (2C_g^2)^k \frac{1}{T_r} \sum_{t=0}^{T_r-1} \mathbb{E}[\|p_{r,t}\|^2]
$$

$$
+ \frac{4\alpha^2 \eta_{x,r}^2}{\rho_x} C_p^2 \sigma^2 \sum_{k=1}^{K} \sum_{j=1}^{k} (2C_g^2)^{j-1} + 2\rho_x \eta_{x,r}^2 C_p^2 \sigma^2 \tag{139}
$$

$$
\leq \frac{10}{\rho_y \eta_{y,r}^2 T_r} \mathbb{E}[\|\nabla_x f(H(x_{r,0}), y_{r,0}) - p_{r,0}\|^2] + \frac{2L_\beta^2}{10 c_0^2 \rho_y} \frac{1}{T_r} \sum_{t=0}^{T_r-1} \mathbb{E}[\|p_{r,t}\|^2] + \frac{20 L_\beta^2}{\rho_y} \frac{1}{T_r} \sum_{t=0}^{T_r-1} \mathbb{E}[\|q_{r,t}\|^2]
$$

$$
+ \frac{1}{250} \frac{1}{T_r} \sum_{t=0}^{T_r-1} \sum_{k=1}^{K} \lambda_k' \mathbb{E}[\|h_{r,t}^{(k)} - g^{(k)}(h_{r,t}^{(k-1)})\|^2] + \frac{\rho_y \eta_{y,r}^2}{250} C_p^2 \sigma^2 \sum_{k=1}^{K} \sum_{j=1}^{k} (2C_g^2)^{j-1} + \frac{1}{5} \rho_y \eta_{y,r}^2 C_p^2 \sigma^2 ,
$$

where the last step holds due to the definition of $\lambda_k'$ and $\alpha \eta_{x,r}^2 \leq 1$. Then, due to the random sampling in each outer iteration, it is easy to know

$$
\sigma_{r+1,0}^x = \mathbb{E}[\|\nabla_x f(H(x_{r+1,0}), y_{r+1,0}) - p_{r+1,0}\|^2] = \frac{1}{T_r} \sum_{t=0}^{T_r-1} \mathbb{E}[\|\nabla_x f(H(x_{r,t}), y_{r,t}) - p_{r,t}\|^2]
$$

$$
\leq \frac{10}{\rho_y \eta_{y,r}^2 T_r} \mathbb{E}[\|\nabla_x f(H(x_{r,0}), y_{r,0}) - p_{r,0}\|^2] + \frac{200 L_\beta^2}{\rho_y} \left( \frac{1}{T_r} \sum_{t=0}^{T_r-1} \mathbb{E}[\|p_{r,t}\|^2] + \frac{1}{10} \frac{1}{T_r} \sum_{t=0}^{T_r-1} \mathbb{E}[\|q_{r,t}\|^2] \right)
$$

$$
+ \frac{1}{250} \left( \frac{150}{\rho_y \eta_{y,r}^2 T_r} \sigma_{r,0}^h + \frac{1}{640} \left( \frac{1}{T_r} \sum_{t=0}^{T_r-1} \mathbb{E}[\|p_{r,t}\|^2] + \frac{1}{10} \frac{1}{T_r} \sum_{t=0}^{T_r-1} \mathbb{E}[|q_{r,t}\|^2] \right) + \frac{1}{50} \rho_y \eta_{y,r}^2 L_\beta^2 \sigma^2 \right)
$$

$$
+ \frac{4\rho_y \eta_{y,r}^2}{1000} C_p^2 \sigma^2 \sum_{k=1}^{K} \sum_{j=1}^{k} (2C_g^2)^{j-1} + \frac{1}{5} \rho_y \eta_{y,r}^2 C_p^2 \sigma^2
$$

$$
\leq \frac{10 \sigma_{r,0}^x}{\rho_y \eta_{y,r}^2 T_r} + \frac{\sigma_{r,0}^h}{\rho_y^2 \eta_{y,r}^2 T_r} + \frac{201}{640} \left( \frac{1}{T_r} \sum_{t=0}^{T_r-1} \mathbb{E}[\|p_{r,t}\|^2] + \frac{1}{10} \frac{1}{T_r} \sum_{t=0}^{T_r-1} \mathbb{E}[\|q_{r,t}\|^2] \right) + \frac{2}{5} \rho_y \eta_{y,r}^2 L_\beta^2 \sigma^2 , \tag{140}
$$

where the last step holds due to $\rho_y = 640 L_\beta^2$, $c_0 > 1$ and $\frac{c_0 \eta_{x,r}}{\eta_{y,r}} = \frac{1}{10}$.

Similarly, from Lemma B.7, we get

$$
\frac{1}{T_r} \sum_{t=0}^{T_r-1} \mathbb{E}[\|\nabla_y f(H(x_{r,t}), y_{r,t}) - q_{r,t}\|^2]
$$

$$
\leq \frac{1}{\rho_y \eta_{y,r}^2 T_r} \mathbb{E}[\|\nabla_y f(H(x_{r,0}), y_{r,0}) - q_{r,0}\|^2] + \frac{4\alpha^2 \eta_{x,r}^4}{\rho_y \eta_{y,r}^2} L_f^2 \frac{1}{T_r} \sum_{t=0}^{T_r-1} \sum_{k=1}^{K} (2C_g^2)^{K-k} \mathbb{E}[\|h_{r,t}^{(k)} - g^{(k)}(h_{r,t}^{(k-1)})\|^2]
$$

$$
+ \frac{2\eta_{x,r}^2}{\rho_y \eta_{y,r}^2} L_f^2 (2C_g^2)^K \frac{1}{T_r} \sum_{t=0}^{T_r-1} \mathbb{E}[\|p_{r,t}\|^2] + \frac{2L_f^2}{\rho_y} \frac{1}{T_r} \sum_{t=0}^{T_r-1} \mathbb{E}[\|q_{r,t}\|^2] + \frac{4\alpha^2 \eta_{x,r}^4}{\rho_y \eta_{y,r}^2} L_f^2 \sigma^2 \sum_{k=1}^{K} (2C_g^2)^{k-1} + 2\rho_y \eta_{y,r}^2 \sigma^2
$$

$$\leq \frac{1}{\rho_y \eta_{y,r}^2 T_r} \mathbb{E}[\|\nabla_y f(H(x_{r,0}), y_{r,0}) - q_{r,0}\|^2] + \frac{1}{25} \frac{1}{T_r} \sum_{t=0}^{T_r-1} \sum_{k=1}^{K} \lambda_k' \mathbb{E}[\|h_{r,t}^{(k)} - g^{(k)}(h_{r,t}^{(k-1)})\|^2] \qquad (141)$$

$$+ \frac{L_\beta^2}{50 c_0^2 \rho_y} \frac{1}{T_r} \sum_{t=0}^{T_r-1} \mathbb{E}[\|p_{r,t}\|^2] + \frac{2L_\beta^2}{\rho_y} \frac{1}{T_r} \sum_{t=0}^{T_r-1} \mathbb{E}[\|q_{r,t}\|^2] + \frac{4\rho_y \eta_{y,r}^2}{100} L_f^2 \sigma^2 \sum_{k=1}^{K} (2C_g^2)^{k-1} + 2\rho_y \eta_{y,r}^2 \sigma^2 ,$$

where the second step holds due to the definition of $\lambda_k'$, and the last step holds due to $\alpha \eta_{x,r}^2 \leq 1$. Then, due to the randomly sampling operation in each outer iteration, it is easy to know

$$\sigma_{r+1,0}^y = \mathbb{E}[\|\nabla_y f(H(x_{r+1,0}), y_{r+1,0}) - q_{r+1,0}\|^2] = \frac{1}{T_r} \sum_{t=0}^{T_r-1} \mathbb{E}[\|\nabla_y f(H(x_{r,t}), y_{r,t}) - q_{r,t}\|^2]$$

$$\leq \frac{1}{\rho_y \eta_{y,r}^2 T_r} \mathbb{E}[\|\nabla_y f(H(x_{r,0}), y_{r,0}) - q_{r,0}\|^2] + \frac{L_\beta^2}{50 c_0^2 \rho_y} \frac{1}{T_r} \sum_{t=0}^{T_r-1} \mathbb{E}[\|p_{r,t}\|^2] + \frac{2L_\beta^2}{\rho_y} \frac{1}{T_r} \sum_{t=0}^{T_r-1} \mathbb{E}[\|q_{r,t}\|^2]$$

$$+ \frac{1}{25} \frac{1}{T_r} \sum_{t=0}^{T_r-1} \sum_{k=1}^{K} \lambda_k' \mathbb{E}[\|h_{r,t}^{(k)} - g^{(k)}(h_{r,t}^{(k-1)})\|^2] + \frac{4\rho_y \eta_{y,r}^2}{10000} L_f^2 \sigma^2 \sum_{k=1}^{K} (2C_g^2)^{k-1} + 2\rho_y \eta_{y,r}^2 \sigma^2$$

$$\leq \frac{1}{\rho_y \eta_{y,r}^2 T_r} \mathbb{E}[\|\nabla_y f(G(x_{r,0}), y_{r,0}) - q_{r,0}\|^2] + \frac{20 L_\beta^2}{\rho_y} \left( \frac{1}{T_r} \sum_{t=0}^{T_r-1} \mathbb{E}[\|p_{r,t}\|^2] + \frac{1}{10} \frac{1}{T_r} \sum_{t=0}^{T_r-1} \mathbb{E}[\|q_{r,t}\|^2] \right)$$

$$+ \frac{1}{25} \left( \frac{150}{\rho_y \eta_{y,r}^2 T_r} \sigma_{r,0}^h + \frac{1}{640} \left( \frac{1}{T_r} \sum_{t=0}^{T_r-1} \mathbb{E}[\|p_{r,t}\|^2] + \frac{1}{10} \frac{1}{T_r} \sum_{t=0}^{T_r-1} \mathbb{E}[|q_{r,t}\|^2] \right) + \frac{1}{50} \rho_y \eta_{y,r}^2 L_\beta^2 \sigma^2 \right)$$

$$+ \frac{4\rho_y \eta_{y,r}^2}{10000} L_f^2 \sigma^2 \sum_{k=1}^{K} (2C_g^2)^{k-1} + 2\rho_y \eta_{y,r}^2 \sigma^2 \qquad (142)$$

$$\leq \frac{\sigma_{r,0}^y}{\rho_y \eta_{y,r}^2 T_r} + \frac{6\sigma_{r,0}^h}{\rho_y \eta_{y,r}^2 T_r} + \frac{21}{640} \left( \frac{1}{T_r} \sum_{t=0}^{T_r-1} \mathbb{E}[\|p_{r,t}\|^2] + \frac{1}{10} \frac{1}{T_r} \sum_{t=0}^{T_r-1} \mathbb{E}[|q_{r,t}\|^2] \right) + \frac{11}{5} \rho_y \eta_{y,r}^2 L_\beta^2 \sigma^2 ,$$

where the last step holds due to $\rho_y = 640 L_\beta^2$.

Then, we combine these three inequalities together as follows:

$$\sigma_{r+1,0}^x + \sigma_{r+1,0}^y + 56 \sigma_{r,0}^h$$

$$\leq \frac{10\sigma_{r,0}^x}{\rho_y \eta_{y,r}^2 T_r} + \frac{\sigma_{r,0}^h}{\rho_y^2 \eta_{y,r}^2 T_r} + \frac{201}{640} \left( \frac{1}{T_r} \sum_{t=0}^{T_r-1} \mathbb{E}[\|p_{r,t}\|^2] + \frac{1}{10} \frac{1}{T_r} \sum_{t=0}^{T_r-1} \mathbb{E}[|q_{r,t}\|^2] \right) + \frac{2}{5} \rho_y \eta_{y,r}^2 L_\beta^2 \sigma^2$$

$$+ \frac{\sigma_{r,0}^y}{\rho_y \eta_{y,r}^2 T_r} + \frac{6\sigma_{r,0}^h}{\rho_y \eta_{y,r}^2 T_r} + \frac{21}{640} \left( \frac{1}{T_r} \sum_{t=0}^{T_r-1} \mathbb{E}[\|p_{r,t}\|^2] + \frac{1}{10} \frac{1}{T_r} \sum_{t=0}^{T_r-1} \mathbb{E}[|q_{r,t}\|^2] \right) + \frac{11}{5} \rho_y \eta_{y,r}^2 L_\beta^2 \sigma^2$$

$$+ \frac{8400 \sigma_{r,0}^h}{\rho_y \eta_{y,r}^2 T_r} + \frac{56}{640} \left( \frac{1}{T_r} \sum_{t=0}^{T_r-1} \mathbb{E}[\|p_{r,t}\|^2] + \frac{1}{10} \frac{1}{T_r} \sum_{t=0}^{T_r-1} \mathbb{E}[|q_{r,t}\|^2] \right) + \frac{56}{50} \rho_y \eta_{y,r}^2 L_\beta^2 \sigma^2 \qquad (143)$$

$$\leq \frac{10\sigma_{r,0}^x}{\rho_y \eta_{y,r}^2 T_r} + \frac{\sigma_{r,0}^y}{\rho_y \eta_{y,r}^2 T_r} + \frac{8407 \sigma_{r,0}^h}{\rho_y \eta_{y,r}^2 T_r} + \frac{1}{2} \left( \frac{1}{T_r} \sum_{t=0}^{T_r-1} \mathbb{E}[\|p_{r,t}\|^2] + \frac{1}{10} \frac{1}{T_r} \sum_{t=0}^{T_r-1} \mathbb{E}[|q_{r,t}\|^2] \right) + 8\rho_y \eta_{y,r}^2 L_\beta^2 \sigma^2 .$$

Then, we need to bound $\frac{1}{T_r} \sum_{t=0}^{T_r-1} \mathbb{E}[\|p_{r,t}\|^2] + \frac{c_0 \eta_{y,r}}{\eta_{x,r}} \frac{1}{T_r} \sum_{t=0}^{T_r-1} \mathbb{E}[\|q_{r,t}\|^2]$. In particular, we have

$$\mathbb{E}[\|p_{r,t}\|^2] + \frac{c_0 \eta_{x,r}}{\eta_{y,r}} \mathbb{E}[\|q_{r,t}\|^2]$$

$$\leq 2\mathbb{E}[\|p_{r,t} - \nabla \Phi(x_{r,t})\|^2] + 2\mathbb{E}[\|\nabla \Phi(x_{r,t})\|^2]$$

$$+ \frac{c_0 \eta_{x,r}}{\eta_{y,r}} \left( 2\mathbb{E}[\|q_{r,t} - \nabla_y f(H(x_{r,t}), y_{r,t}) + \nabla_y f(H(x_{r,t}), y_{r,t}) - \nabla_y f(G(x_{r,t}), y_{r,t})\|^2] \right.$$

$$\left. + 2\mathbb{E}[\|\nabla_y f(G(x_{r,t}), y_{r,t})\|^2] \right)$$

$$\leq 4\mathbb{E}[\|\nabla \Phi(x_{r,t}) - \nabla_x f(G(x_{r,t}), y_{r,t})\|^2] + 8\mathbb{E}[\|\nabla_x f(G(x_{r,t}), y_{r,t}) - \nabla_x f(H(x_{r,t}), y_{r,t})\|^2]$$

$$+ 8\mathbb{E}[\|\nabla_x f(H(x_{r,t}), y_{r,t}) - p_{r,t}\|^2] + 2\mathbb{E}[\|\nabla \Phi(x_{r,t})\|^2]$$

$$+ \frac{c_0 \eta_{x,r}}{\eta_{y,r}} \left( 4\mathbb{E}[\|\nabla_y f(H(x_{r,t}), y_{r,t}) - q_{r,t}\|^2] + 4\mathbb{E}[\|\nabla_y f(H(x_{r,t}), y_{r,t}) - \nabla_y f(G(x_{r,t}), y_{r,t})\|^2] \right.$$

$$+ 2\mathbb{E}[\|\nabla_y f(G(x_{r,t}), y_{r,t})\|^2]\Big)$$

$$\leq \frac{8}{\eta_{x,r}} \Big( \frac{\eta_{x,r}}{4} \mathbb{E}[\|\nabla \Phi(x_{r,t})\|^2] + \frac{\eta_{x,r}}{4} \frac{c_0 \eta_{x,r}}{\eta_{y,r}} \mathbb{E}[\|\nabla_y f(G(x_{r,t}), y_{r,t})\|^2]\Big) \tag{144}$$

$$+ 4\frac{L_f^2}{\mu^2} \mathbb{E}[\|\nabla_y f(G(x_{r,t}), y_{r,t})\|^2] + 8K \sum_{k=1}^{K} A_k \mathbb{E}[\|g^{(k)}(h_{r,t}^{(k-1)}) - h_{r,t}^{(k)}\|^2] + 8\mathbb{E}[\|\nabla_x f(H(x_{r,t}), y_{r,t}) - p_{r,t}\|^2]$$

$$+ \frac{4c_0 \eta_{x,r}}{\eta_{y,r}} \mathbb{E}[\|\nabla_y f(H(x_{r,t}), y_{r,t}) - q_{r,t}\|^2] + \frac{4c_0 \eta_{x,r}}{\eta_{y,r}} L_f^2 \sum_{k=1}^{K} C_g^{2(K-k)} \mathbb{E}[\|g^{(k)}(h_{r,t}^{(k-1)}) - h_{r,t}^{(k)}\|^2].$$

By plugging Eq. (131), we obtain

$$\mathbb{E}[\|p_{r,t}\|^2] + \frac{c_0 \eta_{x,r}}{\eta_{y,r}} \mathbb{E}[\|q_{r,t}\|^2] \leq \frac{8(\mathcal{H}_{r,t} - \mathcal{H}_{r,t+1})}{\eta_{x,r}} - \frac{c_0}{2} \mathbb{E}[\|\nabla_y f(G(x_{r,t}), y_{r,t})\|^2]$$

$$- \frac{68}{5} \mathbb{E}[\|\nabla_x f(H(x_{r,t}), y_{r,t}) - p_{r,t}\|^2] - \frac{152\eta_{y,r}}{5\eta_{x,r}} \mathbb{E}[\|\nabla_y f(H(x_{r,t}), y_{r,t}) - q_{r,t}\|^2]$$

$$- \Big( \frac{68}{5} K \sum_{k=1}^{K} A_k + 16c_0 L_f^2 \sum_{k=1}^{K} C_g^{2(K-k)} \Big) \mathbb{E}[\|h_{r,t}^{(k)} - g^{(k)}(h_{r,t}^{(k-1)})\|^2] + 664c_0 L_\beta^2 \rho_y \eta_{y,r}^2 \sigma^2$$

$$+ 4\frac{L_f^2}{\mu^2} \mathbb{E}[\|\nabla_y f(G(x_{r,t}), y_{r,t})\|^2] + 8K \sum_{k=1}^{K} A_k \mathbb{E}[\|g^{(k)}(h_{r,t}^{(k-1)}) - h_{r,t}^{(k)}\|^2] + 8\mathbb{E}[\|\nabla_x f(H(x_{r,t}), y_{r,t}) - p_{r,t}\|^2]$$

$$+ \frac{4c_0 \eta_{x,r}}{\eta_{y,r}} \mathbb{E}[\|q_{r,t} - \nabla_y f(H(x_{r,t}), y_{r,t})\|^2] + \frac{4c_0 \eta_{x,r}}{\eta_{y,r}} L_f^2 \sum_{k=1}^{K} C_g^{2(K-k)} \mathbb{E}[\|g^{(k)}(h_{r,t}^{(k-1)}) - h_{r,t}^{(k)}\|^2]$$

$$\leq \frac{8(\mathcal{H}_{r,t} - \mathcal{H}_{r,t+1})}{\eta_{x,r}} + 660c_0 L_\beta^2 \rho_y \eta_{y,r}^2 \sigma^2, \tag{145}$$

where $c_0 = \frac{25\ell^2}{\mu^2}$.

By summing up $t$ from 0 to $T_r - 1$, we get

$$\frac{1}{T_r} \sum_{t=0}^{T_r-1} \Big( \mathbb{E}[\|p_{r,t}\|^2] + \frac{c_0 \eta_{x,r}}{\eta_{y,r}} \mathbb{E}[\|q_{r,t}\|^2] \Big) \leq \frac{8(\mathcal{H}_{r,0} - \mathcal{H}_{t,T_r})}{\eta_{x,r} T_r} + 660c_0 L_\beta^2 \rho_y \eta_{y,r}^2 \sigma^2$$

$$\leq \frac{80c_0 \mathcal{V}_{r,0}}{\eta_{y,r} T_r} + \frac{320\sigma_{r,0}^x}{\rho_y \eta_{y,r}^2 T_r} + \frac{320c_0 \sigma_{r,0}^y}{\rho_y \eta_{y,r}^2 T_r} + \frac{17920c_0 \sigma_{r,0}^h}{\rho_y \eta_{y,r}^2 T_r} + 660c_0 L_\beta^2 \rho_y \eta_{y,r}^2 \sigma^2$$

$$\leq \frac{80c_0 \mathcal{V}_{r,0}}{\eta_{y,r} T_r} + \frac{320c_0}{\rho_y \eta_{y,r}^2 T_r} \Big( \sigma_{r,0}^x + \sigma_{r,0}^y + 56\sigma_{r,0}^h \Big) + 660c_0 L_\beta^2 \rho_y \eta_{y,r}^2 \sigma^2. \tag{146}$$

By plugging this inequality to Eq. (143), we get

$$\sigma_{r+1,0}^x + \sigma_{r+1,0}^y + 56\sigma_{r+1,0}^h$$

$$\leq \frac{10\sigma_{r,0}^x}{\rho_y \eta_{y,r}^2 T_r} + \frac{\sigma_{r,0}^y}{\rho_y \eta_{y,r}^2 T_r} + \frac{8407\sigma_{r,0}^h}{\rho_y \eta_{y,r}^2 T_r} + \frac{1}{2} \Big( \frac{1}{T_r} \sum_{t=0}^{T_r-1} \mathbb{E}[\|p_{r,t}\|^2] + \frac{1}{10} \frac{1}{T_r} \sum_{t=0}^{T_r-1} \mathbb{E}[\|q_{r,t}\|^2] \Big) + 8\rho_y \eta_{y,r}^2 L_\beta^2 \sigma^2$$

$$\leq \frac{10\sigma_{r,0}^x}{\rho_y \eta_{y,r}^2 T_r} + \frac{\sigma_{r,0}^y}{\rho_y \eta_{y,r}^2 T_r} + \frac{8407\sigma_{r,0}^h}{\rho_y \eta_{y,r}^2 T_r}$$

$$+ \frac{1}{2} \Big( \frac{40c_0 \mathcal{V}_{r,0}}{\eta_{y,r} T_r} + \frac{320c_0}{\rho_y \eta_{y,r}^2 T_r} \Big( \sigma_{r,0}^x + \sigma_{r,0}^y + 56\sigma_{r,0}^h \Big) + 660c_0 L_\beta^2 \rho_y \eta_{y,r}^2 \sigma^2 \Big) + 8\rho_y \eta_{y,r}^2 L_\beta^2 \sigma^2$$

$$\leq \frac{320c_0}{\rho_y \eta_{y,r}^2 T_r} \Big( \sigma_{r,0}^x + \sigma_{r,0}^y + 56\sigma_{r,0}^h \Big) + \frac{20c_0 \mathcal{V}_{r,0}}{\eta_{y,r} T_r} + 338c_0 \rho_y \eta_{y,r}^2 L_\beta^2 \sigma^2. \tag{147}$$

$$\square$$

In the following, we prove Theorem C.2.

*Proof.* Under the two-sided PL condition, we get

$$2\mu \Big( \mathbb{E}[\Phi(x_{r,t})] - \Phi(x_*) + \frac{c_0 \eta_{x,r}}{\eta_{y,r}} \big( \mathbb{E}[\Phi(x_{r,t})] - \mathbb{E}[f(G(x_{r,t}), y_{r,t})] \big) \Big)$$

$$\leq \mathbb{E}[\|\nabla\Phi(x_{r,t})\|^2] + \frac{c_0\eta_{x,r}}{\eta_{y,r}}\mathbb{E}[\|\nabla_y f(G(x_{r,t}), y_{r,t})\|^2] . \tag{148}$$

Due to the random sampling in each outer iteration, we get

$$\mathcal{V}_{r+1,0} = \frac{1}{T_r}\sum_{t=0}^{T_r-1}\left(\mathbb{E}[\Phi(x_{r,t})] - \Phi(x_*) + \frac{c_0\eta_{x,r}}{\eta_{y,r}}(\mathbb{E}[\Phi(x_{r,t})] - \mathbb{E}[f(g(x_{r,t}), y_{r,t})])\right)$$

$$\leq \frac{1}{2\mu}\frac{1}{T_r}\sum_{t=0}^{T_r-1}\left(\mathbb{E}[\|\nabla\Phi(x_{r,t})\|^2] + \frac{c_0\eta_{x,r}}{\eta_{y,r}}\mathbb{E}[\|\nabla_y f(g(x_{r,t}), y_{r,t})\|^2]\right)$$

$$\leq \frac{1}{2\mu}\left(\frac{40c_0\mathcal{V}_{r,0}}{\eta_{y,r}T_r} + \frac{160c_0}{\rho_y\eta_{y,r}^2 T_r}(\sigma_{r,0}^x + \sigma_{r,0}^y + 56\sigma_{r,0}^h) + 330c_0\rho_y\eta_{y,r}^2 L_\beta^2\sigma^2\right)$$

$$\leq \frac{1}{\mu}\left(\frac{20c_0\mathcal{V}_{r,0}}{\eta_{y,r}T_r} + \frac{320c_0}{\rho_y\eta_{y,r}^2 T_r}\left(\sigma_{r,0}^x + \sigma_{r,0}^y + 56\sigma_{r,0}^h\right) + 338c_0\rho_y\eta_{y,r}^2 L_\beta^2\sigma^2\right) . \tag{149}$$

Therefore, when $r = 0$, we have $\sigma_{0,0}^x + \sigma_{0,0}^y + 56\sigma_{0,0}^h = 58L_\beta^2\sigma^2$. Based on Eq. (149) and Lemma C.7, we have

$$\sigma_{1,0}^x + \sigma_{1,0}^y + 56\sigma_{1,0}^h \leq \frac{18560c_0 L_\beta^2}{\rho_y\eta_{y,0}^2 T_0}\sigma^2 + \frac{20c_0\mathcal{V}_{0,0}}{\eta_{y,0}T_0} + 338c_0\rho_y\eta_{y,0}^2 L_\beta^2\sigma^2 ,$$

$$\mathcal{V}_{1,0} \leq \frac{1}{\mu}\left(\frac{18560c_0 L_\beta^2}{\rho_y\eta_{y,0}^2 R_0}\sigma^2 + \frac{20c_0\mathcal{V}_{0,0}}{\eta_{y,0}R_0} + 338c_0\rho_y\eta_{y,0}^2 L_\beta^2\sigma^2\right) . \tag{150}$$

When $r = 0$, by setting $\eta_{y,0} = \frac{1}{30L_\beta}$ and $R_0 = \max\{225, \frac{16\mathcal{V}_{0,0}}{L_\beta\sigma^2}\}$, we have

$$\sigma_{1,0}^x + \sigma_{1,0}^y + 56\sigma_{1,0}^h \leq \frac{18560 \times 45c_0 L_\beta^2}{32R_0}\sigma^2 + \frac{20 \times 30c_0 L_\beta\mathcal{V}_{0,0}}{R_0} + 338c_0 L_\beta^2\sigma^2 \leq 500c_0 L_\beta^2\sigma^2 ,$$

$$\mathcal{V}_{1,0} \leq \frac{500c_0 L_\beta^2\sigma^2}{\mu} , \tag{151}$$

where the second step holds due to $\rho_y = 640L_\beta^2$.

Therefore, we denote $\epsilon_1 \triangleq 500c_0 L_\beta^2\sigma^2/\mu$ such that

$$\sigma_{1,0}^x + \sigma_{1,0}^y + 56\sigma_{1,0}^h \leq \mu\epsilon_1, \ \mathcal{V}_{1,0} \leq \epsilon_1 . \tag{152}$$

In the following, we use the inductive approach to prove the desired result. Specifically, suppose $\sigma_{r,0}^x + \sigma_{r,0}^y + 56\sigma_{r,0}^h \leq \mu\epsilon_r$ and $\mathcal{V}_{r,0} \leq \epsilon_r$, we will prove $\sigma_{r+1,0}^x + \sigma_{r+1,0}^y + 56\sigma_{r+1,0}^h \leq \mu\epsilon_r/2$ and $\mathcal{V}_{r+1,0} \leq \epsilon_r/2$. At first, we have

$$\sigma_{r+1,0}^x + \sigma_{r+1,0}^y + 56\sigma_{r+1,0}^h$$

$$\leq \frac{320c_0 L_\beta^2}{\rho_y\eta_{y,r}^2 T_r}\mu\epsilon_r + \frac{20c_0}{\eta_{y,r}T_r}\epsilon_r + 338c_0\rho_y\eta_{y,r}^2 L_\beta^2\sigma^2 . \tag{153}$$

To make $\sigma_{r+1,0}^x + \sigma_{r+1,0}^y + 56\sigma_{r+1,0}^h \leq \mu\epsilon_r/2$, we enforce each term to be smaller than $\epsilon_r/6$. In particular, by setting

$$338c_0\rho_y\eta_{y,r}^2 L_\beta^2\sigma^2 \leq \frac{\mu\epsilon_r}{6} , \tag{154}$$

we set

$$338c_0 640L_\beta^2\eta_{y,r}^2 L_\beta^2\sigma^2 \leq \frac{\mu\epsilon_r}{6} ,$$

$$\eta_{y,r} = \frac{\sqrt{\mu\epsilon_r}}{1140\sqrt{c_0}L_\beta^2\sigma} . \tag{155}$$

It is easy to verify that $\rho_y\eta_{y,r}^2 < 1$ for $t \geq 1$.

By setting

$$\frac{20c_0}{\eta_{y,r}T_r}\epsilon_r \le \frac{\mu\epsilon_r}{6} \;, \tag{156}$$

we get

$$T_r \ge \frac{120 \times 1140 c_0 L_\beta^2 \sqrt{c_0}\sigma}{\mu\sqrt{\mu\epsilon_t}} \;. \tag{157}$$

By setting

$$\frac{320c_0 L_\beta^2}{\rho_y \eta_{y,r}^2 T_r}\mu\epsilon_r \le \frac{\mu\epsilon_r}{6} \;, \tag{158}$$

we get

$$T_r \ge \frac{570 \times 1140 c_0^2 L_\beta^4 \sigma^2}{\mu\epsilon_t} \;. \tag{159}$$

Therefore, by setting $\eta_{y,r} = \frac{\sqrt{\mu\epsilon_r}}{1140\sqrt{c_0}L_\beta^2\sigma}$ and $T_r = O(\frac{c_0 L_\beta^2 \sqrt{c_0}\sigma}{\mu\sqrt{\mu\epsilon_r}} \bigvee \frac{c_0^2 L_\beta^4 \sigma^2}{\mu\epsilon_r})$, we get

$$\sigma_{r+1,0}^x + 56\sigma_{r+1,0}^y + \sigma_{r+1,0}^h \le \frac{\mu\epsilon_r}{6} + \frac{\mu\epsilon_r}{6} + \frac{\mu\epsilon_r}{6} \le \frac{\mu\epsilon_r}{2} \tag{160}$$

and

$$\mathcal{V}_{r+1,0} \le \frac{1}{\mu}\left(\frac{320c_0 L_\beta^2}{\rho_y \eta_{y,r}^2 T_r}\mu\epsilon_r + \frac{20c_0^2}{\eta_{y,r}T_r}\epsilon_r + 338c_0\rho_y\eta_{y,r}L_\beta^2\sigma^2\right) \le \frac{\epsilon_r}{2} \;. \tag{161}$$

Since $\epsilon_r = \frac{\epsilon_1}{2^{r-1}} = \frac{500c_0 L_\beta^2 \sigma^2}{2^{r-1}\mu}$, we get

$$\frac{c_0^2 L_\beta^4 \sigma^2}{\mu\epsilon_r} = \frac{c_0^2 L_\beta^4 \sigma^2}{\mu} \times \frac{2^{r-1}\mu}{c_0 L_\beta^2 \sigma^2} = L_\beta^2 c_0 \times 2^{r-1} \;,$$

$$\frac{c_0 L_\beta^2 \sqrt{c_0}\sigma}{\mu\sqrt{\mu\epsilon_r}} = \frac{c_0 L_\beta^2 \sqrt{c_0}\sigma}{\mu\sqrt{\mu}} \times \frac{\sqrt{2^{r-1}\mu}}{\sqrt{c_0}L_\beta\sigma} = \frac{c_0 L_\beta}{\mu} \times \sqrt{2^{r-1}} \le \frac{c_0 L_\beta}{\mu} \times 2^{r-1} \;. \tag{162}$$

Therefore, we set $T_r = O(\frac{c_0}{\mu} \times 2^{r-1})$. Finally, to achieve $\mathcal{V}_{R,0} \le \epsilon$, we need $\frac{\epsilon_1}{2^{(R-1)}} = \epsilon$ so that $R = \log_2 \frac{2\epsilon_1}{\epsilon}$. As such, the total number of iterations is

$$O(T_0 + \sum_{r=1}^{R} T_r) = O(\max\{225, \frac{16\mathcal{V}_{0,0}}{L_\beta\sigma^2}\} + \sum_{r=1}^{R} \frac{c_0}{\mu} \times 2^{r-1}) = O\left(\frac{c_0\epsilon_1}{\mu\epsilon}\right) = O\left(\frac{\kappa^6}{\epsilon}\right) \;, \tag{163}$$

where the second step holds due to

$$\sum_{r=1}^{R} 2^{(r-1)} = \frac{c_0}{\mu}\frac{2^R - 1}{2 - 1} = O\left(\frac{c_0}{\mu}2^{\log_2 \frac{2\epsilon_1}{\epsilon}}\right) = O\left(\frac{c_0\epsilon_1}{\mu\epsilon}\right) \;. \tag{164}$$

Moreover, we get

$$\eta_{y,r} = \frac{\sqrt{\mu\epsilon_r}}{1140\sqrt{c_0}L_\beta^2\sigma} = \frac{\sqrt{\mu}}{1140\sqrt{c_0}L_\beta^2\sigma}\sqrt{\frac{500c_0 L_\beta^2 \sigma^2}{2^{r-1}\mu}} = O(1/\sqrt{2^{r-1}}L_\beta) \;, \tag{165}$$

and it is easy to know $\eta_{x,r} = O(\mu^2/\sqrt{2^{r-1}}L_\beta)$.

$\square$

## C.4 Proof of the Theorem 5.1

*Proof.* Because $\hat{f}(G(x), y)$ is strongly convex with respect to $x$ and satisfies the PL condition with respect to $y$, we have

$$\mathbb{E}[\|x_{\tilde{R}} - x^*\|^2] \leq \frac{2}{\ell}\mathbb{E}[\hat{\Phi}(x_{\tilde{R}}) - \hat{\Phi}(x^*)] , \tag{166}$$

where we set $\omega = 2\ell$ such that $\hat{f}(G(x), y)$ is $\ell$-strongly convex with respect to $x$, and we define $\hat{\Phi}(x) = \max_y \hat{f}(G(x), y^*)$ with $y^* = \arg\max_{y \in \mathbb{R}^{d_y}} \hat{f}(G(x), y)$. Then, according to Proposition 2.1 in [30], to guarantee $\mathbb{E}[\|x_{\tilde{R}} - x^*\|^2] \leq O(\epsilon^2)$ such that $\mathbb{E}[\|\nabla\Phi(\tilde{x}_R)\|^2] \leq O(\epsilon^2)$, we can enforce $\mathbb{E}[\hat{\Phi}(x_{\tilde{R}}) - \hat{\Phi}(x^*)] \leq O(\epsilon^2)$. Then, from Theorem C.2, it is easy to see that after running Algorithm 2 for the total number of iterations $O(1/\epsilon^2)$ (Note that $1/\epsilon$ is usually large in practice [30] so that we omit other factors.), we have $\mathbb{E}[\|x_{\tilde{R}} - x^*\|^2] \leq O(\epsilon^2)$ and then $\mathbb{E}[\|\nabla\Phi(\tilde{x}_R)\|^2] \leq O(\epsilon^2)$. $\quad\square$

