# OpenReview forum: "On the Convergence of Stochastic Smoothed Multi-Level Compositional Gradient Descent Ascent"
_NeurIPS.cc/2025/Conference — NeurIPS 2025 poster_

### Official Review · Reviewer_g7hY · 2025-06-29

**Clarity:** 2
**Significance:** 2
**Originality:** 2
**Rating:** 4
**Confidence:** 2

**Summary:**

This paper studies the stochastic multi-level compositional minimax optimization problem, where the objective function has a multi-level compositional structure.

The authors propose a variance-reduced algorithm that incorporates smoothing techniques and provide a convergence analysis under a non-convex PL condition.

Theoretical analysis demonstrates improvements in the dependency on the condition number and the convergence rate, compared to existing methods.

The experiments cover key applications, such as deep AUC maximization and multi-instance learning.

**Questions:**

Questions
1. How would a smaller batch size affect the algorithm's performance? Have the authors considered any techniques to mitigate this issue to make the algorithm more suitable for large-scale training scenarios?

2. Could the authors provide a more detailed comparison of the per-iteration computation time and the overall training time between the Smoothed-SMCGDA-VR algorithm and the baseline methods?

3. Have the authors conducted experiments to investigate how the algorithm's performance scales with the number of compositional levels, K?

**Ethical Concerns:**

["NO or VERY MINOR ethics concerns only"]

**Final Justification:**

My main concerns have been resolved.

**Limitations:**

Yes

**Quality:**

2

**Strengths And Weaknesses:**

**Strengths**:

1. The proposed Smoothed-SMCGDA-VR algorithm integrates stochastic smoothing with **STORM-like variance reduction techniques**, offering an effective solution to the biased gradient issue in the multi-level compositional minimax optimization problem.

2. The paper presents a thorough convergence analysis for the proposed algorithm, demonstrating that the method improves the dependency on the condition number.

3. Experiments on deep AUC maximization and multi-instance learning are conducted, showcasing the superior performance of the proposed algorithm.

**Weaknesses**:

1. The proposed method is not a mini-batch algorithm. The convergence result in Theorem 4.1 requires the batch size $S$ to satisfy a sufficiently large condition with $B = \mathcal{O}(1/\epsilon)$ in order to achieve the claimed convergence rate. This may necessitate a large batch size, which could become a bottleneck in practical applications.

2. The iteration complexity of the proposed algorithm is $O(\kappa^{1/2}\epsilon^{-3})$, which shows only a limited improvement over the most representative algorithms, whose iteration complexity is $O(\kappa^{3}\epsilon^{-3})$, especially when $\kappa$ is close to 1. Moreover, the algorithm’s fast convergence seems to rely on a relatively large mini-batch size $B = \mathcal{O}(1/\epsilon)$ as a trade-off.

3. The algorithm presented is a stochastic extension of the methods in [R1,R2], where the authors use the component-wise smoothness of the objective function $f(\cdot,\cdot)$ and variance-reduction technique used in [R3] to achieve an iteration complexity of $O(\epsilon^{-3})$. The notation in this paper is highly complex, and the conclusions appear predictable, making the paper less engaging. It is unclear where the core challenge and primary novelty of the work lie.

4. The pseudocode for Algorithm 1 (Smoothed-SMCGDA-VR) involves numerous hyperparameters that require manual tuning. It would be more informative to explicitly present their roles and suggested values to guide implementation.

References

[R1] J. Yang, A. Orvieto, A. Lucchi, and N. He. Faster single-loop algorithms for minimax optimization without strong concavity. AISTATS 2022.

[R2] T. Zheng, L. Zhu, A. So, J. Blanchet, J. Li. Universal Gradient Descent Ascent Method for Nonconvex-Nonconcave Minimax Optimization, NeurIPS 2023.

[R3] A. Cutkosky and F. Orabona. Momentum-based variance reduction in non-convex sgd, NeurIPS 2019.

---

> ### Author Rebuttal · Authors · 2025-07-26
>
> Thank you for carefully reading our paper.  The reviewer's questions can be classified two categories: **misunderstanding**  and **clarification of specific details.** We appreciate your feedback and address your questions below.
>
> ---
>
> ## **I. Misunderstanding**
>
>
>
> **Mini-batch:** Our proposed algorithm is a mini-batch algorithm with batch size $S=O(1)$. The larger batch size **$B=O(1/\epsilon)$ is used only in the initial iteration**, when $t=0$. Throughout the remainder of training ($ t \in \\{1, \cdots, T-1\\}$), the batch size remains $O(1)$, which is consistent with the STORM algorithm[7]. Thus, our algorithm is suitable for large-scale training scenarios. We will make this point clearer in the revised version to avoid potential misunderstanding.
>
>
>
>
>
>
>
>
> ---
>
> ## **II. Clarification of specific details**
>
> **Iteration complexity:** First, we would like to highlight that $O(\kappa^2\epsilon^{-3})$ is obtained for the **two-level** problem, rather than the **multi-level** compositional minimax problem.  The multi-level structure brings more challenges for convergence analysis. Therefore, the improvement in the dependence on $\kappa$ is non trivial from an analytical perspective, particularly given that our work addresses a new problem class: stochastic **multi-level** compositional minimax optimization. Second, the condition number $\kappa$ is typically large for practical machine learning applications. Therefore, the improvement in terms of $\kappa$ is significant.
>
> **Core challenge and primary novelty:** Our algorithm is not a straightforward extension of [R1, R2]. Those works consider only standard minimax problems and do not encounter the gradient bias issue during updates. In contrast,
>
> * the **multi-level structure** and the **compositional structure** in our problem introduce accumulated bias across levels, which poses a significant challenge;
>
> * another challenge lies in **relating two different stationary measures in the multi-level compositional minimax setting**.
>
> These challenges are discussed in detail in lines 55–67 and Section 3.3.  The novelty of our work lies in how we address these challenges, through both a novel algorithm and improved convergence analysis, as summarized in lines 68–79. In particular, we developed a novel Algorithm 2  with a stage-wise structure for translating different stationary measures, which has never been studied in [R1-R3]. This is an important novelty and contribution of our paper.
>
> **Hyperparameters:**
>
> - **$\eta$**: Learning rate.
> - **$\omega$**: Used in the smoothing term.
> - **$\gamma_x, \gamma_y, \gamma_z$**: Coefficients for learning rate updates.
> - **$\rho_x, \rho_y$**: Coefficients for momentum updates.
>
> In practice, we first tune $\eta$ and $\omega$. $\omega$ should not be set too large to avoid deviating too far from the original problem. The coefficient $\gamma$ can be fixed to constants. Once $\eta$ is selected, tuning $\rho$ is straightforward since $\rho\eta^2 \in (0, 1)$.
>
> **Experiments:** We have included experiments with the number of compositional levels $k$, as shown in Figure 4. Additional comparisons on per-iteration computation time and the overall training time can be added in the revised version.
>
> ---
> It appears that **the main concern lies in the mini-batch setting, which may stem from a misunderstanding**. We believe all the reviewer's questions are minor and can be easily addressed, hence not affecting the technical contribution of our paper.  If our clarification resolves reviewer's questions, we kindly ask the reviewer to reconsider the evaluation.

---

> > ### Author Response · Authors · 2025-08-02
> >
> > Dear Reviewer g7hY,
> >
> > Thank you for carefully reviewing our paper and rebuttal.
> >
> > If you have any further questions, we would be happy to address them.
> >
> > Best regards,
> >
> > Authors of 18077

---

> > > ### Author Response · Authors · 2025-08-05
> > >
> > > Dear Reviewer g7hY,
> > >
> > > Thank you again for carefully reviewing our paper and rebuttal.
> > >
> > > We kindly would like to know if you have more questions for our paper.  If our clarification resolves all your questions, we would be truely grateful if you could consider a higher rating. Thanks again!
> > >
> > > Best regards,
> > >
> > > Authors of 18077

---

> ### Comment · Reviewer_g7hY · 2025-08-07
> **Mini-batch Algorithm?**
>
> The authors reply that: We set the mini-batch size in iteration $t$ as
> $$S^{t} = O(1/\varepsilon), ~~~ \text{ if $t=0$};$$
> $$S^{t} = O(1), ~~~ \text{ if $t=\{1,2,3,...\}$}.$$
>
>
>
> ---
>
> 1. From an implementation standpoint, such a **drastic swing in batch size** from one iteration to the next is highly unusual and raises practical concerns. I am therefore not persuaded by the authors’ claim.
>
> 2. The authors adopt a **uniform step size** $\eta^t = O(\varepsilon)$ **for all $t$**. This choice necessitates
>
> $$||p_t - \nabla_x f_{\omega} ( H(x_t),y_t; z_t )||^{2} \le O(\sigma^{2} \varepsilon)$$
>
> $$|| q_t - \nabla_y f_{\omega} (H(x_t),y_t;z_t )||^{2} \le O(\sigma^{2}\varepsilon)$$
>
> and these inequalities must hold **for every iteration $t$**.
>
> 3. The authors could instead adopt a diminishing schedule such as $\eta^t = O(1/t)$ and $S^t=O(t)$, which would relax the required variance bounds to
>
> $$||p_t - \nabla_x f_{\omega} ( H(x_t),y_t; z_t )||^{2} \le O(\sigma^{2} /t)$$
>
> $$|| q_t - \nabla_y f_{\omega} (H(x_t),y_t;z_t )||^{2} \le O(\sigma^{2}  /t)$$
>
> 4. Ultimately, the algorithm must specify a coherent policy for choosing $\eta^{t}$, $S^{t}$ (mini-batch size), and the total number of iterations $T$. No matter which schedule is selected, the required batch sizes $S^{t}$ still grow with either $1/\varepsilon$ or $t$, and Algorithm 1 is not a min-batch algorithm.

---

> > ### Author Response · Authors · 2025-08-07
> >
> > To further illustrate that STORM is a mini-batch algorithm, we provide several published works where STORM is used. In all these papers, the authors explicitly state that STORM operates with a mini-batch size of $O(1)$.
> >
> > STORM with constant learning rate: using an inital batch size of $O(1/\epsilon)$.
> >
> > [1]Wu, Xidong, et al. "Solving a class of non-convex minimax optimization in federated learning." Advances in Neural Information Processing Systems 36 (2023): 11232-11245.
> >
> > [2]Xian, Wenhan, et al. "A faster decentralized algorithm for nonconvex minimax problems." Advances in Neural Information Processing Systems 34 (2021): 25865-25877.
> >
> >
> > STORM with diminishing learning rate: with a factor $\ln(T)$ in their proof.
> >
> > [3] Jiang, Wei, et al. "Optimal algorithms for stochastic multi-level compositional optimization." International Conference on Machine Learning. PMLR, 2022.
> >
> > [4] Liu, Jin, et al. "Faster stochastic variance reduction methods for compositional minimax optimization." Proceedings of the AAAI Conference on Artificial Intelligence. Vol. 38. No. 12. 2024.
> >
> > We hope the reviewer could let us know whether these additional clarifications help resolve the concern. Thank you very much!

---

> > > ### Comment · Reviewer_g7hY · 2025-08-07
> > >
> > > The original STORM algorithm never relies on a full-batch gradient at any iteration. Accordingly, the assertion in the authors' reply that “Using a different setting for the batch size of the initial step and other steps is a property of STORM variance-reduced gradient estimator” is not true.
> > >
> > > Could the authors clarify why choosing the mini-batch size $S = O(1/\varepsilon)$ in the first iteration is critical to the convergence analysis?
> > >
> > > Moreover, is it possible to relax—or even eliminate—this requirement in the proof?

---

> ### Author Response · Authors · 2025-08-07
>
> Thank you for your response.
>
> The reviewer may have an understanding of the variance reduction approach. Our algorithm uses the STORM variance-reduced gradient estimator [1]. This gradient estimator has been studied for many years, and it is well known that it only requires $O(1)$ batch size when $t>0$.  The batch size does NOT grow with $1/\epsilon$ or $t$.
>
> ---
>
> Response to 1: Using a different setting for the batch size of the initial step and other steps is a property of STORM variance-reduced gradient estimator. It has been used for many other papers relying on STORM. From an implementation standpoint, the user can also use $O(1)$ batch size in the initialization, it does not hurt the practical convergence performance.
>
> [1]. Momentum-Based Variance Reduction in Non-Convex SGD
>
> ---
>
> Response to 2: Our convergence analysis does **NOT require** those two inequalities. **We would appreciate it if the reviewer could specify where in our proof those two inequalities are required.**
>
> In our proof, our Lemma B.6 bounds $\mathbb{E}[\|{p}\_{t+1} -  \nabla\_x f\_{\omega}(H({x}\_{t+1}), y\_{t+1};z\_{t+1}) \|^2]$. There are many terms in its upper bound, which is in the following form:
>
> $\mathbb{E}[\|{p}\_{t+1} -  \nabla_x f\_{\omega}(H({x}\_{t+1}), y\_{t+1};z\_{t+1}) \|^2] \leq   T\_1 + T\_2+ T\_3 + T\_4+  4C\_p^2\alpha^2\eta^4  \sum\_{k=1}^{K}\sum\_{j=1}^{k}(2 C\_g^2)^{j-1}\sigma^2 + 2\rho\_x^2\eta^4C\_p^2\sigma^2$
>
> where
>
> $T\_1=(1-\rho_x\eta^2)\mathbb{E}[\|  p\_{t}- \nabla\_x f\_{\omega}(H({x}\_{t}), y\_{t};z\_{t})  \|^2]$
>
> $T\_2=4C\_p^2 \alpha^2\eta^4\sum\_{k=1}^{K} \Big( \sum\_{j=k}^{K}(2 C\_g^2)^{j-k}\Big)\mathbb{E}[\|{h}\_{t}^{(k)}  - g^{(k)}({h}\_{t}^{(k-1)}) \|^2]$
>
> $T\_3=  2C\_p^2 2\alpha^2\eta^4\sum\_{k=1}^{K} \Big( \sum\_{j=k}^{K}(2 C\_g^2)^{j-k}\Big)\mathbb{E}[\|{h}\_{t}^{(k)}  - g^{(k)}({h}\_{t}^{(k-1)}) \|^2]$
>
> $T\_4= 2C\_p^2\sum\_{k=0}^{K}(2 C\_g^2)^{k}\gamma\_x^2\eta^2\mathbb{E}[\| {p}\_{t}\|^2]$
>
> **Note that all terms in the upper bound are positive**. From this upper bound, we **cannot obtain** that $\mathbb{E}[\|{p}\_{t+1} -  \nabla_x f\_{\omega}(H({x}\_{t+1}), y\_{t+1};z\_{t+1}) \|^2] \leq O(\sigma^2\epsilon)$.
>
> Let's consider **a counterexample for your inequalities**, when $\mathbb{E}[\|{p}\_{t+1} -  \nabla_x f\_{\omega}(H({x}\_{t+1}), y\_{t+1};z\_{t+1}) \|^2] =10$, $T\_1=10$,    $T\_2=10$, $T\_3=10$, $T\_4=10$, and $O(\sigma^2\epsilon)=1$, **the inequality in our Lemma B.6 still holds, but we cannot get the inequality that the reviewer mentioned**:
> $\mathbb{E}[\|{p}\_{t+1} -  \nabla_x f\_{\omega}(H({x}\_{t+1}), y\_{t+1};z\_{t+1}) \|^2] \leq O(\sigma^2\epsilon)$
>
> ---
>
> Response to 3: For STORM gradient estimator, both the constant learning rate and the diminishing learning rate can be used. However, we would like to emphasize **the diminishing learning rate leads to a loose upper bound**.  As shown in its Theorem 1 of STORM, the convergence upper bound has a factor $\ln(T)$ on its numerator. On the contrary, in our paper, we use a constant learning rate, there does not exist such a factor.
>
> ---
>
> Response to 4: In 2 and 3, the reviewer discussed how the learning rate affects those upper bounds by using some incorrect inequalities. Therefore, **the reviewer's conclusion that the batch size grows with $t$ or $\epsilon$ is not true**.
>
> In addition, it would be good if the reviewer could take a look at our proof to verify what batch size is used in our proof. Specifically, in the proof of Lemma B.6, we use a batch size $B=1$ to get the last step from the last to second step. Specifically, $2 \rho\_x^2\eta^4\mathbb{E}[\| \nabla\_x f\_{\omega}({H}(x\_{t}), y\_{t};z\_{t}; \hat{\xi}\_{t+1}) - \nabla\_x f\_{\omega}({H}(x\_{t}), y\_{t};z\_{t})\|^2] \leq 2\rho\_x^2\eta^4C\_p^2\sigma^2$. If we use other batch sizes $B$, this step should be
> $2 \rho\_x^2\eta^4\mathbb{E}[\| \nabla\_x f\_{\omega}({H}(x\_{t}), y\_{t};z\_{t}; \hat{\xi}\_{t+1}) - \nabla\_x f\_{\omega}({H}(x\_{t}), y\_{t};z\_{t})\|^2] \leq 2\rho\_x^2\eta^4C\_p^2\textcolor{red}{\frac{\sigma^2}{B}}$.
>
> In summary, our algorithm use a constant batch size $O(1)$ in $t>0$ iterations.
>
> We hope this response helps clarify the reviewer's misunderstanding. If our clarification addresses all of your concerns, we would be truly grateful if you could consider a higher rating. Thank you again!

---

> ### Author Response · Authors · 2025-08-07
>
> Thank you for your new insightful questions.
>
> ---
>
> As we mentioned in the prior rebuttal, STORM can use a constant learning rate or a diminishing learning rate. In the original STORM paper, a diminishing learning rate is used at the cost of loose upper bound, i.e., introducing the factor $O(\ln T)$.  When the constant learning rate is used, we can get a smaller upper bound, i.e., removing the factor $O(\ln T)$, but at the cost of a large batch size in the initialization.
>
> The reason for the batch size in the initialization is due to the coefficient of the the potential function. Specifically, in Eq.(101), the coefficient of $\mathbb{E}[\| p\_{0} - \nabla\_x f\_{\omega}(H(x\_{0}), y\_{0}; z\_{0})  \|^2] $ is $\frac{\gamma_x}{\rho_x\eta}$, which depends on $\frac{1}{\eta}$. As a result, when putting the initial potential function into the convergence upper bound, this coefficient leads to $O\Big(\frac{\kappa\sigma^2}{\rho_x\eta^2TS} \Big)$ as shown in Eq. (102), which has a higher order dependence on $\frac{1}{\eta}$ than other terms. As a result, we need  a large batch size  $S$ in the initialization step to cancel out the high-order dependence.
>
> ---
>
> To relax the requirement for the initial batch size, we can consider an another convergence criteria, which is an upper bound of the gradient norm. For example, we can use the convergence metric in Page 9 of [1]. However, we don't think this approach truly address this problem. It just hides this requirement by providing a loose upper bound.
>
> Moreover, the reviewer could take a look at the proof of Theorem 5.2 and Theorem 5.3 of [1]. It shows how the diminishing learning rate and the constant learning affect the convergence upper bound of the STORM gradient estimator.
>
> [1] Single-Timescale Stochastic Nonconvex-Concave Optimization for Smooth Nonlinear TD Learning
>
> ---
>
> We hope the reviewer could let us know whether these additional clarifications help resolve the concern. Thank you very much!
>
> If necessary, we are glad to provide the detailed proof of the STORM method when using a constant learning rate in the rebuttal to clarify the reviewer's concern.

---

> ### Author Response · Authors · 2025-08-07
>
> We provide **the proof of STORM when using a constant learning rate**. In the following, we use the symbols of the original STORM paper.  Note that we use a constant learning rate in this proof, i.e., $\eta_t=\eta$.
>
> **Step 1**: Based on the smoothness of the loss function, it is easy to know $\mathbb{E}[F\left({x}\_{t+1}\right)]  \leq \mathbb{E}[F\left({x}\_t\right)-\frac{\eta}{2}\left\|\nabla F\left({x}\_t\right)\right\|^2-\frac{ \eta}{4}\left\|{d}\_t\right\|^2+\frac{\eta}{2}\left\|{\epsilon}\_t\right\|^2]$.
>
> **Step 2**: Based on the first six steps in the last inequality of Lemma 2 of STORM, we can get $\mathbb{E}  [\|{\epsilon}\_t\|^2]\leq \mathbb{E}[\left(1-c\eta^2\right)^2\left\|{\epsilon}\_{t-1}\right\|^2 +2\left(1-c\eta^2\right)^2 L^2 \eta^2\left\|{d}\_{t-1}\right\|^2+ 2 c^2 \eta^4 G^2]$.
>
>
> By setting $\eta< \frac{1}{\sqrt{c}}$ such that $c\eta^2<1$, we can get
>
> $\mathbb{E}  [\|{\epsilon}\_t\|^2]\leq \mathbb{E}[(1-c\eta^2)\left\|{\epsilon}\_{t-1}\right\|^2 +2L^2 \eta^2\left\|{d}\_{t-1}\right\|^2+ 2 c^2 \eta^4 G^2]$.
>
> **Step 3**: By defining the potential function $H\_{t} =\mathbb{E}[F({x}\_{t})]  + \frac{1}{2c\eta}\mathbb{E}  [\|{\epsilon}\_t\|^2] $ and setting $c=4L^2$, it is easy to get
>
> $H\_{t+1}-H\_{t}\leq  \mathbb{E}[-\frac{\eta}{2}\left\|\nabla F\left({x}\_t\right)\right\|^2-\frac{ \eta}{4}\left\|{d}\_t\right\|^2+\frac{\eta}{2}\left\|{\epsilon}\_t\right\|^2] + \mathbb{E}[-\frac{\eta}{2}\left\|{\epsilon}\_{t}\right\|^2 +\frac{ \eta}{8}\left\|{d}\_{t}\right\|^2+  4L^2 \eta^3 G^2]\leq  \mathbb{E}[-\frac{\eta}{2}\left\|\nabla F\left({x}\_t\right)\right\|^2+  4L^2 \eta^3 G^2] $.
>
> As a result, we can get
>
>  $\frac{1}{T}\sum\_{t=0}^{T-1}\left\|\nabla F\left({x}\_t\right)\right\|^2\leq \frac{2(H\_{0}-H\_{T})}{\eta T} + 8\eta^2 L^2  G^2$.
>
> **Step 4**: Let's consider the potential function when $t=0$:
>
> $H_0=\mathbb{E}[F({x}\_{0})]  + \frac{1}{8L^2\eta}\mathbb{E}  [\|{\epsilon}\_0\|^2] =  \mathbb{E}[F({x}\_{0})]  + \frac{1}{8L^2\eta}\frac{\sigma^2}{S}$, where $S$ is the batch size in the initialization step.
>
> Then, we can get
>
>  $\frac{1}{T}\sum\_{t=0}^{T-1}\left\|\nabla F\left({x}\_t\right)\right\|^2\leq  \frac{2(H\_{0}-H\_{T})}{\eta T} + 8\eta^2 L^2  G^2 \leq \frac{2H\_{0}}{\eta T} + 8\eta^2 L^2  G^2 \leq \frac{2F(x\_{0})}{\eta T} + \frac{2}{\eta T} \frac{1}{8L^2\eta}\frac{\sigma^2}{S} + 8\eta^2 L^2  G^2$.
>
> By setting $\eta=1/T^{1/3}$ and $S=T^{1/3}$, we can know that  $\frac{1}{T}\sum_{t=0}^{T-1}\left\|\nabla F\left(\boldsymbol{x}_t\right)\right\|^2\leq  O(1/T^{2/3})$. **It is worth noting that there does not exist a factor $O(\ln T)$ in the convergence upper bound when using a constant learning rate**. Moreover, from this proof, we can know that the large batch size $S$ is only used when $t=0$, and the $O(1)$ batch size is used in other iterations.
>
> ---
>
> This proof shows how the batch size in the initialization affects the convergence rate. Hope this can clarify the reviewer's concern.

---

> > ### Comment · Reviewer_g7hY · 2025-08-08
> >
> > Thank you for the detailed reply; my concerns have been resolved, so I’m willing to increase my rating to 4.

---

### Official Review · Reviewer_GHtS · 2025-06-29

**Clarity:** 2
**Significance:** 3
**Originality:** 2
**Rating:** 4
**Confidence:** 2

**Summary:**

The paper discusses the problem of multi-level compositional optimization and proposes a smoothed variance-reduced algorithm that finds a $(\epsilon,\epsilon/\sqrt \kappa)$-stationary point with $O(\kappa ^{3/2})$ iterations.

**Questions:**

1. The definition of $\omega$ is unclear. Is it an algorithm parameter? If so, how is it chosen? If not, the bound should depend on $\omega$. In addition, in line 582, the bound should be multiplied by $\omega$, which appears in the third term of the $\max$. Moreover, the proof assumes $\omega \ge L_f$, but choosing $\omega$ too large would worsen the result because $f_\omega$ diverges from $f$. How is this fact expressed in the proof or the algorithm?

2. Could the authors elaborate on where the improvement in the result comes from? Is it due to a better algorithm, a sharper analysis, or both?

**Ethical Concerns:**

["NO or VERY MINOR ethics concerns only"]

**Final Justification:**

My concern regarding the transition from line 580 to line 582 has been addressed. However, I still found the proofs difficult to follow and believe they would benefit from further clarification. Given that the authors intend to include a proof sketch and clarify the transitions, and since I was unable to fully follow or verify the original proofs, I am increasing my score to 4 while maintaining low confidence.

**Limitations:**

Yes.

**Paper Formatting Concerns:**

N.A

**Quality:**

2

**Strengths And Weaknesses:**

Strengths

1. The paper is well written and presents its motivation clearly.

2. The results improve the bounds achieved by previous work, even for the case $k = 2$.

3. The authors clearly discuss the challenges they tackled while solving the problem.

4. The authors validated their results experimentally.

Weaknesses

1. My major concern is about the correctness of the bound. . In the transition from line 580 to line 582, the term $H_0$ should be multiplied by $\kappa/\eta$ (the second term in the $\max$), shouldn't it? If so, this would weaken the bound stated in the paper. Could the authors clarify?  If I’m mistaken, I will revise my score accordingly.

2.  Although the authors describe the difficulties they faced, it would help readers if they also explained how each challenge was overcome and which part of their algorithm or analysis addressed it.

3.  The proofs are hard to follow; many inequalities need further explanation. A proof sketch outlining the key ideas behind each central lemma would be valuable.

4. The formality of the proofs is somewhat lacking. For example, the authors do not specify concrete values for the algorithm’s constants (they provide only $O$-notation). Because there are many constraints on these constants, it is hard to verify that they are mutually consistent.

---

> ### Author Rebuttal · Authors · 2025-07-26
>
> Thank you for carefully reading our paper. The reviewer's questions can be classified two categories: **writing issues** and **clarification of specific details.** We appreciate your feedback and address your questions below.
>
> ---
>
> ## **I. Writing issues**
>
> **Correctness of the bound:** There is a typo in line 580. $
> \frac{\kappa(\mathcal{H}\_0-\mathcal{H}\_T)}{\gamma\_x \eta T}$ should be $
> \frac{\mathcal{H}\_0-\mathcal{H}\_T}{T}$. Specifically, eq.(100) is derived from eq.(98). It is easy to know that the coefficient of $\frac{\mathcal{H}\_0-\mathcal{H}\_T}{T}$ is $
> \frac{8}{\gamma\_x \eta}$, which is moved to the max operator. However, this typo does not affect the final result in eq.(102), as it is easy to verify that the coefficient of $\mathcal{P}_0$ in eq.(102) is correct. Hence, the overall convergence rate remains valid.   Thanks for pointing this out this typo, and we will revise it accordingly.
>
> **Proof:** We provide proof sketch outlining the high-level ideas in line 204-216 for the first algorithm, and line 250-258 for the second. The use of $O$-notation is standard in optimization analysis, and we provide complete proofs alongside it. Additional explanations for each lemma will be included in the appendix to improve clarity.
>
> **The authors do not specify concrete values for the algorithm’s constants:** They have already been provided in our proof. Specifically, Eq.(97) summarizes the upper bound of hyperparameters.
>
> ---
>
> ## **II. Clarification of specific details**
>
>
>
> **Addressing challenges:** We summarize how the challenges are addressed in line 68-79. For the algorithmic challenge, we propose a variance-reduced algorithm based on smoothed techniques. For the theoretical challenges, after obtaining an $O(\epsilon, \sqrt{\epsilon})$-stationary point, we further develop an algorithm under the two-sided PL condition, successfully performing the translation between two stationary measures.
>
>
>
>
>
> **Hyperparameter $\omega$:**  $\omega$ is defined in eq.(4), as a hyperparameter in the smoothing term. In Theorem 4.1, we state that choosing $\omega = O(L_f)$, e.g., $\omega=2L_f$,  is a common practice for the smoothing technique.  Moreover, since $\omega$ is in the same scale as $L_f$, it does not worsen the convergence. In particular, Eq.(97) shows how $\omega$ affects the hyperparameter. It is easy to see that $\omega=2L_f$ does not have a significant influence on them, as well as the convergence rate.
>
> **Elaborate on where the improvement in the result comes from:** It is due to both a better algorithm design and sharper analysis. On the one hand, compared to [9, 13], the improvement is due to the variance-reduced estimator, while the improvement over [23] is due to the smoothing technique. On the other hand, we developed a novel Algorithm 2 and provided sharper analysis to translate the $(\epsilon, \epsilon/\sqrt{\kappa})$-stationary point to $\epsilon$-stationary point. This has never been investigated for compositional miniax optimization, so this is another important contribution of our paper.
>
> ---
>
> If we have addressed the concerns, particularly regarding the correctness of the bound, we kindly ask the reviewer to reconsider the evaluation.

---

> > ### Author Response · Authors · 2025-08-02
> >
> > Dear Reviewer GHtS,
> >
> > Thank you for carefully reviewing our paper and rebuttal.
> >
> > If you have any further questions, we would be happy to address them.
> >
> > Best regards,
> >
> > Authors of 18077

---

> > > ### Comment · Reviewer_GHtS · 2025-08-03
> > >
> > > Thank you for the response. My concern regarding the transition from line 580 to line 582 has been resolved. However, I found the proofs difficult to follow and believe they would benefit from further clarification. That said, I have increased my score by 1.

---

### Official Review · Reviewer_Z4zL · 2025-07-01

**Clarity:** 4
**Significance:** 2
**Originality:** 3
**Rating:** 5
**Confidence:** 3

**Summary:**

## Summary

This paper addresses the **multi-level stochastic compositional minimax optimization** problem, a challenging framework where the objective involves nested expectations and a min-max structure over decision variables. Such settings introduce significant complexities, particularly due to the biased nature of stochastic gradient estimates when applied to both primal and dual variables.

To tackle these difficulties, the authors propose a **stochastic gradient ascent algorithm** tailored for multi-level composition. Their method integrates **variance reduction** and a **smoothing strategy** under the **nonconvex Polyak–Łojasiewicz (PL) condition**. They theoretically establish a convergence guarantee to an $(\epsilon, \epsilon / \sqrt{\kappa})$-stationary point, with iteration complexity scaling as $\mathcal{O}(\kappa^{3/2} / \epsilon^2)$, where $\kappa$ denotes the condition number. Additionally, they design a stage-wise scheme under a two-sided PL condition that enables translation between different notions of stationarity.

While prior work has shown the advantages of smoothing techniques in classical minimax problems, especially in improving convergence under PL-type conditions, their application in **multi-level compositional settings** has remained largely unexplored. This paper fills that gap, showing that such techniques can be effectively adapted to the multi-level setting, achieving both improved theoretical bounds and empirical performance.

**Questions:**

N/A - Please comment on the weakness of the paper

**Ethical Concerns:**

["NO or VERY MINOR ethics concerns only"]

**Limitations:**

yes

**Quality:**

2

**Strengths And Weaknesses:**

## Strengths

- The paper is **clearly written** and well-structured, making the technical contributions accessible even within a complex multi-level stochastic minimax setting.
- It addresses a **concrete and timely gap** in the literature—namely, the challenge of biased stochastic gradients in compositional min-max problems—by introducing a method that is both novel and theoretically justified.
- The proposed algorithm achieves an improved complexity bound in terms of the condition number $\kappa$, specifically $\mathcal{O}(\kappa^{3/2} / \epsilon^2)$, which improves over previous works.
- The incorporation of smoothed techniques into the multi-level compositional setting is original and motivated by recent advances in classical minimax optimization.

## Weaknesses / Questions

- It is not entirely clear whether the proposed algorithm can be **extended to ReLU networks**, where subgradients rather than gradients are used. Would the variance reduction or convergence guarantees still hold under such non-smooth settings?
- A natural question is how the **complexity would change** in the case where the problem satisfies the **two-sided PL condition** (i.e., the function is PL in both $x$ and $y$). Can the current framework be adapted, and would it yield better rates?
- There is some concern regarding **practical relevance**: how far is this multi-level compositional model from real-world applications, such as adversarial training or bilevel learning in neural networks?

---

> ### Author Rebuttal · Authors · 2025-07-26
>
> Thank you for carefully reading our paper.  Regarding questions, they are mainly about **the clarification of specific details**.  We appreciate your feedback and provide detailed clarification for each of them below.
>
> ---
>
> ## **I. Clarification of specific details**
>
> **Extended to ReLU networks:** Extending the proposed algorithm to non-smooth settings using subgradients is a promising direction for future work.
>
> **Two-sided PL condition:**  Thank you for the insightful question. We believe the smoothing framework can be applied to the two-sided PL problem; however, it would require new convergence analysis, and the achievable convergence rate remains unclear. We will consider this promising direction for future work.
>
>
> **Real-world applications:** Multi-level compositional models can be applied to practical scenarios such as hyperparameter optimization (bilevel optimization) and deep AUC maximization (bilevel minimax optimization).

---

> > ### Comment · Reviewer_Z4zL · 2025-08-05
> >
> > Could you provide more details about relu nets

---

> > > ### Author Response · Authors · 2025-08-05
> > >
> > > Thank you for your reply!
> > >
> > > We assume the ReLU network refers to the neural network using ReLU activation function. Because ReLU function is not a smooth function, we cannot use the standard gradient to update model parameters. Instead, we need to use the method designed for the non-smooth function to solve this kind of optimization problem, such as subgradient or Moreau envelop. In fact, there are some recent works trying to optimize the non-smooth compositional minimization problem. For example, [1,2] uses the subgradient method to handle the non-smoothness issue, while [3] relies on the Moreau envelop approach to deal with it. However, it is important to note that these methods focus on the **two-level** compositional **minimization** problem. It is still unclear how to handle the non-smoothness issue for the **multi-level** compositional **minimax** problem, because the multi-level property and the minimax property introduce more challenges. For example, $y^*(x)$ may not be smooth any more, and then $\Phi(x)$ may be ill-defined. We will leave this challenging setting in our future work. Thank you for this insightful question!
> > >
> > > [1] Hu, Quanqi, Dixian Zhu, and Tianbao Yang. "Non-smooth weakly-convex finite-sum coupled compositional optimization." Advances in Neural Information Processing Systems 36 (2023): 5348-5403.
> > >
> > > [2] Zhu, Landi, Mert Gürbüzbalaban, and Andrzej Ruszczyński. "Distributionally robust learning with weakly convex losses: Convergence rates and finite-sample guarantees." arXiv preprint arXiv:2301.06619 (2023).
> > >
> > > [3] Chen, Xingyu, et al. "Stochastic Momentum Methods for Non-smooth Non-Convex Finite-Sum Coupled Compositional Optimization." arXiv preprint arXiv:2506.02504 (2025).

---

### Official Review · Reviewer_bNHb · 2025-07-02

**Clarity:** 3
**Significance:** 3
**Originality:** 3
**Rating:** 5
**Confidence:** 4

**Summary:**

The paper studies "stochastic multi-level compositional minimax" optimisation, a setting that arises in deep AUC maximisation and multi-instance learning.

Two main algorithms are proposed:

1. Smoothed-SMCGDA-VR – a variance-reduced gradient descent ascent method that uses smoothing term and STORM-style recursive estimators at every compositional level. Theorem 4.1 shows it finds an $(\epsilon,\;\epsilon/\sqrt\kappa)$-stationary point in $\tilde O(\kappa^{3/2}/\varepsilon^{3})$ stochastic first-order calls, improving the $\kappa$-dependence over prior two-level results.

2. Stagewise-SMCGDA-VR – a two-phase scheme that, given the first algorithm’s output, solves a smoothed auxiliary game under a two-sided PL condition, converting to a true $\epsilon$-stationary point with an extra $\tilde O(1/\mu^{3}\log(1/\epsilon))$ cost.

**Questions:**

1) Regarding the lines 38 to 42, "Unfortunately, these minimization-targeted algorithms cannot directly address the stochastic multi-level compositional minimax optimization problem in Eq. (1), as the stochastic gradients for both primal and dual variables are biased estimators in stochastic multi-level compositional minimax problems—posing greater algorithmic and theoretical challenges.", do you claim that the challenge in the stochastic multi-level compositional minimax optimization problem is different from the generic min-targeted one? If yes, could you elaborate on that?

2) Regarding lines 55 and 56, "Addressing this question is not straightforward and presents substantial algorithmic and theoretical
challenges." This sentence is not clear to me. Could you explain it more?

3) Is there any application for two-sided PL?

4) I couldn't follow the lines 163 to 176; what is the smoothed algorithm which outputs \tilde{x}? What does it mean a function satisfies two-sided PL WITH iteration complexity O(\eps^{-2})?

5) Have you tried replacing STORM with SPIDER/SARAH updates at each level; if so, how did constants affect performance?

**Ethical Concerns:**

["NO or VERY MINOR ethics concerns only"]

**Final Justification:**

The author’s reply resolves my concerns; I’m keeping my current rating.

**Limitations:**

Yes.

**Paper Formatting Concerns:**

None.

**Quality:**

3

**Strengths And Weaknesses:**

Strengths:

1) Multi-level *minimax* problems are largely unexplored; to me, the bias in both primal and dual stochastic gradients is well motivated.
2) Their theoretical and experimental contributions in this work seem good to me.
3) Improved complexity: The $\kappa$-exponent drops from 3 or 4 in prior work to 3/2, and the conversion phase is provably negligible in cost.


Weaknesses:

While the paper makes a good theoretical step for stochastic multi-level minimax optimisation, its practical impact is circumscribed by strong structural assumptions, heavy hyper-parameter dependence, and sizeable computational demands. Relaxing the PL requirements, developing adaptive parameter-free variants, and broadening empirical validation would substantially strengthen the work.

---

> ### Author Rebuttal · Authors · 2025-07-26
>
> Thank you for carefully reading our paper.  Suggestions such as relaxing PL conditions, developing adaptive parameter-free variants, and exploring more empirical application are insightful direction for future work. Regarding questions, they are mainly about **the clarification of specific details**.  We appreciate your feedback and provide detailed clarification for each of them below.
>
> ---
>
> ## **I. Clarification of specific details**
>
> **Lines 38 to 42:** The challenge in stochastic multi-level compositional minimax optimization differs from that in the standard minimization setting. In the minimization case, there is only one variable, whereas in the minimax setting, biases arise in both the primal and dual variables, making the problem more difficult to handle.
>
> **Line 55 and 56:** We provide an explanation of this sentence in lines 56–65. The challenges lie in two aspects:
>
> * Algorithmic challenge: the stochastic gradients for both primal and dual variables are biased;
>
> * Theoretical challenge: comparing convergence rates requires translating between two different stationarity measures.
>
> A more detailed discussion of these challenges is given in Section 3.3.
>
> **Application for two-sided PL:** The two-sided PL condition is easy to satisfy. Our experiments, such as deep AUC maximization and multi-instance learning when using over-parameterized deep neural networks, satisfy this condition, because [R1] shows that over-parameterized deep neural networks satisfy the PL condition.
>
> [R1] Chaoyue Liu, Libin Zhu, and Mikhail Belkin. Loss landscapes and optimization in over-parameterized non-linear
> systems and neural networks. arXiv preprint arXiv:2003.00307, 2020a.
>
> **Line 163 to 176:** $\tilde{z}$ refers to the updated primal variable $x_{T}$ obtained using a smoothed algorithm. This interpretation follows from Proposition 2.1 in [30], which includes the relevant proof.
>
> **STORM:** STORM is a more practical technique compared to SPIDER/SARAH, as it avoids the need for large batch sizes to compute a checkpoint gradient. This makes STORM more suitable for large-scale applications.

---

> > ### Comment · Reviewer_bNHb · 2025-08-08
> >
> > My concerns have been taken care of by them!

---

### Official Review · Reviewer_7Drm · 2025-07-02

**Clarity:** 1
**Significance:** 2
**Originality:** 3
**Rating:** 4
**Confidence:** 4

**Summary:**

The paper provides a new algorithm and results for the stochastic multi-level compositional minmax problem. The approach includes variance reduction and smoothing techniques for which have yet to be applied to the stochastic case. Stationary point convergence is claimed for different metrics. To get an $\epsilon$-stationary point convergence, from an $(\epsilon_1,\epsilon_2)$ stationary point requires solving a two-sided stochastic PL multi-level problem for which the paper proposes a new method and analysis. The new algorithm is also tested empirically in deep AUC maximization and multi-instance learning.

**Questions:**

Below I include some questions and comments that I think will shed light on my previous concerns as well potentially improve the paper.


+ Lemma B.1 proof seems incomplete, where is the reasoning for property (3)? Also the reference for property (4) seems missing?
+ What are the $\gamma$ constants in the algorithm input or in the bound?
+ Why is there an expectation in in Lemma B.3? h is a deterministic function?
Why is the dual function smooth? Taking the min over a smooth function can yield a non-differentiable function unless assumptions such as strong convexity are used (e.g. like in Nesterov smoothing). It is smooth if $\omega$ is sufficiently large, which I think is the implied assumption but this should be included somewhere explicitly in the assumptions.
+ I think there is a typo on line 486 in appendix, I believe we should have $f_{\omega}(x^{\ast}(y_{t+1};z_{t}), y_{t+1};z_t)$.
+ What are the meaning of the $\rho$ constants in the main Theorem? Looking at the proofs they seem to be related to different Lipschitz constants but this is not clear at all.
+ The stepsize condition (97) is very far in the appendix. It should be brough up earlier.
+ Requiring $g^{(k)}$ to be Lipschitz for all k seems a bit restrictive? I understand that existing works make this assumption but  it seems this would be violated even in the simple convex example $x^2 = x * x$.
+ What does $\nabla f(\cdot,\cdot)$ mean exactly in Assumption 3.1? Do you mean  the gradient of each player holding the other player fixed is Lipschitz with respect to both players?
+ second player has noise variable $\zeta_{t+1}$, is there a reason why we are using a different source of noise here? Also since the dual variable has no multi-level structure why is the gradient estimator for the dual variable biased?
+ Why would it be bad to say take say $K$ independent samples  so that the gradient can be unbiased. Is this possible?  I think  better discussion on the problem setting might answer this question.

**Ethical Concerns:**

["NO or VERY MINOR ethics concerns only"]

**Final Justification:**

Technical issues were clarified by reviewers, there still remains issues on clarity which was also shared with at least one more reviewer.

**Limitations:**

Some of the implied assumptions from the theory can be better stated for example how large should $\omega$ be? and how easy is it to satisfy condition (97) in practice?

**Quality:**

1

**Strengths And Weaknesses:**

## Quality
Unfortunately I cannot verify the technical soundness of the paper. The overall paper is very difficult to follow, below I point out several examples on clarity issues. On the technical side, I cannot verify a Lemma (Lemma B.3). More precisely, I don't see how line (31) is true in the appendix. Applying the difference of squares does not explain this line. You would get something like $\|z'-x\|^2 - \|z-x\|^2 = (\|z'-x\|-\|z-x\|)(\|z'-x\|+\|z-x\|)$ which is not equivalent to what is written. Additionally Lemma B.1 seems incomplete (see below for details).

## Clarity
The paper significantly lacks clarity. There is a lot of discussion motivating the problem and highlighting different challenges on the theoretical side especially in comparison to related setting (non-stocastic min-max or stochastic minimization) but it is very difficult to understand the problem setting and the theoretical results. Below I list some examples:
+ The stochastic oracle is never formally defined. I had to look at other papers to understand why the naive gradient estimator in (5) is biased. My understanding it is because in this setting we are incrementally sampling $\xi_t^{(i)}$ across different $i$, e.g. sample $\xi_t^{(1)}$ then $\xi_t^{(2)}$. Additionally terms like $\hat{\xi}_t$ are never defined.  And in even in equation (6) where the biased is claimed, it is not obvious at this point without reading about the multilevel setting elsewhere.
+ It is also difficult to follow the proofs either because of typos or incomplete reasoning or conditions like on the stepsize that are in the middle of a long proof instead of stated outside  (see below for more details)
+ There are also constants that seem important for the Theory and algorithm that are never discussed such as $\rho_x, \rho_y, \gamma$.

## Significance
The authors do a good job placing where the contribution lies in the multilevel literature. I think there is interest in this community for this contribution. But due to the issues mentioned above the paper should be revised due to clarity issues on both the problem setting and the theory.

## Originality
See above in significance.

---

> ### Author Rebuttal · Authors · 2025-07-26
>
> Thank you for carefully reading our paper. The reviewer's questions can be classified two categories: **writing issues** and **clarification of specific details.** We appreciate your feedback and address your concerns below.
>
> ---
>
> ## **I. Writing issues**
>
> **Problem setting and biased issue:** We provide a formal stochastic problem formulation in Eq. (1), where different noise sources arise naturally: the outer-level function $f$ involves noise $\zeta$, and each inner-level function $g^k$ involves noise ${\xi}^k$. In Eq. (5), the stochastic set  $\hat{\xi}\_{t} = \\{ \xi\_{t}^{(1)}, \cdots, \xi\_{t}^{(K)}, \zeta\_{t}\\}$ is defined implicitly through the second equality. For better clarity, we will make this explicit in the revised version.
>
> The bias issue in Eq.(5) is explained by Eq.(6). Just as shown in Eq. (6), because the $k$-th level function $g^{(k)}$ is a non-linear function, those stochastic estimators in Eq.(6) are biased estimators.  We will revise the paper to clarify this discussion around Eq. (6) for better understanding.
>
> **Follow the proofs:** We have provided the proof sketches for the two theorems in Lines 204–216 and Lines 250–258. These sketches are helpful for understanding the overall structure and intuition behind our proofs.
>
>
> **Other typos or writing issues:** We will carefully revise our writing by following the reviewer's suggestions.
>
> ---
>
> ## **II. Clarification of specific details**
>
> **Lemma B.1:**  The first **three** properties follow from Lemma B.1 in [37]; the last follows from Lemma 4.3 in [22]. We apologize for the typo and any misunderstanding it may have caused.
>
> **Lemma B.3:**
> * In Eq. (31), the second-to-last step uses the vector version of that equation: $||a||^2-\|\|b\|\|^2= < a-b, a+b >$. It is easy to verify it is true: $< a-b, a+b >=\|\|a\|\|^2 +a^Tb-b^Ta-\|\|b\|\|^2=\|\|a\|\|^2-\|\|b\|\|^2$. We will make it clear.
>
> * The dual variable $y$ is updated by _stochastic_ gradient,  so we should take the expectation.
>
> * The smoothness of the dual function can be found in Lemma B.3 of [36].
>
> * The scale of $\omega$ is provided in Line 478. We will make it clear.
>
> **Hyperparameters:** $\gamma_x, \gamma_y, \gamma_z$ are coefficients for learning rate updates, $\rho_x, \rho_y$ are coefficients for momentum updates. O(1) means they are independent of $\epsilon$ or $\kappa$, as can be seen from Eq.(97). In practice, to determine learning rate, we fix the coefficient $\gamma$ to constants and only tune $\eta$. Once $\eta$ is selected, tuning $\rho$ is straightforward since $\rho\eta^2 \in (0, 1)$. Thus, satisfying Eq.(97) is not difficult, and all of our experiments comply with this condition.
>
> **Assumptions:**
> * Lipschitz continuity of $g^k$ is common and not restrictive in non-convex multi-level optimization [5, 19, 31, 35]. To cover more cases, such as that the reviewer mentioned, we can consider the generalized Lipschitz condition, e.g., [R1], and we will leave it for future work.
>
>
>
> * The smoothness of $\nabla f(\cdot, \cdot)$ in Assumption 3.1 is also a common assumption in compositional minimax optimization [9, 13, 23]. Its formal definition is: $\|\nabla\_1 f(x\_1, y\_1)-\nabla\_1 f(x\_2, y\_2)\| \leq l[\|x\_1-x\_2\|+\|y\_1-y\_2\|]$, $\|\nabla\_2 f(x\_1, y\_1)-\nabla\_2 f(x\_2, y\_2)\| \leq l[\|x\_1-x\_2\|+\|y\_1-y\_2\|],$
>
>
> [R1] Li, Haochuan, et al. "Convex and non-convex optimization under generalized smoothness." Advances in Neural Information Processing Systems 36 (2023): 40238-40271.
>
> **Second player has noise variable**: Because each level function has its own source of noise, we use $\zeta$ to differentiate the noise source of the dual variable from those levels of functions.  For the stochastic gradient with respect to the dual variable: $\nabla\_y f(\hat{G}_t, y\_t ; \zeta_t)$ where $\hat{G}\_t$ is the biased stochastic estimator of $G(x_t)$ as discussed in Eq.(6),  it is obvious that $\mathbb{E}[\nabla\_y f(\hat{G}_t, y\_t ; \zeta_t)] \neq \nabla\_y f(G(x\_t), y\_t )$.
>
> **The gradient can be unbiased. Is this possible?** When each level function $g^{(k)}$ is a linear function, it is possible. But we are not aware of such applications in machine learning, so our focus is to study the more general setting that can benefit real-world machine learning models.
>
> ---
> As shown above, **the reviewer’s questions primarily concern writing issues and clarification of specific details. We believe these points are minor and can be easily addressed, hence not affecting the technical contribution of our paper**.
>
> Moreover, we would like to emphasize that all theoretical results are correct and supported with complete proofs. Additionally, we provide high-level proof sketches for both algorithms in lines 204–216 and 250–258 for better understanding. To further improve clarity, we will add more explanations of the problem setting and key equations for readers. Given that the reviewer has acknowledged the significance and originality of our work, we respectfully ask for a reconsideration of the overall evaluation.

---

> > ### Author Response · Authors · 2025-08-02
> >
> > Dear Reviewer 7Drm,
> >
> > Thank you for carefully reviewing our paper and rebuttal.
> >
> > If you have any further questions, we would be happy to address them.
> >
> > Best regards,
> >
> > Authors of 18077

---

> > ### Comment · Reviewer_7Drm · 2025-08-03
> > **Response to Rebuttal**
> >
> > Thank you for the thorough response. They have clarified my doubts on parts of the theory I have mentioned. I will increase my score accordingly. However, I disagree that the points around clarity of the technical details and theory as minor issues. Reviewer GHtS also mentioned "The proofs are hard to follow; many inequalities need further explanation."
> >
> > I understand there are proof sketches but several important details are hidden in the appendix and not easy to find, which was also mentioned by other reviewers, such as:
> > + Choice of $\omega$
> > + Stepsize condition line (97)
> > + Other hyperparameters for the algorithm
> >
> > Of course these can be adjusted, but I don't think these are just minor details. I would suggest to refer to stepsize condition 97 or at least one example of hyperparameters in the paper with explicit bounds/choices. I think such improvements would greatly improve the clarity of the results and discussion.

---

> ### Author Response · Authors · 2025-08-05
>
> Thank you very much for the feedback and increasing your score! We address your additional concerns below.
>
> # I. Question: several important details are hidden in the appendix
>
> Our answer: We respectfully disagree with this argument. We have **already included the scale of all hyperparameters in Corollary 4.2**.
>
> **Providing the scale of hyperparameters in the main text while deferring detailed values to the appendix is a common practice in optimization papers**. This strategy is widely adopted in optimization papers. For example, in an ICML 2025 spotlight paper [1], when presenting the convergence rate in Theorem 2, the authors do not provide the exact value of the learning rate in the main text, instead deferring the detailed values to the appendix.
> Actually, **presenting only the scale in the main body allows authors to emphasize key insights  without overwhelming readers with numerical detail**, while placing the full hyperparameter settings in the appendix provides transparency for those interested in implementation.
>
> Our paper follows this common practice by providing the scale of hyperparameters in the main text and deferring the detailed values to the appendix, in order to enhance readability. Therefore, **this should not be considered a limitation of our paper**.
>
> [1]  Achieving Linear Speedup and Near-Optimal Complexity for Decentralized Optimization over Row-stochastic Networks, ICML 2025 spotlight.
>
> ---
>
> # II. Question: The proofs are hard to follow
>
> Our answer: In lines 204–216 and 250–258, we provide the proof sketch of our two algorithms. For example, in the proof sketch of our first algorithm, Eq.(9) shows our developed potential function. It contains four terms:
>
> * The first term $\mathcal{P}\_{t}$ is about the optimization error caused by the smoothed technique.
>
> * The second and third terms are to bound the error caused by the stochastic gradient regarding $x$ and $y$.
>
> * The last term is to bound the error caused by the stochastic inner-level function.
>
> Then, we only need to bound each term in the potential function, as described in Lines 208–212. By **following this structure, the proof becomes easier to follow**. In addition, **regarding the hyperparameters, we provide detailed steps for deriving their bounds**.  Particularly, in Eqs.(58-96), we provide very detailed steps for obtaining the upper bound of hyperparameters.  Therefore, **both a high-level sketch and detailed derivations are included in our proof. With these elements in place, we are confident that the proof is not hard to follow**.
>
> ---
> Thank you again for increasing your score! We hope our new response has fully addressed your concerns, and we sincerely hope the reviewer will continue to endorse our paper.

---

> > ### Author Response · Authors · 2025-08-05
> >
> > Regarding the reviewer's new concern about the clarity of our proof, we provide a more detailed explanation.
> >
> > # I. The proof of Theorem 4.1.
> >
> > As shown in Eq. (9), we propose a novel potential function $\mathcal{H}_{t}$. Then, we prove the Theorem 4.1 based on this potential function.
> >
> > * First, we bound first term $\mathcal{P}\_t=f\_{\omega}(G({x}\_{t}), {y}\_{t}; {z}\_{t}) - 2 h\_{\omega, d}({y}\_{t}; {z}\_{t}) + 2h({z}\_{t})$. **This term is about the optimization error caused by the smoothed technique**. To prove the convergence rate, we need to study how each term in $\mathcal{P}\_t$ changes when updating $x$, $y$, and $z$ and how different terms affect each other. This is challenging because each term in $\mathcal{P}\_t$ **is affected by $x$, $y$, and $z$ simultaneously** and the **compositional gradient is a biased estimator**. Our proof addresses this challenge with the following procedure:
> >
> >     * We first establish the upper bound for $f\_{\omega}(G({x}\_{t}), {y}\_{t}; {z}\_{t})$ in Lemma B.5, $-h\_{\omega, d}({y}\_{t}; {z}\_{t})$ in Lemma B.3 (Eq.(27)),  and $h({z}\_{t})$ in Lemma B.3 (Eq.(28)).
> >
> >     * Then, to study the dependence among those three terms, we combine the upper bound of those three terms  in  Appendix B.2.1, where we provide very detailed steps to identify the dependence among the upper bounds of $f\_{\omega}(G({x}\_{t}), {y}\_{t}; {z}\_{t})$, $-h\_{\omega, d}({y}\_{t}; {z}\_{t})$,  and $h({z}\_{t})$.
> >
> > * Second, the potential function in Eq. (9) has three additional terms. **All these three terms are caused by the multi-level compositional loss function**.  Specifically: $\mathbb{E}[\| p_{t} - \nabla_x f_{\omega}(H(x_{t}), y_{t}; z_{t})  \|^2]$ is the error caused by the stochastic compositional gradient regarding $x$, $\mathbb{E}[\| q_{t} -\nabla_y f_{\omega}(H(x_{t}), y_{t}; z_{t}) \|^2]$ is the error caused by the stochastic compositional gradient regarding $y$, and $\sum\_{k=1}^{K}\lambda\_k\mathbb{E}[\| {h}\_{t}^{(k)}  - g^{(k)}({h}\_{t}^{(k-1)} )  \|^2]$ is the error caused by the stochastic inner-level function. Then, in Lemmas B.6, B.7, and B.4, we provide their upper bounds by carefully handling the compositional structure.
> >
> > * Third, **the four terms in the potential function $\mathcal{H}_{t}$ affect each other**. For example, Eq.(56) shows $\mathcal{P}\_t$ is affected by all the other three terms in $\mathcal{H}_{t}$. Then, **when combining them together to constitute $\mathcal{H}_{t}$, we need to precisely figure out how they affect each other**. In Section B.2.2, we provide very detailed steps for this procedure.
> >
> > This is the high-level idea of our proof, which has been discussed in Lines 204-216 of our original paper.
> >
> > ---
> > # II. The proof of Theorem 5.1.
> > The proof of  Theorem 5.1 is also challenging, as we need to handle the stage-wise structure in our algorithm 2.  The high-level idea of this proof is using the induction approach to handle the stage-wise structure. The details are shown below.
> >
> > * First, we introduce two metrics in Eq. (11), where $\mathcal{V}\_t$ denotes the optimization error regarding both $x$ and $y$, and $\mathcal{U}\_t$ is similar to the last three terms of Eq.(9).  Then, we provide the upper bound of $\mathcal{V}\_t$ in Lemma C.3 and the upper bound of $\mathcal{U}\_t$ in Lemma C.5. Importantly, based on these two lemmas, we **figure out how $\mathcal{V}\_t$ and $\mathcal{U}\_t$ affect each other  across stages as shown in Eq.(12). This is a very critical step in our proof**.
> >
> > * Second, **with the relationship between $\mathcal{V}\_t$ and $\mathcal{U}\_t$ in two consecutive stages, we use the induction approach to prove Theorem 5.1 in Section C.2**, where we also provide very detailed steps to show how to set the hyperparameter to perform the induction procedure.
> >
> >
> > This is the high-level idea of our second proof, which has been discussed in Lines 250-257 of our original paper.
> >
> >
> > ---
> > We would like to clarify that, with the help of our proof sketch, the full proof is not hard to follow. We sincerely hope the reviewer will consider the novelty of both our algorithm and proof, and kindly consider increasing your score. Thank you once again for reviewing our paper and providing insightful feedback!

---

> > > ### Author Response · Authors · 2025-08-07
> > >
> > > Dear Reviewer 7Drm,
> > >
> > > Thank you again for your new feedback.
> > >
> > > As the discussion period will end tomorrow,  we would like to check whether our answer has addressed your concerns. If you have any further questions, we would be happy to address them.
> > >
> > > If our clarification addresses all of your concerns, we would be truly grateful if you could consider a higher rating. Thank you again!
> > >
> > > Best regards,
> > >
> > > Authors of 18077

---

### Decision · Program_Chairs · 2025-09-17

**Decision:**

Accept (poster)

**Comment:**

This work focuses on stochastic, multi-level compositional (in the language of some earlier work, compositional-on-x) min-max optimization by assuming nonconvexity on one side and PL on the other side. The proposed method has complexity $O(\kappa^{3/2} \epsilon^{-3})$ in the nonconvex-PL case for finding a two sided $(\epsilon, \epsilon/\kappa)$ stationary point where the two sided refers to the norms of gradients w.r.t. $x$ and $y$. Since the earlier work focused on $\epsilon$ stationarity of the objective function which is given in the $\max$ form, the authors specialize to PL-PL case and show $O(\kappa^{3/2} \epsilon^{-3})$ complexity for $\epsilon$-stationarity and then argue that they improve the earlier complexity by $O(\kappa^{3/2})$ whereas the earlier work focused on nonconvex-PL case (a more general case than PL-PL) and showed $\epsilon$-stationarity.

Here, the condition number $\kappa$ is defined as the ratio of the Lipschitz constant and the PL constant.

The reviewing team found the studied problem and the complexity improvement interesting, however significant concerns remain about the writing of the paper. In particular, the reviewers and the AC shares concerns on the difficulty of following the developments in the paper and the proofs in the appendix.

Below are some remarks from the private discussion between myself and the reviewers:

- "In my view, the appendices need to be revised, and it could take considerable time and effort to produce fully justified proofs. I agree with the AC that the authors did not make a significant effort to make it easy for readers to verify their arguments."

- "I agree that it is difficult to follow the proofs, I was not able to verify every step. I got stuck on multiple parts, and though the authors addressed the parts for which I got stuck, to be honest, I would most likely have to spend another few days just to go over all the proofs again. Moreover, I am not satisfied in the author's responses defending sufficient clarity. For example, their main arguments were:

- We have a proof sketch

- This other paper does is similarly

- Although a proof sketch is nice, for me it has not helped. I know at least one other reviewer (GHtS) mentioned clarity issues, maybe they also agree.

For the second point, this is in the context of so many parameters and variables that are not well explained to me in the paper. Moreover, important stepsize conditions like (97) in the appendix are quite hidden. The authors' response was that we only give $O$ scale for such parameters. Another example is the  momentum constant $\rho$, which are featured everywhere in the Theorems but I believe there is no discussion on them. They are technically in the pseudocode of Algorithm 1 but they should be atleast discussed in detail if they are main hyperparameters featured in the Theorems. Maybe I missed this discussion somewhere but I can't currently find anything on it."

I also wish to emphasize my agreement with the reviewer that just because another paper from ICML 2025 presents hyperparameters in terms of their orders in the main text and in their detailed form in the appendix **does not justify anything**. The authors should not use such justifications. It seems appropriate to acknowledge that the concern of the reviewer is fair (I also totally agree with this concern) but unfortunately the analysis is extremely technical and hence the conditions at Eq. (97) are not very readable.

- "For example, looking at the estimations in pages 18, 22, 23, 28, 31, 32, 33, 36, 37, 38, 39, 40 (and honestly, most of the pages in the 30 page appendix), it looks like the authors unfortunately did not give much effort to make their proof readable."

As a result, the reviewing team has significant concerns on the lack of clarity in the writing of this paper. Given that the authors have the full responsibility in making their main text and proofs readable and convincing to the readers, I find the lack of clarity a significant problem, especially for a theoretical paper with 30 pages of proofs. As a result **the authors need to spend a considerable amount of time to streamline their proof**. They should avoid long equality chains in the pages referenced above and make sure to have ample justification for each of their proof steps. **To be published, a proof have to be readable.** Otherwise, neither the reviewers nor the readers of the paper can verify or understand the proof: **the authors should do every effort to make their proof much more readable than its current state.**

- Another issue is that the presentation of the contribution is a bit misleading. In particular, in Remark 5.2, the authors say

> To the best of our knowledge, this is the first algorithm to achieve an $O(\kappa^{3/2})$ dependence for (multi-level compositional) minimax problems under the nonconvex-PL setting.

which is technically correct but the authors are unfortunately not acknowledging here that they actually need to assume PL in the primal to get this result. That is, even thought technically a PL-PL problem is still a nonconvex-PL problem, it is a significantly smaller class than nonconvex-PL problems. Indeed the complexity that the authors are comparing do not assume PL in the primal, [23] seems to only assume smoothness in the primal. The authors should make this distinction **crystal clear**. I find the comparison right now unfairly presented with respect to [23].

As it stands, it is not clear if there is a strict improvement compared to the state-of-the-art in the 2-level case. The comparison is either with a different stationarity measure (Section 4) and hence not comparable to other works or the comparison is with different assumptions (Section 5), that is the authors assume PL in the primal. On the other hand, this work is more general because they are allowing more than 2-level composition.

As a result, the authors should make this comparison very clear and stop claiming strict improvement. From what I see, their work is complementing the state-of-the-art, generalizing in some senses, but more restrictive in other senses. Hence a direct comparison is not possible.

Yet, the smoothing idea and the result is potentially interesting to the line of research in this direction.